# Complete sequencing of ape genomes

DongAhn Yoo[1], Arang Rhie[2], Prajna Hebbar[3], Francesca Antonacci[4], Glennis A. Logsdon[1,5], Steven J. Solar[2], Dmitry Antipov[2], Brandon D. Pickett[2], Yana Safonova[6], Francesco Montinaro[4,7], Yanting Luo[8], Joanna Malukiewicz[9,10], Jessica M. Storer[11], Jiadong Lin[1], Abigail N. Sequeira[12], Riley J. Mangan[13,14,15], Glenn Hickey[3], Graciela Monfort Anez[16], Parithi Balachandran[17], Anton Bankevich[6], Christine R. Beck[11,17,18], Arjun Biddanda[19], Matthew Borchers[16], Gerard G. Bouffard[20], Emry Brannan[21], Shelise Y. Brooks[20], Lucia Carbone[22,23], Laura Carrel[24], Agnes P. Chan[25], Juyun Crawford[20], Mark Diekhans[3], Eric Engelbrecht[26], Cedric Feschotte[27], Giulio Formenti[28], Gage H. Garcia[1], Luciana de Gennaro[4], David Gilbert[29], Richard E. Green[30], Andrea Guarracino[31], Ishaan Gupta[32], Diana Haddad[33], Junmin Han[34], Robert S. Harris[12], Gabrielle A. Hartley[11], William T. Harvey[1], Michael Hiller[35,36,37], Kendra Hoekzema[1], Marlys L. Houck[38], Hyeonsoo Jeong[1], Kaivan Kamali[12], Manolis Kellis[13,14], Bryce Kille[39], Chul Lee[40], Youngho Lee[41], William Lees[26,42], Alexandra P. Lewis[1], Qiuhui Li[43], Mark Loftus[44,45], Yong Hwee Eddie Loh[46], Hailey Loucks[3], Jian Ma[47], Yafei Mao[34,48,49], Juan F. I. Martinez[6], Patrick Masterson[33], Rajiv C. McCoy[19], Barbara McGrath[12], Sean McKinney[16], Britta S. Meyer[9], Karen H. Miga[3], Saswat K. Mohanty[12], Katherine M. Munson[1], Karol Pal[12], Matt Pennell[50], Pavel A. Pevzner[32], David Porubsky[1], Tamara Potapova[16], Francisca R. Ringeling[51], Joana L. Rocha[52], Oliver A. Ryder[38], Samuel Sacco[30], Swati Saha[26], Takayo Sasaki[29], Michael C. Schatz[43], Nicholas J. Schork[25], Cole Shanks[3], Linnéa Smeds[12], Dongmin R. Son[53], Cynthia Steiner[38], Alexander P. Sweeten[2], Michael G. Tassia[19], Françoise Thibaud-Nissen[33], Edmundo Torres-González[12], Mihir Trivedi[1], Wenjie Wei[54,55], Julie Wertz[1], Muyu Yang[47], Panpan Zhang[27], Shilong Zhang[34], Yang Zhang[47], Zhenmiao Zhang[32], Sarah A. Zhao[13], Yixin Zhu[50], Erich D. Jarvis[40,56], Jennifer L. Gerton[16], Iker Rivas-González[57], Benedict Paten[3], Zachary A. Szpiech[12], Christian D. Huber[12], Tobias L. Lenz[9], Miriam K. Konkel[44,45], Soojin V. Yi[53,58], Stefan Canzar[51], Corey T. Watson[26], Peter H. Sudmant[52,59], Erin Molloy[60], Erik Garrison[31], Craig B. Lowe[8], Mario Ventura[4], Rachel J. O'Neill[11,18,21 ✉], Sergey Koren[2], Kateryna D. Makova[12 ✉], Adam M. Phillippy[2 ✉] & Evan E. Eichler[1,56 ✉]

The most dynamic and repetitive regions of great ape genomes have traditionally been excluded from comparative studies[1–3]. Consequently, our understanding of the evolution of our species is incomplete. Here we present haplotype-resolved reference genomes and comparative analyses of six ape species: chimpanzee, bonobo, gorilla, Bornean orangutan, Sumatran orangutan and siamang. We achieve chromosome-level contiguity with substantial sequence accuracy (<1 error in 2.7 megabases) and completely sequence 215 gapless chromosomes telomere-to-telomere. We resolve challenging regions, such as the major histocompatibility complex and immunoglobulin loci, to provide in-depth evolutionary insights. Comparative analyses enabled investigations of the evolution and diversity of regions previously uncharacterized or incompletely studied without bias from mapping to the human reference genome. Such regions include newly minted gene families in lineage-specific segmental duplications, centromeric DNA, acrocentric chromosomes and subterminal heterochromatin. This resource serves as a comprehensive baseline for future evolutionary studies of humans and our closest living ape relatives.

High-quality sequencing of ape genomes has been a high priority of the human genetics and genomics community since the initial sequencing of the human genome in 2001 (refs. 1,2). Sequencing of these genomes is crucial for reconstructing the evolutionary history of every base pair of the human genome. Moreover, such an undertaking is one of the grand challenges put forward to the genomics community after the release of the first draft of the Human Genome Project[3]. As a result, there have been numerous publications ranging from initial draft genomes to notable updates over the past two decades[4–8]. However, owing to the repetitive nature of ape genomes, complete assemblies have not been achieved. Current references lack sequence resolution of some of the most dynamic genomic regions, including regions corresponding to lineage-specific gene families.

Advances in long-read sequencing and new assembly algorithms were needed to overcome the challenge of repeats and to achieve the first complete, telomere-to-telomere (T2T) assembly of the human

## Table 1 | Summary of ape genome assemblies

| Sample information | | | | Assembly stats (v.2.0) Haplotype 1 (haplotype 2) or maternal (paternal) | | | | | |
|---|---|---|---|---|---|---|---|---|---|
| Common name (abbreviation) | Scientific name | Tissue | Sex | Accession | Total no. of bases (Gb) | Contig N50 (Mb) | No. of T2T contigs | No. of non-rDNA gaps or missing telomeres | QV |
| Chimpanzee (PTR) | *Pan troglodytes* | Lymphoblastoid | Male | PRJNA916736 | 3.14 (3.03) | 146.29 (140.84) | 19 (17) | 0 (2) | 66.0 |
| Bonobo (PPA)[a] | *Pan paniscus* | Fibroblast or lymphoblastoid | Male | PRJNA942951 | 3.21 (3.07) | 147.03 (147.48) | 20 (19) | 0 (0) | 62.7 |
| Gorilla (GGO)[a] | *Gorilla gorilla* | Fibroblast | Male | PRJNA942267 | 3.55 (3.35) | 151.43 (150.80) | 19 (22) | 4 (0) | 61.7 |
| Bornean orangutan (PPY) | *Pongo pygmaeus* | Fibroblast | Male | PRJNA916742 | 3.16 (3.05) | 140.59 (137.91) | 15 (13) | 1 (2) | 65.8 |
| Sumatran orangutan (PAB) | *Pongo abelii* | Fibroblast | Male | PRJNA916743 | 3.17 (3.08) | 146.20 (140.60) | 14 (13) | 2 (3) | 63.3 |
| Siamang (SSY) | *Symphalangus syndactylus* | Lymphoblastoid | Male | PRJNA916729 | 3.24 (3.12) | 146.71 (144.67) | 24 (20) | 0 (5) | 66.4 |
| Average | | | | – | 3.24 (3.12) | 146.38 (143.72) | 18.5 (17.3) | 1.2 (2) | 64.3 |

[a]Sample with parental sequence data. Contig N50 is the shortest contiguous assembled fragment length that contains 50% of the genome (values approximate the length of chromosomes). QV represents the accuracy score from the Illumina–HiFi hybrid-based approach. Numbers in parentheses denote the values for haplotype 2 or the paternal haplotype.

genome[9,10]. Using these same methods, we recently published six additional pairs of complete sex chromosomes from distinct branches of the ape phylogeny[11]. Although these initial projects targeted haploid chromosomes and required substantial manual curation, improved assembly methods now enable the complete assembly of diploid chromosomes[12,13]. Using these methods, we present here complete, phased, diploid genomes of six ape species and make all data and curated assemblies freely available to the scientific community. We organize the article into three main sections, with a focus on the following topics: (1) the generation of the assemblies; (2) the added value for standard evolutionary analyses; and (3) new evolutionary insights into previously unassembled regions. Although the interior of rDNA arrays and some small portions of the largest centromeres remain unresolved, these genomes represent an improvement in quality over previous ape reference genomes and are of equivalent quality to the T2T-CHM13 human reference genome.

## Ape genome assembly

Unlike previous reference genomes that selected female individuals to improve representation of the X chromosome[4–7], we focused on male samples (Table 1) to also sequence the Y chromosome and to provide a complete chromosomal complement for each species. Both sex chromosomes from these samples were analysed by us in a separate article[11]. Samples from two of the species we analysed, bonobo and gorilla, originated from parent–child trios (Supplementary Note I), which facilitated the phasing of parental haplotypes. For samples for which parental data were not available, deep Hi-C datasets were used (Table 1) to achieve chromosome-scale haplotype phasing. For all samples, we prepared high-molecular-weight DNA and generated deep PacBio high-fidelity (HiFi; mean = 90-fold sequence coverage) and Oxford Nanopore Technologies (ONT; mean = 136.4-fold sequence coverage) sequence data (Supplementary Table II.3). For the latter, we specifically focused on producing at least 30-fold ultra-long (UL > 100 kilobases (kb)) ONT sequence data to scaffold assemblies across larger repetitive regions, including centromeres and segmental duplications (SDs). To generate the backbone of the assembly, we used Verkko[12] (v1.4.1), which is a hybrid assembler that leverages the accuracy of HiFi data (Supplementary Note II). UL-ONT sequencing was used for repeat resolution, local phasing and scaffolding, and Hi-C or trio data were used to convert chromosome-scale phasing of haplotypes into a fully diploid assembly. For each chromosome, the most accurate and complete haplotype was selected as the primary assembly for each chromosome pair. When rDNA was present in only one haplotype, it was chosen as the primary haplotype regardless of the completeness status. Both sex chromosomes and the mitochondrial genome were included in the primary assembly.

When considering the diploid genomes of each species, 74% (215 out of 290) of all chromosomes were T2T assembled (gapless with a telomere on both ends), and at least 80.8% of chromosomes were T2T in at least one haplotype (Table 1, Fig. 1, Extended Data Fig. 1 and Supplementary Table II.4 for siamang). Overall, there was an average of six gaps or breaks in assembly contiguity per haplotype (range = 1–12), which were typically localized to the rDNA array. The number was reduced to an average of 1.6 gaps if those regions were excluded. All assemblies were curated to extend partially into each rDNA array from both sides, which ensured that no non-rDNA sequence was missed. In addition to gaps, we specifically searched for collapsed and misassembled sequences using dedicated methods (Supplementary Note II ('Assessment of the genome assembly') and Supplementary Table II.5). We estimated on average 0.2–1.6 megabases (Mb) of non-rDNA-related issues per haplotype assembly (Supplementary Table II.5). Comparisons with Illumina data[14] from the same samples provided a lower-bound base QV accuracy of 49.3 (about 1 error every 100,000 bp), which was limited by Illumina coverage loss in high-GC regions. By contrast, comparisons that included HiFi data suggested even higher accuracy (QV = 61.7, about 1 error every 1 million base pairs; Supplementary Table II.3). Overall, we estimated that 99.2–99.9% of each genome (99.98–99.99% for non-rDNA bases) was completely and accurately assembled, including heterochromatin. These values are consistent with the T2T-CHM13v1.1 assembly, for which potential issues remain for 0.3% of the genome[14]. In summary, these ape diploid-genome assemblies represent an advancement in terms of sequence accuracy and contiguity with respect to all previous ape genome assemblies[4–8]. Each genome assembly was annotated by the National Center for Biotechnology Information (NCBI) and has been adopted as the main reference in RefSeq. That is, the previous short-read-based or long-read-based and less complete versions of the genomes have been replaced, and the sex chromosomes have been updated with the newly assembled and polished versions.

This new genomics resource provides two important advantages to the evolutionary community. First, it enhances the ability to perform more refined and complete standard evolutionary comparative analyses than previously possible. Second, it provides access to the complete sequence of previously inaccessible regions, including regular centromeric and acrocentric regions, large blocks of SDs and subterminal

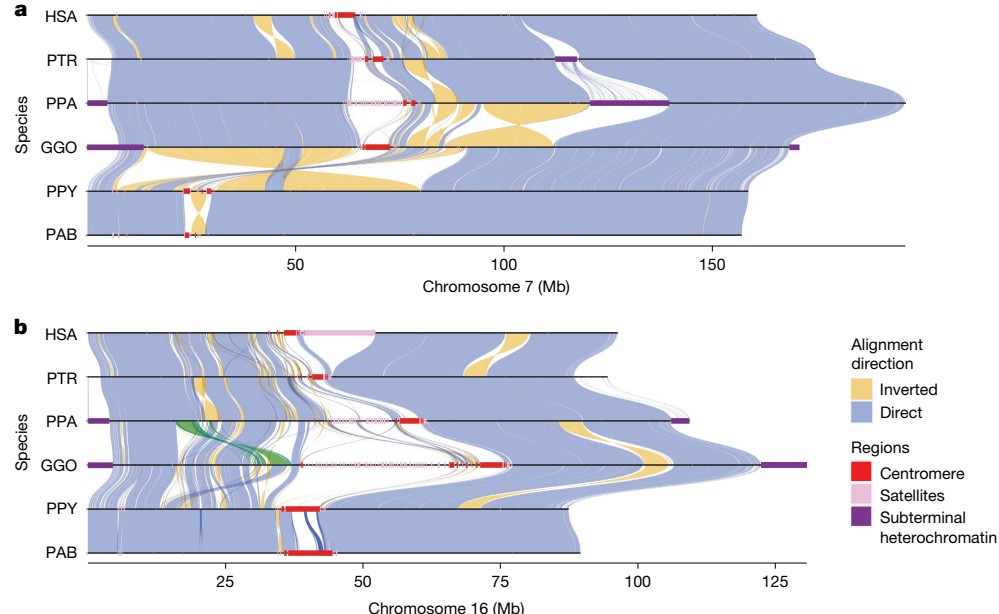

**Fig. 1 | Chromosomal-level assembly of complete genomes for great apes.**
**a**,**b**, A comparative alignment of HSA7 (**a**) and HSA16 (**b**) compared with syntenic chromosomes from chimpanzee, bonobo, gorilla, Bornean orangutan and Sumatran orangutan. Each chromosome is compared with the chromosome below in this stacked representation using the tool SVbyEye (https://github.com/daewoooo/SVbyEye)[116]. Regions of collinearity and synteny (positive in blue) are contrasted with inverted regions (negative in yellow) and with regions beyond the sensitivity of minimap2 (homology gaps), including centromeres, subterminal and interstitial heterochromatin or other regions of satellite expansion. A single transposition (green in **b**) relocates about 4.8 Mb of gene-rich sequence in gorilla from HSA16p13.11 to HSA16p11.2 (see Fig. 3 for more detail).

heterochromatin. Below, we divide our analyses into two sections: the first focused on highlighting improvements in the resource and the second focused on evolutionary analyses of previously inaccessible regions.

## Highlights of improvements in the resource

### Ape pangenome

Most previous genome-wide comparative studies of apes have been limited by the mapping of inferior assemblies to a higher quality human genome. Consequently, human reference biases were introduced. Because the assemblies here were of comparable quality and contiguity, we sought to mitigate this bias by creating an ape pangenome as an unbiased framework to understand evolution. First, we used Progressive Cactus[15] to construct several pangenomes using different parameters, which included human and ape haplotypes (Supplementary Note III and Data availability). Predictably, the resulting interspecies graphs were more complex than the recently released human pangenome for 47 individuals[16]. For example, one ape pangenome was 3.38 Gb, with three times the number of edges and nodes as compared to previous human-only pangenomes. The resulting alignments enabled us to annotate genes and repeats and the ancestral state of more base pairs of the human GRCh38 reference. After applying the parsimony-like method used by the 1000 Genomes Project and Ensembl[17], we observed a genome-wide increase of 6.25% in the total ancestrally annotated base pairs compared with the existing Ensembl annotation (release 112), with the greatest increase in autosomes observed for chromosome 19 (21.5%; Supplementary Fig. III.13). As a second approach, we applied pangenome graph builder[18] to construct all-to-all pairwise alignments for all 12 nonhuman primate (NHP) haplotypes (of 6 species) along with 3 T2T human haplotypes (T2T-CHM13v2.0 and 2 haplotypes of T2T-HG002v1.0). We used these pairwise alignment data to construct an implicit graph (Supplementary Note III, 'Implicit pangenome graph') of all six species and computed a conservation score using phastCons v1_5 for every base pair in the genome (Supplementary Note III ('Conservation analyses') and Supplementary Fig. III.9). The approach was

transitive without a reference bias and considered both assembled haplotypes for each genome and unique and repetitive regions. As a result, we identified the most rapidly evolving regions in each primate lineage, including the major histocompatibility complex (MHC) and the chromosome 8p23.1 inversion (Supplementary Fig. III.9).

### Divergence and selection

Overall, sequence comparisons among the complete ape genomes revealed greater divergence than previously estimated (Supplementary Notes III–IV). Indeed, 12.5–27.3% of an ape genome failed to align or was inconsistent with a simple one-to-one alignment, thereby introducing gaps. Gap divergence showed a 5-fold to 15-fold difference in the number of affected megabases when compared to single-nucleotide variants, which was due to rapidly evolving and structurally variant regions of the genome as well as technical limitations of alignment in repetitive regions (Supplementary Figs. III.11 and III.12). We catalogued all structurally divergent regions (SDRs) among the ape genomes and found an average of 327 Mb of sequence (10%) per ape lineage (Fig. 2a and Supplementary Note V). Predictably, these included centromeres, acrocentric short arms and subterminal heterochromatic caps, but also numerous gene-rich SD regions enriched at the breakpoints of large-scale rearrangements (Supplementary Fig. V.15). We focused on segments that could be reliably aligned and then we estimated speciation times and modelled incomplete lineage sorting (ILS) across the ape species tree[19] (Fig. 2b and Supplementary Table VI.26). Our analyses dated the human–chimpanzee split between 5.5 and 6.3 million years ago (Ma; minimum to maximum estimate of divergence), the African ape split at 10.6–10.9 Ma and the orangutan split at 18.2–19.6 Ma (Fig. 2a). We inferred ILS for an average of 39.5% of the autosomal genome and 24% of the X chromosome. Compared with recent reports[20], these values represent a 7.5% increase in autosomal ILS, results that are in part due to the inclusion of more repetitive DNA (Supplementary Fig. VI.16). Consequently, we estimated that the human–chimpanzee–bonobo ancestral population size (average $N_e$ = 198,000) is larger than that of the human–chimpanzee–gorilla ancestor ($N_e$ = 132,000), a result

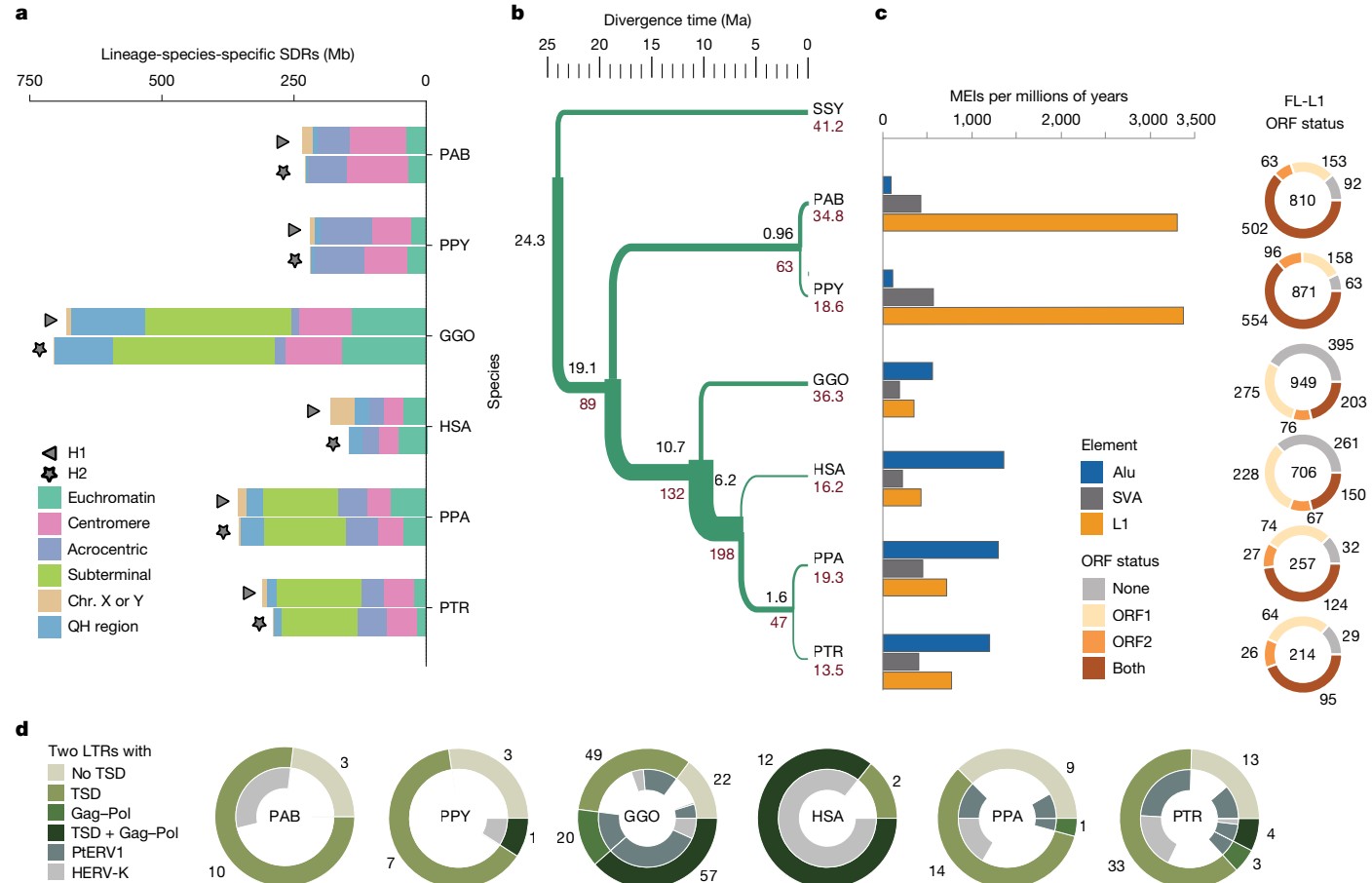

**Fig. 2 | Divergent regions and repeats. a**, The number of lineage-specific SDRs detected on two haplotypes (H1 and H2) and classified by different genomic content. QH region, heterochromatic q-banded satellite-containing region. **b**, Ape phylogeny with speciation times in millions of years (black) and effective population size ($N_e$) in thousands (red) for terminal and ancestral nodes. For $N_e$, values in inner branches refer to TRAILS estimates, whereas that of terminal nodes is predicted using MSMC2 considering the harmonic mean of the $N_e$ after the last inferred split. **c**, Left, species-specific *Alu*, SVA and L1 mobile element insertion (MEI) counts normalized by millions of years (using speciation times from **b**). Right, species-specific full-length (FL) L1 ORF status. The inner number in each circle represents the absolute count of species-specific FL L1 elements. **d**, Species-specific ERV retrotransposons depict PtERV1 and HERV-K elements (inner ring) and long terminal repeats (LTRs) and potential protein-coding domains (outer ring). TSD, target-site duplication.

consistent with an increase in the ancestral population 6–10 Ma (Supplementary Note VI). We next searched for signatures of adaptation in each ape species by identifying regions of hard[21] and soft (partial)[22] selective sweeps based on the mapping of short-read whole-genome sequencing data generated from the great ape genetic diversity project[23–26] (Supplementary Note VII) back to the T2T genomes. Among the ape species, we identified 143 and 86 candidate regions for hard and partial selective sweeps, respectively (Supplementary Table VII.31). We also identified seven duplication-overlapping genes that showed signatures of selection on both of the unique flanking sequences of the duplications. Approximately half of the hard selective sweeps (74 out of 143) and more than 80% of the partial selective sweeps (70 out of 86) were previously unknown, and a total of 43 regions overlapped with sweeps previously found in humans[27].

## Gene annotation

We applied two gene-annotation pipelines (comparative annotation toolkit (CAT) and NCBI) to identify both protein-coding and noncoding RNA (ncRNA) genes for primary assembly for each NHP. We complemented the annotation pipelines by directly mapping the PacBio Iso-Seq transcriptome long reads (50 Gb of full-length non-chimeric (FLNC) cDNA) generated from each sample and searching for transcriptional support for genes with multiple exons. The number of protein-coding genes was comparable among different apes ($n$ = 22,114–23,735),

with slightly more than 1,000 genes predicted to be gained or duplicated or lost when compared to humans (Supplementary Table VIII.34). Using the University of California, Santa Cruz (UCSC) gene set, based on GENCODE[28], we estimated that 99.0–99.6% of corresponding human genes were now represented, and >90% of genes were full-length. We identified a fraction (3.3–6.4%) of protein-coding genes present in the NHP T2T genomes that contained new transcript models relative to the annotation for the human genome. This included 770–1,482 previously unknown gene models corresponding to 315–528 gene families in the NHPs, with about 68.6% corresponding to lineage-specific SDs, all supported by Iso-Seq transcripts (Supplementary Tables VIII.34 and VIII.35). In this list, we also found biomedical genes such as *LRPAP1*, for which paralogues show tissue-specific expression across fibroblast and testes (Supplementary Note VIII). We also identified human-specific genes associated with human evolution of the frontal cortex, including *SRGAP2C*[29,30], *ARHGAP11*(ref. 31) (Supplementary Table VIII.40) and *NOTCH2NL*[32,33] (implicated in neuronal intranuclear inclusion disease). The non-SD overlapping gene copies consisted of 73% of transcript models that were changed in their sequence by >50%, with the remaining 27% no longer translated. Moreover, 2.1–5.2% of transcripts showed previously unknown NHP splice forms, which were again supported by Iso-Seq data (Supplementary Table VIII.34). We provide a valuable resource in the form of a curated consensus protein-coding gene annotation set by integrating both the NCBI and CAT pipelines

(Supplementary Note VIII). We also analysed FLNC reads obtained from testis from a second individual[11] to quantify the potential impact genome-wide on gene annotation, and we observed improvements in mappability, completeness and accuracy (Supplementary Note VIII and Supplementary Fig. VIII.29). In gorilla, for example, we mapped 28,925 (0.7%) additional reads to the T2T assembly in contrast to only 171 additional reads to the previous long-read assembly[5]. Similarly, we observed 33,032 (0.7%) soft-clipped reads (>200 bp) in the gorilla T2T assembly in contrast to 89,498 (2%) soft-clipped reads when mapping to the previous assembly[5]. These improvements in mappability were nonuniformly distributed with loci at centromeric, telomeric and SD regions, which led to increased copy number counts when compared with previous genome assemblies (Supplementary Fig. VIII.29e–g).

## Repeat annotation

We generated a near-complete census of all high-copy repeats and their distribution across the ape genomes (Extended Data Table 1, Supplementary Note IX and Supplementary Table IX.41). With this new information, we estimated that the autosomes of the ape genomes contain 53.2–58.0% of detectable repeats, which include transposable elements, various classes of satellite DNA and variable number tandem repeats (VNTRs), among others (Supplementary Fig. IX.30). This autosomal content was significantly lower than the sex chromosomes (X, 61.8–66.3%; Y, 71.1–85.9%)[11]. Compared with previous genome assemblies, repeat content was increased from 286 to 706 Mb depending on the species (Supplementary Table IX.42). Gorilla, chimpanzee, bonobo and siamang genomes showed substantially higher satellite content, which was driven in large part by the accumulation of subterminal heterochromatin through lineage-specific satellite and VNTR expansions (Fig. 1, Extended Data Table 1 and Supplementary Fig. IX.30). Satellites accounted for the largest repeat variation (Extended Data Table 1), ranging from 4.9% satellite content in Bornean orangutans (159.2 Mb in total) to 13.0% in gorillas (462.5 Mb in total). An analysis of gaps in exon and repeat annotations led to the identification of 159 previously unknown satellite monomers (Supplementary Tables IX.41–IX.50), ranging from 0.5 to 7.1 Mb in additional base pairs classified per genome (Supplementary Fig. IX.30). Of note, 3.8 Mb of the sequence in the gorilla genome consisted of an approximately 36 bp repeat, herein named VNTR_148, which accounted for only 841.9 kb and 55.9 kb in bonobos and chimpanzees, respectively (Supplementary Table IX.43). This repeat displayed a pattern of expansion similar to that of the unrelated repeat pCht subterminal satellite[11], which suggests that it may have undergone expansion through a similar mechanism.

Multiple sequence alignments enabled us to define a comprehensive set of both truncated and full-length LINE, *Alu*, ERV and SVA retrotransposon insertions (Supplementary Tables IX.53–IX.55). Orangutans seemed to have the highest LINE-1 (L1) mobilization rate based on both the absolute number of insertions and the number of full-length elements with intact open-reading frames (ORFs). By contrast, the African apes (gorilla, chimpanzee, bonobo and human) showed a higher accumulation of *Alu* insertions (Fig. 2c). The number of L1 retrotransposons with intact ORFs varied by a factor of 5.8, with chimpanzees having the lowest (95) and orangutans having the highest, with at least 2.5 times more L1 retrotransposons with intact ORFs (more than 500 in both orangutan species compared with 203 in gorillas). Humans fell in between this range. The overall number and high percentage of full-length L1 elements with intact ORFs in orangutans implies recent high L1 activity. *Alu* activity was quiescent in orangutans, a result consistent with previous reports[34], and indicates a genome environment in which L1 retrotransposons out-compete *Alu* retrotransposons. When considering only full-length ERV elements with both target-site duplications as well as Gag (capsid) and Pol (reverse transcriptase and integrase) coding domains, a marked difference was observed. In detail, higher full-length, species-specific ERV content was observed in gorillas (57), followed by humans (12) and chimpanzees (4). PtERV and HERV-K accounted

for the ERV elements with both target-site duplications and protein domains, along with more degraded ERV elements in gorillas, humans, chimpanzees and bonobos (Fig. 2d and Supplementary Table IX.56).

## Immunoglobulin and T cell receptor loci

The complete ape genomes made it possible to investigate structurally complex regions of high biomedical relevance in more detail, especially with respect to human disease. Moreover, four of the primate genomes generated here (bonobo, gorilla and the two orangutan species) were derived from fibroblast instead of lymphoblastoid cell lines. The latter, in particular, has been the most common source of most previous ape genome assemblies, which has limited the characterization of loci affected by somatic rearrangements (for example, immunoglobulin genes)[35]. Thus, we specifically focused on nine regions associated with the immune response or antigen presentation that are subjected to complex mutational processes or selective forces.

Immunoglobulins and T cell receptors (TCRs) mediate interactions with both foreign and self-antigens and are encoded by large, expanded gene families that undergo rapid diversification both within and between species[36,37]. We conducted a comparative analyses of the immunoglobulin heavy chain (*IGH*), light chain kappa (*IGK*) and light chain lambda (*IGL*) as well as T cell receptor alpha (*TRA*), beta (*TRB*), gamma (*TRG*) and delta (*TRD*) loci in four ape species (Supplementary Note X) for which two complete intact haplotypes were constructed (Extended Data Fig. 2a and Supplementary Fig. X.32a). We identified an average of 60 (*IGHV*), 36 (*IGKV*), 33 (*IGLV*), 46 (*TRAV* and *TRDV*), 54 (*TRBV*) and 8 (*TRGV*) putatively functional immunoglobulin and TCR genes per parental haplotype per species across the seven loci (Extended Data Fig. 2a and Supplementary Data 1). As previously noted for human haplotypes, the ape immunoglobulin and TCR gene loci were characterized by large structural differences within and between species, accounting for as much as 33% of inter-haplotype length differences in immunoglobulin loci and up to 10% in the TCR loci. This result suggests that immunoglobulin loci underwent more rapid divergence than the TCR loci, which may be due to evolutionary and functional constraints placed on TCR loci by required interactions with MHC (Extended Data Fig. 2b,c). Moreover, immunoglobulin loci showed the most substantial structural variation, including gene-spanning expansions of tandem duplications across all *IGH* haplotypes, long-range rearrangements in both gorilla *IGK* haplotypes and a notable 1.4 Mb inversion distinguishing the two *IGL* haplotypes of bonobos (Extended Data Fig. 2a and Supplementary Fig. X.32a,b). These large-scale differences frequently corresponded to species-specific sets of genes (those that constitute phylogenetic clades of genes specific to bonobos, gorillas or both orangutan species; Extended Data Fig. 2a and Supplementary Fig. X.32c). We observed the greatest number of species-specific genes in *IGH* (Supplementary Fig. X.32d), for which we also noted a greater density of SDs longer than 10 kb relative to the other six loci (Extended Data Fig. 2d). This finding indicates that the genomic structure for immunoglobulins is probably a key driver of intra-species evolution.

## MHC loci

We also completely assembled and annotated 12 ape haplotypes corresponding to the 4–5 Mb MHC region (Supplementary Note XI). MHC loci encode diverse cell surface proteins crucial for antigen presentation and adaptive immunity[38], are highly polymorphic among mammals[39] and are strongly implicated in human disease[40]. Comparative sequence analyses confirmed marked sequence divergence and structural variation (an average of 328 kb deletions and 422 kb insertions in NHPs compared to humans), including duplications ranging from 99.3 kb in siamang to 701 kb in the Sumatran orangutan *H-2* (Supplementary Tables XI.57 and XI.58), as well as contractions and expansions associated with specific MHC genes (Extended Data Fig. 3a,b). Overall, MHC class I genes showed greater structural variation within and among the NHPs than MHC class II genes (Extended Data Fig. 3a,b), with threefold

greater average duplication sequences per haplotype (171 kb versus 62 kb). Particularly high divergence in this region was seen in siamangs, which lacked a functional *MHC-C* locus and instead carries an apparently functional *MHC-J*-like locus that is only found as a pseudogene in humans (Extended Data Fig. 3a and Supplementary Figs. XI.33–XI.40). Although MHC I gene content and organization were nearly identical in humans, bonobos and chimpanzees, other apes showed more variation, including additional genes such as *Gogo-OKO*, related but distinct from *Gogo-A*, and the orangutan-specific pseudogene *MHC-Ap*, which is related to *HLA-H*[39] (Extended Data Fig. 2a). We observed expansion of *MHC-A* and *MHC-B* genes in both orangutan species (Extended Data Fig. 3a and Supplementary Figs. XI.38 and XI.39), with *MHC-A* exhibiting copy number variation with one or two copies and *MHC-B* duplicated in both haplotypes of each species. Similarly, both orangutan species showed loss of *MHC-C* on one haplotype but retained it on the other (Extended Data Fig. 3a, Supplementary Table XI.57 and Supplementary Figs. XI.38 and XI.39). All NHPs had a nearly identical set of MHC II loci, with the exception of the *DRB* locus, which is known to exhibit copy number variation in humans[41], and here showed the same pattern among the apes (Extended Data Fig. 3b, Supplementary Table XI.58 and Supplementary Figs. XI.33–XI.40). We also observed two cases in which a MHC II locus was present as a functional gene on one haplotype and as a pseudogene on the other haplotype (for example, the *Gogo-DQA2* locus in gorillas and the *Poab-DPB1 locus* in Sumatran orangutans). Annotation of additional MHC haplotypes from previously published genomes of chimpanzees, bonobos, gorillas and Sumatran orangutans[42] revealed comparable structural variation among the MHC genes (Supplementary Tables XI.62–XI.65). Overall, this observed variation in MHC gene organization is consistent with long-term balancing selection[41].

Given the deep coalescence of the MHC locus[43], we performed a phylogenetic analysis with the complete ape sequences. We successfully constructed 1,906 gene trees encompassing 76% of the MHC region from the six ape species (Extended Data Fig. 3c). We identified 19 distinct topologies (Supplementary Note XI), with 3 representing 96% (1,830 out of 1,906) of the region and generally consistent with the species tree and predominant ILS patterns. The remaining 4% were discordant topologies that clustered within 200–500 kb regions (Supplementary Table XI.59) and corresponded to MHC I and II genes. We estimated coalescence times of these regions to range from 10 to 24 Ma between species and <1–14 Ma within species (Extended Data Fig. 3c and Supplementary Fig. XI.42). We next performed genome-wide tests of selection as described above. Selection signatures and nucleotide diversity in the MHC region were among the top 0.1% genome-wide. These signatures confirmed long-term balancing selection on MHC in multiple great ape lineages, including central and eastern chimpanzees, as well as at least two regions in MHC consistent with positive selection in bonobos and western chimpanzees[43].

## Structural variation and chromosomal rearrangements

We identified at the sequence level all 26 large-scale chromosomal rearrangements originally described decades ago[44] that karyotypically distinguish humans from other great apes (Supplementary Note XII, Supplementary Table XII.66 and Supplementary Fig. XII.43). However, during our breakpoint characterization of known inversions, we identified several events that seemed to be more complex and seemed to be the result of serial inversion events (Fig. 3 and Supplementary Fig. XII.43). As an example, we identified a 4.8 Mb fixed inverted transposition on chromosome 18 in gorillas (Fig. 3a–c) that was incorrectly classified as a simple inversion but more likely to be explained by three consecutive inversions specific to the gorilla lineage transposing this gene-rich segment to 12.5 Mb downstream (Fig. 3a–c). Similarly, the complex organization of orangutan chromosome 2 (human chromosome 3 (HSA3)) can be explained through a model of serial inversions that required three inversions and one centromere repositioning event (evolutionary neocentromere (ENC)) to create Bornean orangutan chromosome 2, and four

inversions and one ENC for Sumatran orangutan (Fig. 3d,e). In addition to known events, complete sequencing of SDs in the ape genomes revealed hundreds of previously undescribed inversions. Focusing on events larger than 10 kb, we curated 1,140 interspecific inversions, of which 522 are newly discovered[7,44–66] (Supplementary Table XII.67). For a fairer comparison with previous inversion calls, we also computed recall rates (<1 Mb) and found that 69.3% (79 out of 114) of the previously reported inversions were confirmed here. Among the 35 cases we missed, 24 corresponded to changes in inversion breakpoints, which were actually recalled and better resolved in this study, nine represented individual differences or previous error, and for the remaining two, our inversion call missed smaller nested inversions. An assessment of the genotypes of the inversion calls revealed that 632 were homozygous (in both assembled ape haplotypes), with the remainder present in only one out of the two ape haplotypes and were therefore probably polymorphisms. In total, 416 inversions had an annotated gene mapping to at least 1 of the breakpoints, with 724 apparently devoid of protein-coding genes (Supplementary Table XII.67). Of these inversions, 63.5% (724 out of 1,140) had annotated human SDs at one or both ends of the inversion, a result that represents a significant 4.1-fold enrichment (one-sided, simulation empirical $P < 0.001$). The strongest signal was for inverted SDs mapping to the breakpoints (6.2-fold; one-sided, simulation empirical $P < 0.001$), which suggested that non-allelic homologous recombination is driving many of these events. We also observed significant enrichment of previously unknown transcripts (Supplementary Table VIII.35) at the breakpoints of the inversions of African great apes (one-sided, simulation empirical $P < 0.036$). We parsimoniously assigned >64% of homozygous inversions to the ape phylogeny (Supplementary Fig. XII.45), with the remaining inversions predicted to be recurrent. The number of inversions assigned to each branch, based on maximum parsimony, correlated with the length of the branch (linear regression, $R^2 = 0.77$), with the greatest number assigned to the siamang lineage ($n = 44$). However, the human lineage showed fivefold less than the expected number of inversions based on the branch length, and the trend still held when using the Bornean orangutan as a reference instead of human.

## Regions of rapid divergence

Because rapidly evolving regions can help pinpoint genes under positive selection[42] or *cis*-regulatory elements undergoing functional changes[67], we applied a mutation-counting approach to identify ancestor quickly evolved regions (AQERs) that diverged during ape evolution[67] (Supplementary Note XIII). These AQERs are the most divergent regions of the genome and do not consider mutation rate differences (for example, tandem repeats or GC bias). Therefore, AQERs are essentially agnostic to the underlying force, or forces, of divergence. We identified 13,128 AQERs (Supplementary Tables XIII.68–XIII.74) across 4 primate lineages, including 3,268 on the human branch (that is, HAQERs). Our analysis more than doubles the number of HAQERs identified from previous gapped primate assemblies ($n = 1,581$; Fig. 4a and Supplementary Fig. XIII.46). Such elements were highly enriched in repetitive DNA (for example, 5-fold for SDs $P < 1 \times 10^{-30}$, 2-fold for simple repeats $P < 1 \times 10^{-30}$ one-sided binomial test). HAQERs also exhibited a significant enrichment for bivalent chromatin states (repressing and activating epigenetic marks) across diverse tissues, and the strongest enrichment was observed for the bivalent promoter state (3-fold enrichment, $P < 1 \times 10^{-30}$ one-sided binomial test; Fig. 4a and Supplementary Table XIII.68). This signal was present to a lesser degree on the chimpanzee, bonobo and gorilla lineages (1.3-fold, 1.5-fold and 1.4-fold, respectively; $P < 1 \times 10^{-19}$, $P < 1 \times 10^{-30}$ and $P < 1 \times 10^{-12}$, respectively, one-sided binomial test). This reduced enrichment may be due to the chromatin-state differences between human cell lines and tissues (Supplementary Note XIII). An example of a human-specific HAQER change included an exon and a potential *cis*-regulatory element in *ADCYAP1* (which encodes adenylate cyclase activating polypeptide 1), which is expressed in layer 5 extratelencephalic projecting neurons of the primary motor cortex.

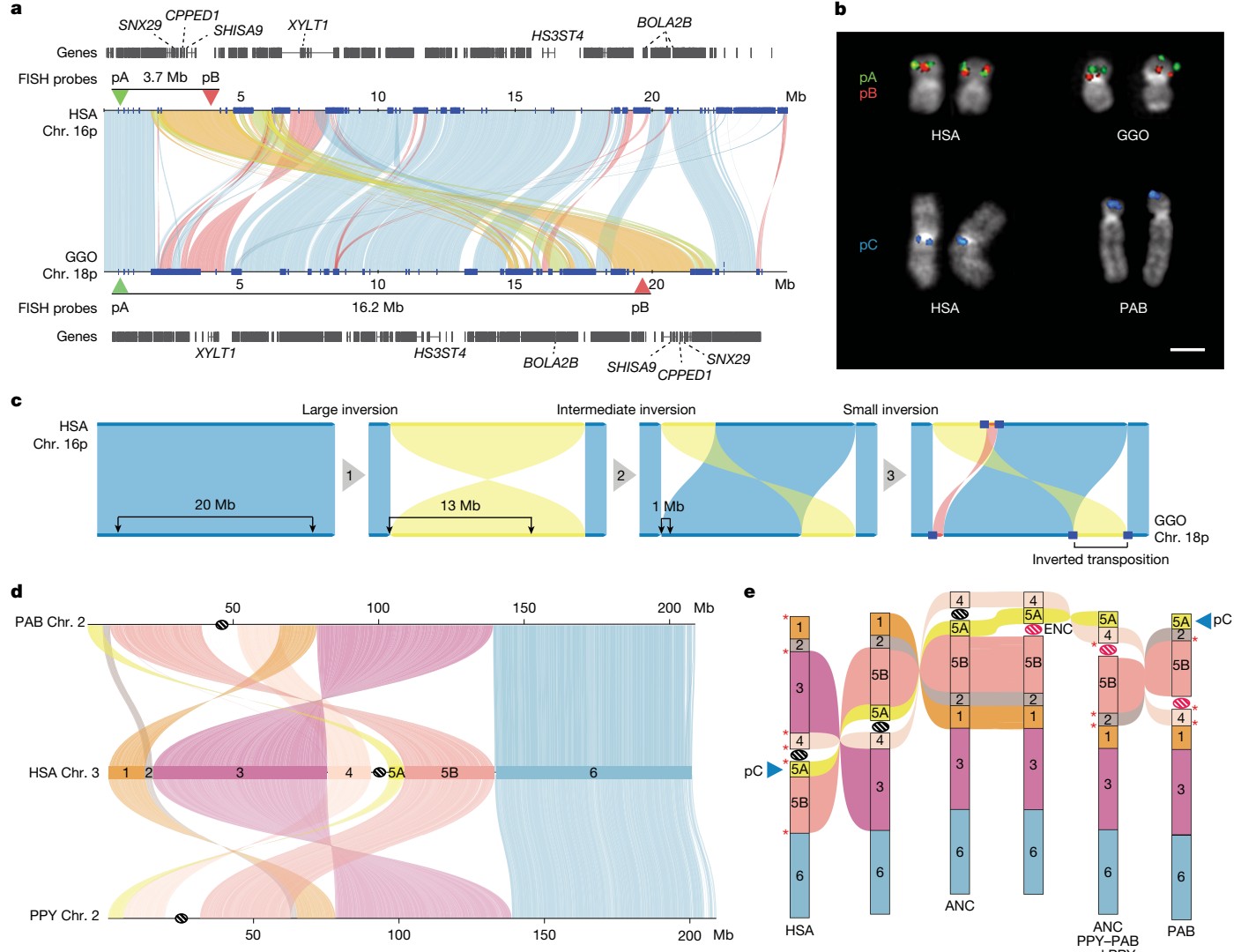

**Fig. 3 | Inversions and evolutionary rearrangements in great apes.**
**a**, Alignment plot of gorilla chromosome 18p and HSA16p shows a 4.8 Mb inverted transposition (yellow). SDs are shown with blue rectangles. **b**, Experimental validation of the gorilla chromosome 18 inverted transposition using FISH with probes pA (CH276-36H14) and pB (CH276-520C10), which overlap in human metaphase chromosomes. The transposition moves the red pB probe further away from the green pA probe in gorillas, resulting in two distinct signals. FISH on metaphase chromosomes using probe pC (RP11-481M14) confirmed the location of a new inversion to the terminal end of the short arm of PAB chromosome 2. Each FISH experiment was repeated three times, and ten metaphase spreads with the corresponding fluorochromes were captured for

each experiment. Scale bar, 1 μm. **c**, An evolutionary model for the generation of the inverted transposition through a series of inversions mediated by SDs. **d**, Alignment plot of orangutan chromosome 2 homologues to HSA3 highlights a more complex organization than previously known by cytogenetics[117]: a new inversion of block 5A is mapping at the terminal end of the short arm of chromosome 2 in both PAB and PPY. **e**, A model of serial inversions requires three inversions and one centromere repositioning event (ENC) to create PPY chromosome 2, and four inversions and one ENC for PAB. Red asterisks show the location of SDs mapping at the seven out of eight inversion breakpoints. ANC, ancestor.

This gene shows convergent downregulation in the speech-related motor cortex in humans and the analogous vocal-learning extratelencephalic projection neurons necessary for speech and song production in songbirds[68,69]. Here we found downregulation in layer 5 neurons of humans relative to macaques (single-cell RNA sequencing (scRNA-seq)) and an associated distinct human epigenetic signature (hypermethylation and decreased ATAC–seq) in the middle HAQER of the gene that was not observed in the same type of neurons of macaques, marmosets or mice (Fig. 4b and Supplementary Figs. XIII.47 and XIII.48).

We also applied a gene-based analysis (TOGA, a tool to infer orthologues from genome alignments) that focused on the loss or gain of orthologous sequences in the human lineage[70] (Supplementary Note XIV). TOGA identified 6 candidate genes from a set of 19,244 primate genes as largely restricted to humans (absent in >80% of the

other NHPs; Supplementary Table XIV.75). Among the candidates was a processed gene, *FOXO3B*, present in humans and gorillas and the paralogue of which, *FOXO3A*, has been implicated in human longevity[71]. Although *FOXO3B* is expressed, its study has been challenging to study because it is embedded in a large, highly identical SD that mediates Smith–Magenis deletion syndrome (Supplementary Fig. XIV.50). Although extensive functional studies will be required to characterize the hundreds of candidates we identified, we release an integrated genomic (Supplementary Table V.24) and genic (Supplementary Table V.25) callset of accelerated regions for future investigation.

### Genome browser and other annotation features
To facilitate downstream analyses with these genomes, we developed a UCSC Browser hub (https://github.com/marbl/T2T-Browser) that

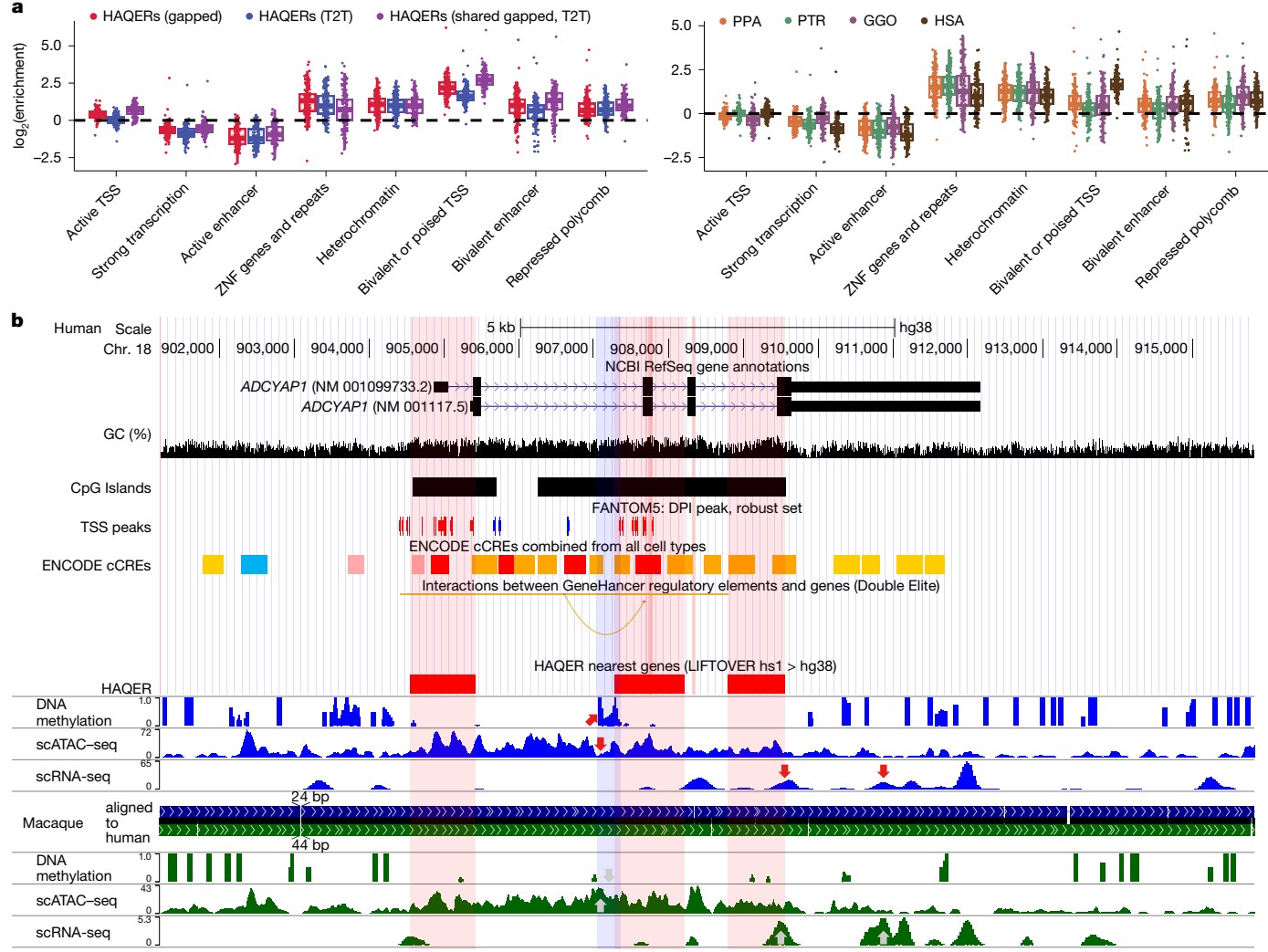

**Fig. 4 | AQERs. a**, Left, HAQER sets identified in gapped (GRCh38) and T2T assemblies show enrichments for bivalent gene regulatory elements across 127 cell types and tissues, with the strongest enrichment observed in the set of HAQERs shared between the two analyses. Right, the tendency for HAQERs to occur in bivalent regulatory elements (defined using human cells and tissues) is less strong in the sets of bonobo, chimpanzee and gorilla AQERs. $n = 127$ biologically independent samples for each chromatin state. Boxes show the interquartile range and median, with whiskers showing data points within 1.5 times the s.d. **b**, HAQERs in the vocal-learning-associated gene *ADCYAP1*, marked as containing alternative promoters near transcription start site (TSS) peaks of the FANTOM5 CAGE analysis, candidate *cis*-regulatory elements (cCREs) from ENCODE and enhancers (ATAC–seq peaks). For the latter, humans have a unique methylated region in layer 5 extratelencephalic projection neurons of the primary motor cortex. Tracks (blue for human, green for macaque) are modified from the UCSC Genome Browser[118] above the HAQER annotations and the comparative epigenome browser[119] below the HAQER annotations. For the NCBI RefSeq annotations, GCF_000001405.40-RS_2023_10 release (11 October 2024) was used. For CpG islands, islands <300 bp are in light green.

includes the underlying genome-wide alignments, gene annotations, accelerated regions and repeats, among other parameters (Data availability). We augmented this browser with additional tracks to be used in future studies and follow-up papers. For example, we mapped Repli-seq, including previously collected NHP datasets[72], to investigate evolutionary patterns of replication timing and identified 20 distinct states of replication timing that can now be correlated with different genomic features (Supplementary Figs. XVII.54 and XVII.55). Because they can interfere with mitochondrial DNA annotation, we created a curated set of integrated nuclear sequences of mitochondrial DNA origin (NUMTs) in ape genomes (Extended Data Table 1 and Supplementary Note IX). We observed a substantial gain in the number (3.7–10.5%) and total length of NUMTs (6.2–30%) (Supplementary Table IX.51) compared to non-T2T assemblies. We also identified and annotated a nearly complete catalogue of sequence motifs capable of forming non-canonical (non-B) DNA (G-quadruplexes, A-phased, direct, mirror, inverted and short tandem repeats in particular), especially in

previously inaccessible regions that have been historically difficult to sequence[73] (Supplementary Note XV and Supplementary Table II.12). We overlaid these data with a multiscale epigenomic map of the apes, including DNA methylation and replication timing (Supplementary Notes XVI–XVII). Using the 5-methylcytosine DNA signature from long-read sequencing data, for example, we distinguished hypomethylated and hypermethylated promoters associated with gene expression and demonstrated that in each cell type, the majority (about 88%) of promoters were consistently methylated (8,174 orthologous ape genes assessed; Supplementary Tables XVI.76 and XVI.77 and Supplementary Fig. XVI.52). The analysis identified on average about 12% of promoters that were variably methylated among species (1,035 for fibroblast cell lines and 922 lymphoblast cell lines) as candidates for regulatory differences among the species (Supplementary Tables XVI.76 and XVI.77). Consistently, methylated promoters were associated with more highly expressed genes and had a higher density of CpG sites compared with variably methylated promoters ($P < 10^{-16}$ two-sided Mann–Whitney

*U*-test; Supplementary Fig. XVI.52), a result that highlights the association between the evolution of CpG sequences, methylomes and transcriptomes. These annotations will facilitate future investigations to understand genome evolution and epigenetic changes associated with gene-expression changes in ape genomes[74,75].

## Newly characterized regions

In addition to these improved insights into genes, repeats and diversity, the contiguity afforded by the complete genomes enabled more systematic investigations of regions typically excluded from previous reference genomes and evolutionary analyses. Here we highlight four of the most notable regions: acrocentrics, centromeres, subterminal heterochromatic caps and lineage-specific SDs.

### Acrocentric chromosomes and nucleolar organizer regions

The human acrocentric chromosomes (13, 14, 15, 21 and 22) contain nucleolar organizer regions (NORs) that encode the rRNA components of the 60S and 40S subunits. The precise sequences of human NORs and the surrounding heterochromatin of the short arms were only recently elucidated in the first T2T human genome[9]. Human acrocentric chromosomes typically contain a single NOR with a head-to-tail rDNA array that is uniformly transcribed in the direction of the centromere. Each NOR is preceded by a distal junction (DJ) region that extends approximately 400 kb upstream of the rDNA array containing a 230 kb palindrome (Fig. 5a, Supplementary Note XVIII and Supplementary Fig. XVIII.58) that encodes a long ncRNA associated with nucleolar function[76]. The NOR is flanked by a variable patchwork of satellites and SDs. Heterologous recombination is thought to occur here, as well as within the rDNA array itself, to maintain NOR homology through the action of concerted evolution[77].

One conspicuous observation confirmed by our assemblies is that the ape NORs exist on different chromosomes for each species (Fig. 5b and Supplementary Fig. XVIII.59). For example, HSA15 is NOR-bearing (NOR⁺) in humans but not in chimpanzees or bonobos (NOR⁻), whereas HSA18 is NOR⁺ in both chimpanzees and bonobos, but NOR⁻ in humans[78] (Fig. 5b and Supplementary Fig. XVIII.59). Among great apes, we found that the total number of NORs per haploid genome varied from as few as two in gorilla to ten in both orangutan species, whereas the siamang maternal genome sequenced here harboured only a single NOR (Fig. 5b and Supplementary Fig. XVIII.59). NORs were also identified on both orangutan and siamang Y chromosomes[11], and partial DJ fragments on the chimpanzee and bonobo Y chromosomes (Fig. 5b and Supplementary Fig. XVIII.59), which suggests that their ancestral Y chromosome may have been NOR⁺. With the exception of rRNA genes, all ape NOR⁺ chromosome short-arms were typically satellite-rich and gene-poor (Fig. 5c,d), with the NORs restricted to the end of an autosomal short-arm or the end of a Y chromosome long-arm. However, we identified multiple acrocentric chromosomes with heterochromatic sequences on their short arm, but without a NOR (for example, gorilla HSA2A, HSA9, HSA13, HSA15 and HSA18). Unlike the NOR⁺ acrocentrics, these NOR⁻ acrocentrics carried multiple predicted protein-coding genes on their short arms. Thus, short-arm heterochromatin is strongly associated with ape NORs although not always predictive of their presence.

The estimated rDNA copy number per array varied from 1 on Bornean orangutan chromosome Y to 287 on chimpanzee HSA21, and the total diploid rDNA copy number similarly varied from 343 in siamangs to 1,142 in chimpanzees (Fig. 5b, Supplementary Note XVIII, Supplementary Table XVIII.78 and Supplementary Fig. XVIII.59). The total rDNA copy number varied widely between individual NOR⁺ haplotypes of the same species as expected, but within the range of human variation. For example, a recent analysis of 10 human genomes estimated that the total rDNA copy number ranged from 532 to 683, with individual chromosomes showing more than 10-fold variation (for example, 22–260 rDNA copies on chromosome 14)[79]. Heterozygous NOR loss

was observed in bonobo HSA21 and Sumatran orangutan HSA13, both of which were mediated by a truncation of the chromosome before the typical NOR location (Fig. 5b and Supplementary Fig. XVIII.59). The structure and composition of both satellites and SDs varied considerably among the apes, as well as between chromosomes and haplotypes within individuals (Fig. 5a,c and Supplementary Fig. XVIII.60). The orangutan acrocentrics were dominated by human satellite 3 (HSat3) and α-satellites compared with the more balanced satellite composition of the other apes. Gorillas were notable for the presence of double NORs on both haplotypes of HSA22, with the additional NORs inverted relative to the first and including a complete DJ but only a single, inactive rDNA unit (Supplementary Fig. XVIII.61).

At the chromosome level, the high level of synteny on the long arms of the NOR⁺ chromosomes rapidly degraded when transitioning to the short arm, with almost no sequence aligning uniquely between different ape species (Fig. 5d). Even the haplotypes of a single human genome aligned best to different reference chromosomes on their distal ends, a result that supports previous observations of extensive heterologous recombination[77]. Despite their widespread structural variation, the ape NOR⁺ chromosomes shared common features such as homogeneous rDNA arrays containing highly conserved rRNA genes. We extracted representative rDNA units from each assembly to serve as a reference for each species and confirmed a similar sequence structure, including the presence of a central microsatellite region in the intergenic spacer sequence for all species (Supplementary Fig. XVIII.62), but with relatively high nucleotide substitution rates outside the >99% identical 18S and 5.8S coding regions (Supplementary Fig. XVIII.63). Notably, certain species contained more variation in the rDNA units than others, which is explained by the fact that units in the same array tended to seem more similar than units in different arrays, and species have different numbers of NORs (Fig. 5b and Supplementary Fig. XVIII.64). This result is likely to be due to the increased prevalence of unequal crossing over of homogenizing units in an array compared to gene conversions or crossover of homogenizing units between arrays[80,81]. Despite its conserved colinear structure, the nucleotide identity of the intergenic spacer varied from 95.2% for humans compared to bonobos to just 80.6% for humans compared to siamangs (considering only single-nucleotide variants; Supplementary Table XVIII.79). The DJ sequence was conserved across all great apes and present as a single copy per NOR, including the palindromic structure typical of the human DJ, with the exception of siamangs, which contained only one-half of the palindrome on each haplotype but in opposite orientations on both haplotypes of chromosome 21 (Supplementary Fig. XVIII.65). The transcriptional direction of all rDNA arrays was consistent in each species, with the chimpanzee and bonobo arrays inverted relative to human arrays (Fig. 5a). This inversion included the entire DJ sequence, a result that confirms previous fluorescence in situ hybridization (FISH) analyses showing that the chimpanzee DJ had been relocated to the centromeric side of the rDNA array[76]. Our comparative analysis supports the idea that the DJ is a functional component of ape NORs that is consistently positioned upstream of rRNA gene transcription rather than distally on the chromosome arm.

### Centromere satellite evolution

The assembly of five NHP genomes enabled us to assess the sequence, structure and evolution of centromeric regions at base-pair resolution. Using these assemblies, we identified 227 contiguous centromeres out of a possible 230 centromeres across five NHPs, each of which were composed of tandemly repeating α-satellite DNA organized into higher-order repeats (HORs) belonging to one or more α-satellite suprachromosomal families (SFs) (Fig. 6, Supplementary Note XIX and Supplementary Fig. XIX.66). In specific primate lineages, different SFs have risen to high frequency, such as SF5 in orangutans and SF3 in gorillas. We carefully assessed the assembly of each of these centromeres, checking for collapses, false duplications and misjoins (Supplementary Note XIX,

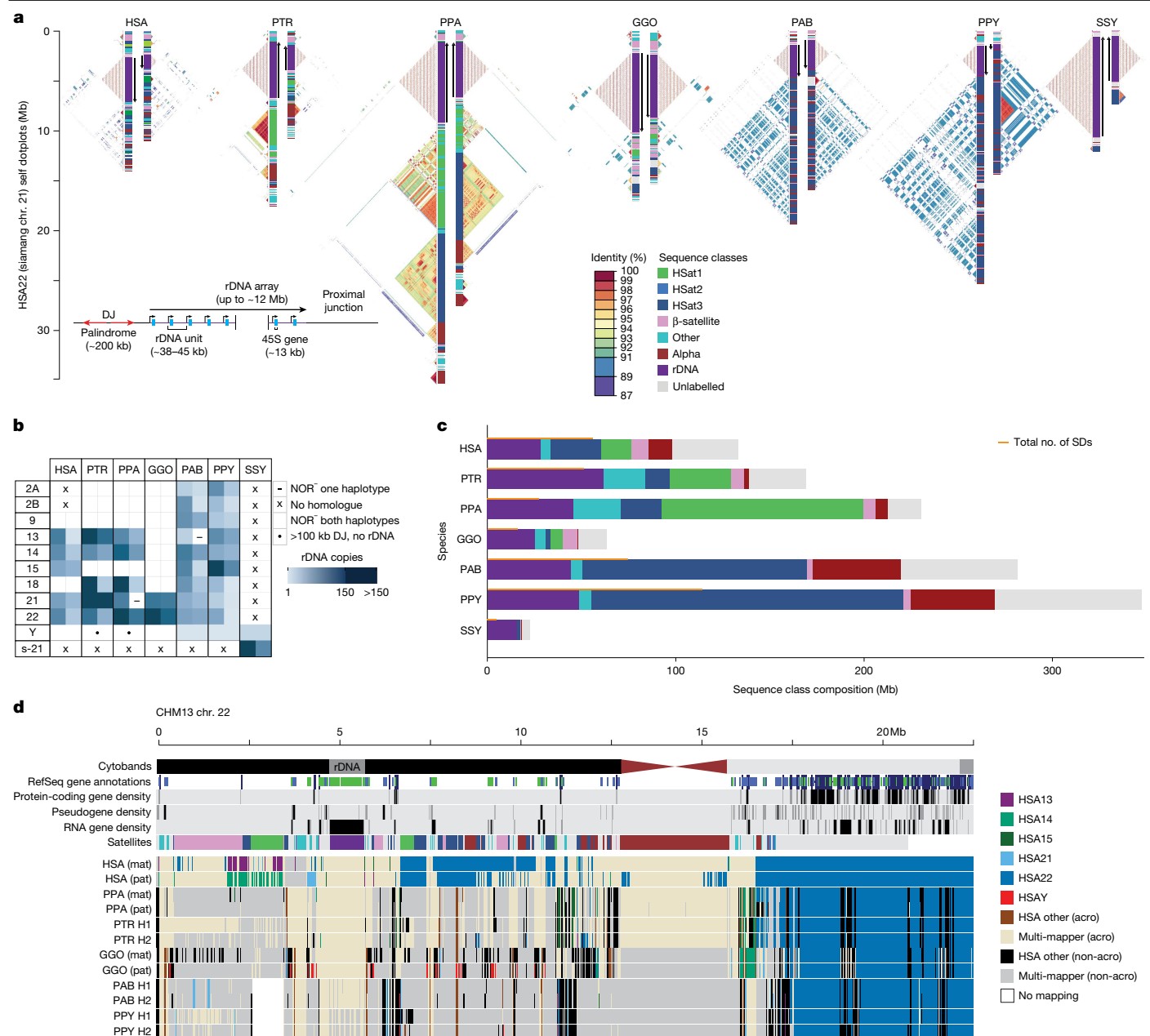

**Fig. 5 | Organization and sequence composition of the ape acrocentric chromosomes. a**, Sequence identity heatmaps and satellite annotations for the NOR⁺ short arms of both HSA22 haplotypes across all the great apes and siamang chromosome 21 (the only NOR⁺ chromosome in siamang) drawn with ModDotPlot[120]. The short-arm telomere is oriented at the top of the plot, with the entirety of the short arm drawn to scale up to but not including the centromeric α-satellite. Heatmap colours indicate self-similarity in the chromosome, and large blocks indicate tandem repeat arrays (rDNA and satellites) with their corresponding annotations given in between. Human is represented by the diploid HG002 genome. **b**, Estimated number of rDNA units per haplotype for each species. HSA chromosome numbers are given in the first column, with the exception of s-21 for siamang chromosome 21, which is NOR⁺ but has no single human homologue. **c**, Sum of satellite and rDNA sequences across all short arms for which one haplotype is NOR⁺ in each species.

'Unlabelled' indicates sequences without a satellite annotation, which mostly comprise SDs. The total number of SD bases is given for comparison, with some overlap between regions annotated as SDs and satellites. Colours for sequence classes are as for **a**. **d**, Top tracks, chromosome 22 in the T2T-CHM13v2.0 reference genome displaying various gene-annotation metrics and the satellite annotation. Bottom tracks, for each primate haplotype, including the human HG002 genome, the chromosome that best matches each 10 kb window of T2T-CHM13 chromosome 22 is colour coded, as determined by MashMap[121]. On the right side of the centromere (towards the long arm), HSA22 is syntenic across all species; however, on the short arm, synteny rapidly degrades, with very few regions mapping uniquely to a single chromosome, a result reflective of extensive recombination on the short arms. Even the human HG002 genome does not consistently align to T2T-CHM13 chromosome 22 in the most distal (left-most) regions. acro, acrocentric; mat, maternal; pat, paternal.

'Validation of centromeric regions'), and found that approximately 85% of bonobo, 69% of chimpanzee, 54% of gorilla, 63% of Bornean orangutan, but only 27% of Sumatran orangutan centromeres were complete and correctly assembled (Fig. 6a and Supplementary Fig. XIX.66). Most of the assembly errors were few (about two per centromere haplotype,

on average) and typically involved a few 100 kb of centromere satellite sequence that still needs further work to resolve.

Focusing on the completely assembled centromeres, we identified several characteristics specific to each primate species. First, bonobo centromeric α-satellite HOR arrays were on average 0.65-fold the length

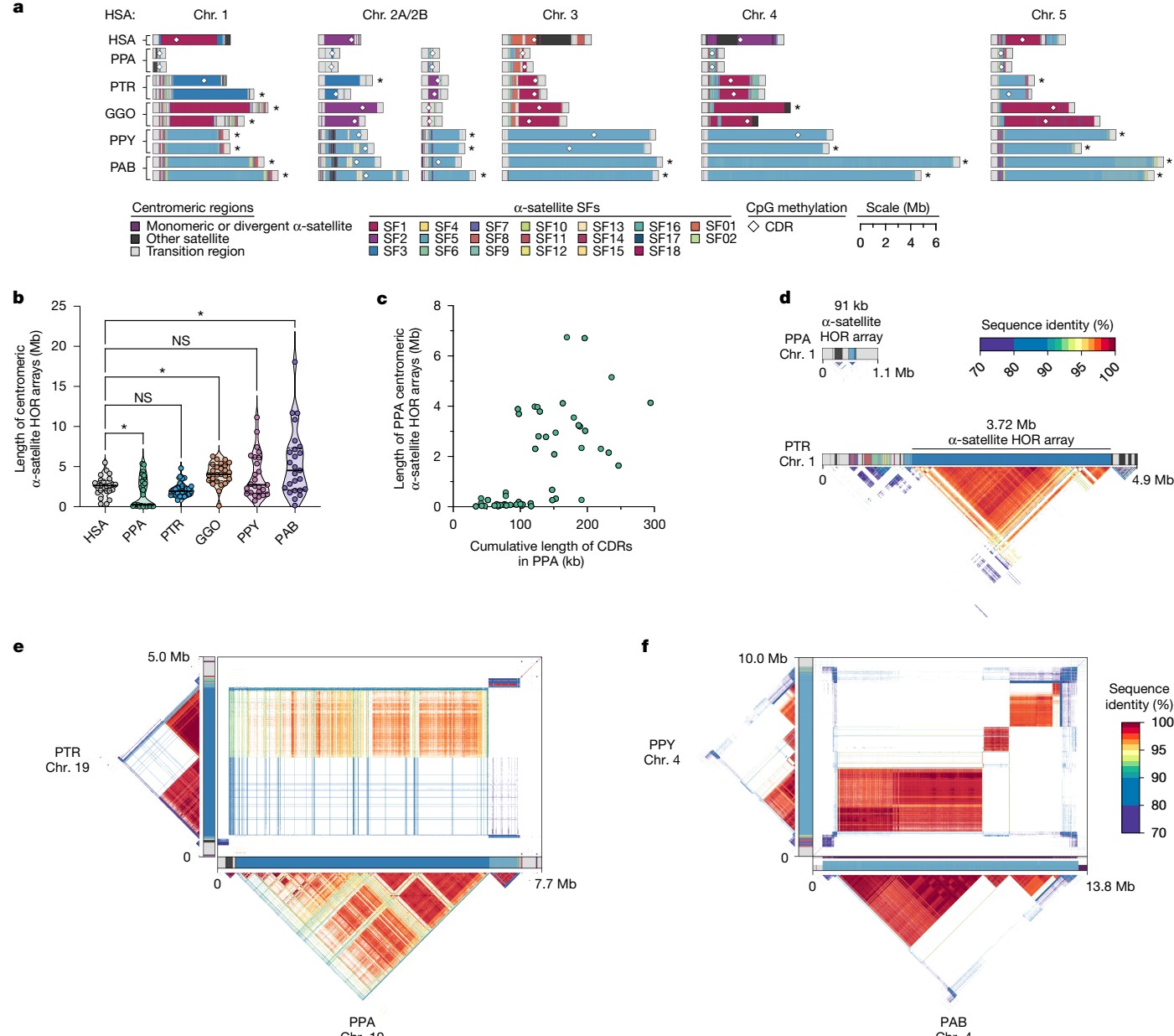

**Fig. 6 | Assembly of 237 NHP centromeres reveals variation in α-satellite HOR array size, structure and composition. a**, Sequence and structure of α-satellite HOR arrays from the human (T2T-CHM13), bonobo, chimpanzee, gorilla, Bornean orangutan and Sumatran orangutan chromosome 1–5 centromeres, with the α-satellite SF indicated for each centromere. The sequence and structure of all completely assembled centromeres is shown in Supplementary Fig. XIX.66. **b**, Variation in the length of the α-satellite HOR arrays for NHP centromeres. Bonobo centromeres have a bimodal length distribution, with 28 chromosomes showing minicentromeres (with α-satellite HOR arrays <700 kb long); two-tailed Mann–Whitney test, *P < 0.05; NS, not significant (compared to human, *P = 0.044, P = 0.103, *P = 0.0001, P = 0.287 and *P = 0.0099 for bonobo, chimpanzee, gorilla, Bornean and Sumatran

orangutans, respectively). **c**, Correlation between the length of the bonobo active α-satellite HOR array and the length of the CDR for the same chromosome. **d**, Example showing that bonobo and chimpanzee chromosome 1 centromeres are divergent in size despite being from orthologous chromosomes. **e**, Sequence identity heatmap between the chromosome 17 centromeres from bonobo and chimpanzee show a common origin of sequence as well as the birth of new α-satellite HORs in the chimpanzee lineage. **f**, Sequence identity heatmap between chromosome 5 centromeres from the Bornean and Sumatran orangutans show highly similar sequence and structure, except for one pocket of α-satellite HORs that is only present in the Bornean orangutan. For **d**–**f**, data are for haplotype 1.

of human centromeric α-satellite HOR arrays and 0.74-fold the length of its sister species the chimpanzee (Fig. 6b). A closer examination of bonobo α-satellite HOR array lengths revealed that they were bimodally distributed, with approximately half of the bonobo centromeres (27 out of 48) with an α-satellite HOR array with a mean length of 110 kb (range of 15–674 kb) and the rest (21 out of 48) with a mean length of 3.6 Mb (range of 1.6–6.7 Mb; Fig. 6c and Supplementary Fig. XIX.67). The bimodal distribution persisted in both sets of bonobo haplotypes.

This >450-fold variation in bonobo α-satellite HOR array length has not yet been observed in any other primate species and implies that there is a wide range of centromeric structures and sizes compatible with centromere function. Indeed, no 'minicentromere' arrays have been observed in chimpanzees, despite its recent speciation from bonobo (about 1.7 Ma; Fig. 6d).

As previously noted[82], we confirmed that chimpanzee α-satellite HOR arrays were consistently smaller: 0.86-fold the length of their human

counterparts (Fig. 6a). Moreover, chimpanzee centromeres were typically composed of a single α-satellite HOR array flanked by short stretches of divergent α-satellite HORs and monomeric sequences, which were interspersed with transposable elements before extending into the p arm and q arm (Fig. 6d). By contrast, gorilla α-satellite HOR arrays were on average 1.58-fold larger than human arrays (Fig. 6a,b), and unlike for bonobos and chimpanzees, they were composed of punctuated regions of α-satellite HORs or regions of α-satellite HORs that have high sequence identity in them but much lower sequence identity with neighbouring regions, flanked by larger transition zones to monomeric α-satellite sequences. Gorilla centromeres showed a high degree of haplotypic variation, with many paternal and maternal centromeres varying in size, sequence and structure. We found that 30.4% (7 out of 23) gorilla α-satellite HOR array pairs varied in size by >1.5-fold (especially HSA1, HSA2a, HSA4, HSA10, HSA15, HSA18 and HSA19), and 9 out of 23 pairs (around 39.1%) had α-satellite HOR arrays with >5% sequence divergence between homologues (HSA1, HSA4–HSA6, HSA10–HSA12, HSA15 and HSA19). Finally, the Bornean and Sumatran orangutan α-satellite HOR arrays were among the largest (1.52-fold and 2.11-fold larger on average, respectively, than humans; Fig. 6b) and were characterized by multiple pockets of divergent α-satellite HORs. A typical Bornean or Sumatran orangutan centromere had three or four distinct pockets of α-satellite HORs, with up to nine distinct HOR arrays observed in a single centromere (Bornean chromosome 19).

Congeneric species of *Pan* (chimpanzee and bonobo) and *Pongo* (orangutans) present an opportunity to assess the evolution of centromeric α-satellites over relatively shorter evolutionary time periods. Comparisons of the centromeric α-satellite HOR arrays from orthologous chromosomes across the bonobo and chimpanzee genomes revealed, for example, that 56% of them (14 out of 25 centromeres, including both X and Y) shared a common identifiable ancestral sequence, such as that present in HSA17 (Fig. 6e). On this chromosome, the entire bonobo α-satellite HOR array was 92–99% identical to one domain of α-satellite HORs present in the chimpanzee centromere. However, the chimpanzee centromere contained a second domain of α-satellite HORs that spanned approximately half of the α-satellite HOR array. This domain was <70% identical to bonobo α-satellite HORs, which indicated that the formation of a new α-satellite HOR array subregion was acquired in the chimpanzee lineage. Thus, in the short evolutionary time since bonobo and chimpanzee divergence (1–2 Ma), a new α-satellite HOR had arisen and expanded to become the predominant HOR that distinguishes these two closely related species (Fig. 6e). Given the shorter speciation time of orangutan (0.9 Ma), α-satellite HOR evolution was more tractable, with α-satellite HORs sharing >97% sequence identity, including domains with 1-to-1 correspondence. However, in about one-fifth of orangutan centromeres, we identified stretches of α-satellite HORs present in Bornean but not Sumatran orangutans (or vice versa). The emergence of lineage-specific α-satellite HOR sequences occurring on five chromosomes (HSA4, HSA5, HSA10, HSA11 and HSA16; Fig. 6f) marked by extremely high sequence identity (>99%) between α-satellite HOR arrays indicates the rapid turnover and homogenization of newly formed orangutan α-satellite HORs.

We leveraged the more complete NHP assemblies to assess the location and distribution of the putative centromere kinetochore. This is a large, proteinaceous structure that binds centromeric chromatin and mediates the segregation of chromosomes to daughter cells during mitosis and meiosis[83,84]. Previous studies of both humans[9] and NHPs[82,85] have shown that centromeres typically contain one kinetochore site, marked by one or more stretches of hypomethylated CpG dinucleotides termed the centromere dip region (CDR)[86]. We carefully assessed the CpG methylation status of all 237 primate centromeres and found that all contained at least one region of hypomethylation, consistent with a single kinetochore site. Focusing on the bonobo centromeres, we found a bimodal distribution in α-satellite HOR array length (Fig. 6b), in

which the CDR length and centromere length correlated (linear regression, $R^2$ = 0.41). In other words, the bonobo minicentromeres tended to associate with smaller CDRs compared with larger centromeres (Fig. 6c). Although more in-depth functional studies need to be performed, this finding indicates that the reduced α-satellite HOR arrays in bonobos are effectively limiting the distribution of the functional component of the centromere.

## Subterminal heterochromatin

In addition to centromeres, we completely sequenced and assembled the subterminal heterochromatic caps of siamangs, chimpanzees, bonobos and gorillas (Fig. 7a and Extended Data Fig. 1). In total, the subterminal satellites accounted for 1.05 Gb (270.0, 261.6 and 522.9 Mb of chimpanzees, bonobos and gorillas, respectively, or 4–7%) of the genome (Supplementary Note XX). In the case of siamangs, 642 Mb (10.1%) of the genome was made up of subterminal satellites (Extended Data Fig. 1). These large structures (up to 26 Mb in length) are thought to be composed almost entirely of tandem repetitive DNA: a 32 bp AT-rich satellite sequence, termed pCht7 in *Pan* and gorilla, or a 171 bp α-satellite repeat present in a subset of gibbon species[87–89]. Although their function is unknown, these chromosomal regions have been associated with late replication[90] and implicated in nonhomologous chromosome exchange and unique features of telomeric RNA metabolism[91,92]. Our analysis indicated that we successfully sequenced 79 gapless subterminal caps in gorillas (average length = 6.6 Mb) and 57 and 46 caps in chimpanzees and bonobos, including both haplotypes (average lengths of 4.8 and 5.2 Mb, respectively) with less than 3.8% of pCht arrays flagged as potentially misassembled (Fig. 7a). Siamangs possessed the largest (average length 6.7 Mb) and most abundant subterminal satellite blocks (96 out of 100 chromosomal ends across the two haplotypes). Compared with the data reported for the sex chromosomes for gorillas and siamangs (8.6 and 7.4 Mb, respectively), autosomal subterminal caps were more comparable in length (7.3 and 7.7 Mb, respectively). By contrast, for chimpanzees and bonobos, both the distribution and length of the heterochromatic caps significantly differed between autosomes and sex chromosomes. For example, we previously detected pCht subterminal arrays on the p arm of chimpanzees and the q arm of bonobo chromosome X, but in autosomes, the caps were much less prevalent in q arms. Moreover, in the *Pan* lineage, the average length of the subterminal repeat arrays were overall much longer on the autosomes for both chimpanzees and bonobos (5.3 and 6.6 Mb, respectively) compared with the sex chromosomes (0.66 and 2.2 Mb, respectively). Integration of replication-timing data confirmed that the subterminal caps in gorillas were late replicating, and there was a significant difference in terms of replication between gorilla chromosomes with and without subterminal heterochromatic caps (Supplementary Figs. XVII.56 and XVII.57).

In gorillas and the *Pan* lineage, the caps were organized into higher order structures, whereby pCht subterminal satellites formed tracts of average length of 335–536 kb interrupted by spacer SD sequences of a modal length of 32 kb (*Pan*) or 34 kb (gorilla; Supplementary Fig. XX.68). The spacer sequences were each specific to the *Pan* and gorilla lineages, but we confirmed that each began originally as a euchromatic sequence that became duplicated interstitially in the common ancestor of human and African apes. For example, the 34 kb spacer in gorillas mapped to a single copy sequence present in orangutans and HSA10, which began to be duplicated interstitially on chromosome 7 in chimpanzee but only in gorilla did it become associated with pCht satellites, expanding to over 477 haploid copies as part of the formation of the heterochromatic cap. Similarly, the ancestral sequence of the *Pan* lineage spacer mapped syntenically to orangutan and HSA9. The ancestral sequence duplicated to multiple regions in gorilla (q arms of chromosomes 4, 5, 8 and X, and p arms of chromosomes 2A and 2B), before being captured and hyperexpanded (>345 copies) to form the structure of subterminal satellites of chimpanzees and bonobos. Analyses of CpG methylation showed that each spacer demarcated a

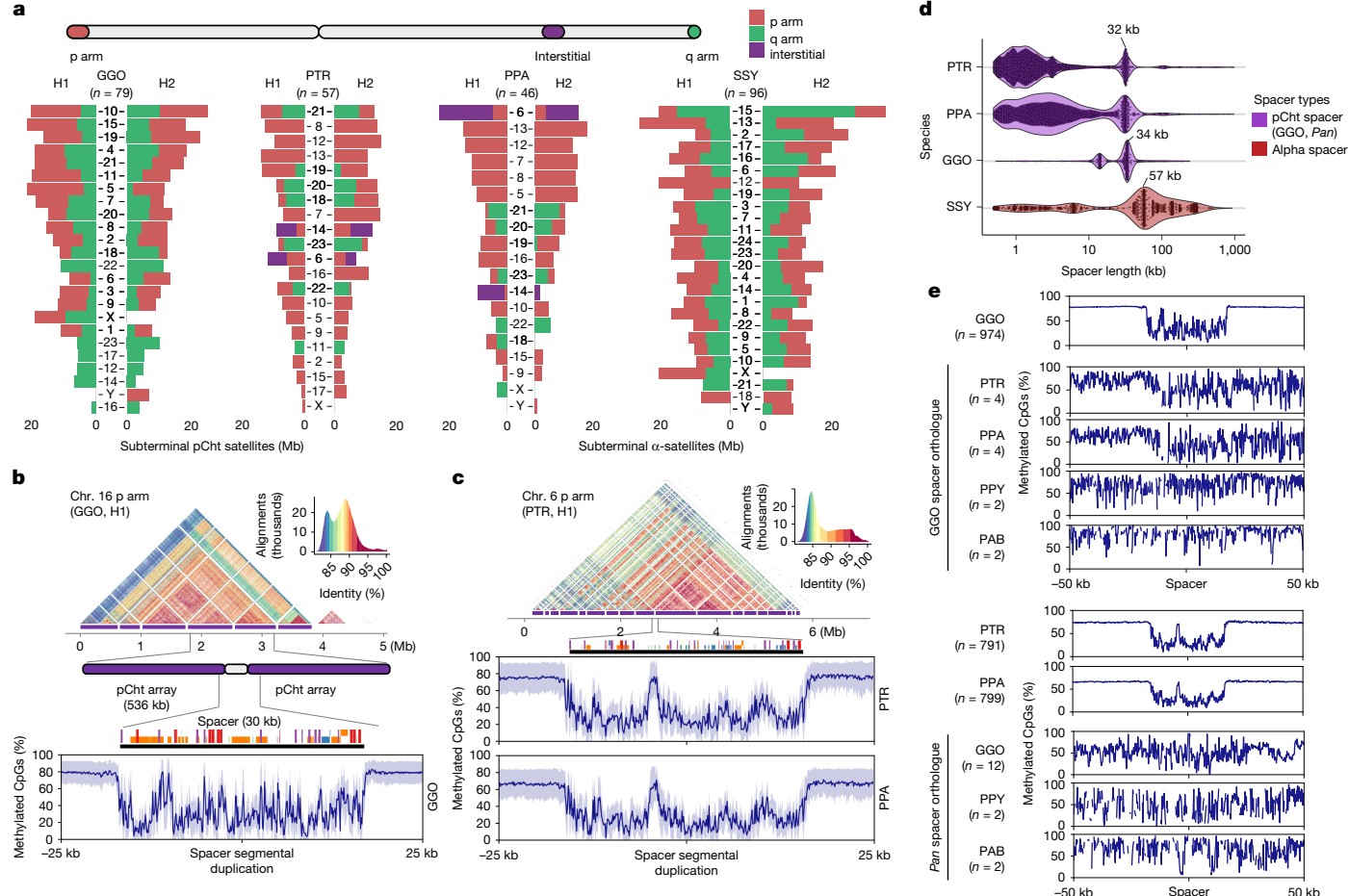

**Fig. 7 | Subterminal heterochromatin analyses. a**, Overall quantification of subterminal pCht and α-satellites in the African great ape and siamang genomes for haplotypes 1 and 2. The number of regions containing the satellite is indicated below the species acronym. The pCht arrays of diploid genomes are quantified by megabases, for ones located in the p arm, the q arm and the interstitial region. **b**,**c**, Organization of the subterminal satellite in gorillas (**b**) and the *Pan* lineages (**c**). The top shows a StainedGlass alignment plot indicating pairwise identity between 2-kb-binned sequences, followed by the higher order structure of subterminal satellite units, as well as the composition of the hyperexpanded spacer sequence and the methylation status across the 25 kb upstream or downstream areas of the spacer midpoint. The average per cent of CpG methylation is indicated as a blue line, and the band of lighter blue represents the s.d. of the methylation. **d**, Size distribution of spacer sequences identified between subterminal satellite arrays. **e**, Methylation profile of the subterminal spacer SD sequences compared to the interstitial orthologue copy.

pocket of hypomethylation flanked by hypermethylated pCht arrays within the cap (Fig. 7b–d). Of note, this characteristic hypomethylation pattern was not observed at the ancestral origin or interstitially duplicated locations (Fig. 7e), which suggested an epigenetic feature not determined solely by sequence but by its association with the subterminal heterochromatic caps. Similar to the great apes, we found evidence of a hypomethylated spacer sequence also present in the siamang subterminal cap; however, its modal length was much larger (57.2 kb in length) and its periodicity was less uniform, occurring every 750 kb (Supplementary Fig. XX.69). Nevertheless, the fact that these similar epigenetic features of the spacer evolved independently may suggest a functional role with respect to subterminal heterochromatic caps.

### Lineage-specific SDs and gene families

Compared with previous read-depth-based approaches that simply estimated copy numbers of SDs[93,94], T2T genomes increase the SD content and resolve sequence structures, which enabled us to distinguish SDs that are previously unknown by location and composition in each species (Supplementary Fig. XXI.70). Nonhuman great ape genomes generally contained more SDs (Fig. 8a) compared to humans (around 192 versus an average of 215 Mb in the other great apes); however, they were comparable when normalized by the genome size. We also

found that the assemblies of great apes on average had the highest SD content (208 Mb per haplotype) compared to non-ape lineages[42,95]: assemblies of mouse lemur, gelada, marmoset, owl monkey and macaque (68.8–161 Mb; Fig. 8a, Supplementary Note XXI, Supplementary Table XXI.80 and Supplementary Fig. XXI.72). In contrast to our previous analysis[96], orangutans showed the greatest number of SDs (225.3 Mb per haplotype) compared to African great apes (204.3 Mb per haplotype), which also exhibited larger interspersion of intrachromosomal SDs (Fig. 8b and Supplementary Fig. XXI.73). The increased SD content in orangutans is due to a greater number of acrocentric chromosomes (ten versus five on average for other apes) and a preponderance of clustered duplications. Consistent with the expansion of Asian great ape SDs, we found the largest number of lineage-specific SDs in the *Pongo* lineage (100.1 Mb), followed by gorilla-specific and human-specific SDs (69.5 and 41.9 Mb, respectively; Supplementary Table XXI.81 and Supplementary Fig. XXI.74). Many SDs (79.3–95.6 Mb per haplotype) in orangutans constituted massive, megabase-scale SD clusters, including a mixture of tandem and inverted duplications up to 21.5 Mb in size; in other species, the total number of such clustered duplications accounted for only 30–40 Mb per haplotype (except bonobos). In general, the number of SDs assigned to different lineages correlated with branch length (linear regression, $R^2 = 0.927$;

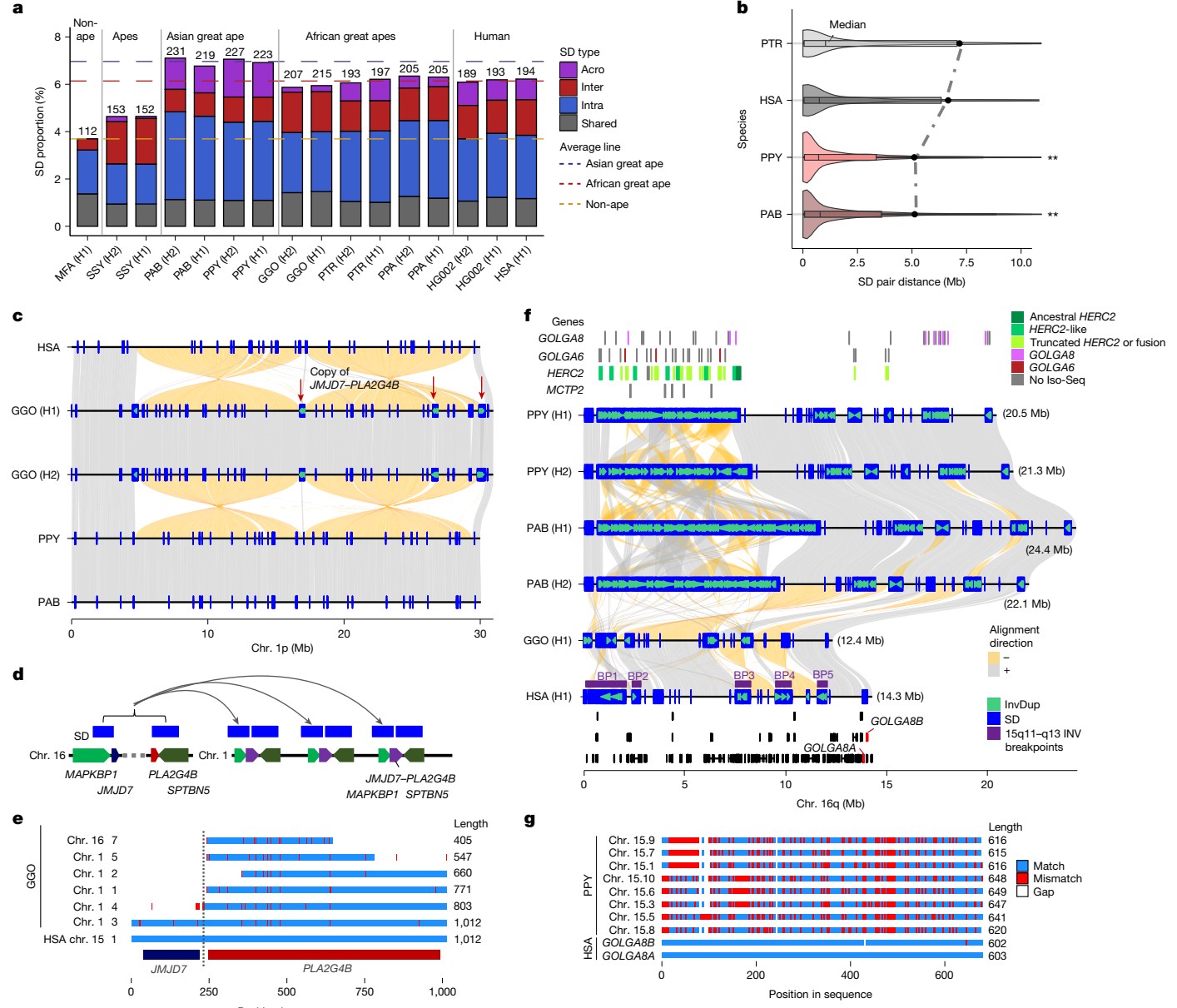

**Fig. 8 | Ape SDs and new genes. a**, Comparative analysis of primate SDs comparing the proportion of acrocentric (Acro), interchromosomal (Inter), intrachromosomal (Intra) and shared interchromosomal and intrachromosomal SDs (Shared). The total SD megabases per genome is indicated above each histogram, with the coloured dashed lines showing the average Asian, African great ape and non-ape SD (MFA, *Macaca fascicularis*[78]; see Supplementary Fig. XXI.71 for additional non-ape species comparison). **b**, A violin plot distribution of pairwise SD distance to the closest paralogue for which the median (black line) and mean (dashed line) are compared for different apes (see Supplementary Fig. XXI.72 for all species and haplotype comparisons; *n* = 17,703, 17,800, 19,979 and 21,066 of SD pairs for chimpanzees, humans and Bornean and Sumatran orangutans, respectively). The box indicates the interquartile range. An excess of interspersed duplications (one-sided Wilcoxon rank sum test; *P* < 2.2 × 10⁻¹⁶) was observed for chimpanzees and humans when compared to orangutans. **c**, Alignment view of chromosome 1 double-inversion for gorillas. Positive alignment direction is indicated in grey and negative as yellow. SDs and

those with inverted orientations are indicated by blue rectangles and green arrowheads. The locations where the *JMJD7–PLA2G4B* gene copies were found are indicated by the red arrows. **d**, Duplication unit containing three genes, including *JMJD7–PLA2G4B*. **e**, Multiple sequence alignment of the translated *JMJD7–PLA2G4B* predicated protein-coding genes. Each sequence is represented by chromosome number and copy number index. Match, mismatch and gaps are indicated with respect to their position in the linear amino acid sequence by blue, red and white, respectively. Regions corresponding to each of *JMJD7* or *PLA2G4B* are indicated by the track below. Data are for haplotype 1. **f**, Alignment view of chromosome 16q. The expansion of *GOLGA6*, *GOLGA8*, *HERC2* and *MCTP2* genes are presented in the top track. Recurrent inversions between species (yellow) are projected to the human genome with respect to genomic disorder breakpoints (BP1–BP5) at chromosome 15q. The track at the bottom indicates the gene track with *GOLGA8* human orthologue in red. InvDup, inverted duplications. **g**, Multiple sequence alignment of the translated *GOLGA8*.

Supplementary Fig. XXI.75), with the exception of siamangs and some ancestral nodes reflecting the great ape expansion of SDs[96]. We further assessed the variability of SDs within species by comparing two haplotypes and found that 30–37% were variable or polymorphic, with the exception of gorillas, which showed reduced variability (26.5%).

For Sumatran orangutans and siamangs, 42–50% were predicted to be variable based on the comparison of the two haplotypes (Supplementary Figs. XXI.76 and XXI.77). A much larger survey of individuals from each species, however, will need to be assessed to more accurately classify fixed versus polymorphic SDs in each species.

Leveraging the increased sensitivity afforded by FLNC Iso-Seq, we annotated the transcriptional content of lineage-specific SDs and identified hundreds of potential new genes, including gene family expansions often occurring in conjunction with chromosomal evolutionary rearrangements. We highlight here two examples in more detail. First, at two of the breakpoints of a 30 Mb double inversion of gorilla chromosome 1, we identified a gorilla-specific expansion of the genes *MAPKBP1* and *SPTBN5* as well as *PLA2G4B–JMJD7* (Fig. 8c–e) originating from an interchromosomal SD from ancestral loci mapping to HSA15 (duplicated in other chromosomes in chimpanzees and bonobos; Supplementary Table VIII.38). We estimated that these duplications occurred early after gorilla speciation 6.1 Ma (Supplementary Fig. XXI.77a), followed by subsequent expansion that resulted in the addition of eight copies (one truncated) mapping to two of the breakpoints of the double inversion. Investigation of the Iso-Seq transcript model of this gene revealed that five of the new gorilla copies are supported by multi-exon transcripts. Two of these additional copies possessed valid start and stop codons spanning at least 70% of the homologous single-copy orthologue gene in humans (Fig. 8e and Supplementary Fig. XXI.77a). Notably, the ancestral copy of this gene in gorillas (HSA15q) is highly truncated (40% of original protein), which suggested that the new chromosome 1 copies may have assumed and refined the function.

Second, in orangutan, we found a restructured 20 Mb region corresponding to the Prader–Willi syndrome (PWS)[63] region and the 15q13 microdeletion syndrome[97] region in humans (Supplementary Fig. XXI.78). This included a massive 6.8–10.8 Mb expansion of clustered tandem and inverted duplications mapping distally to breakpoint 1 of PWS as well as smaller 200–550 kb expansions of *GOLGA6* and *GOLGA8* repeats distal to PWS breakpoints 3 and 4 (Fig. 8f). We estimated that the larger region alone is composed of 87–111 copies of fragments of *GOLGA6*, *GOLGA8*, *HERC2* and *MCTP2*. We found Iso-Seq transcript support for 37–39 distinct orangutan copies. Using *GOLGA8* as a marker, we confirmed that it had expanded to 10–12 copies (>70% of original length) in orangutan but exists as a single copy in gorillas and bonobos and in two copies (*GOLGA8A* and *GOLGAB*) in humans out of multiple *GOLGA8* genes, retaining at least 70% of the sequence compared to orangutans (Fig. 8f,g and Supplementary Fig. XXI.77b). We estimated that the *Pongo* expansion of *GOLGA8* occurred 7.3 Ma (Supplementary Fig. XXI.77b), long before the species diverged. After alignment of the translated peptide sequence, we observed 17.1–23.7% divergence from the human copy (*GOLGA8A*; Fig. 8g). Based on studies of the African great ape genomes and humans, *GOLGA8* was among more than a dozen loci defined as 'core duplicons' that promoted the interspersion of SDs and genomic instability through palindromic repeat structures[31,98]. Our findings extend this recurrent genomic feature for the *GOLGA8* duplicons to the Asian ape genomes.

## Discussion

The complete sequencing of the ape genomes analysed in this study significantly refines previous analyses and provides a valuable resource for all future evolutionary comparisons. These include an improved and more nuanced understanding of species divergence, human-specific ancestral alleles, incomplete lineage sorting, gene annotation, repeat content, divergent regulatory DNA and complex genic regions as well as species-specific epigenetic differences involving methylation. These preliminary analyses revealed hundreds of new candidate genes and regions to account for phenotypic differences among the apes. For example, we observed an excess of HAQERS corresponding to bivalent promoters thought to contain gene-regulatory elements that exhibit precise spatiotemporal activity patterns in the context of development and environmental response[99]. Bivalent chromatin-state enrichments have not yet been observed in fast-evolving regions from other great apes, which may reflect limited cross-species transferability of epigenomic annotations from humans. The finding of a HAQER-enriched

gene, *ADCYAP1*, that is differentially regulated in speech circuits and methylated in the layer 5 projection neurons that make the more specialized direct projections to brainstem motor neurons in humans shows the promise of T2T genomes to identify hard to sequence regions important for complex traits. Perhaps most notably, we provide an evolutionary framework for understanding the about 10–15% of highly divergent, previously inaccessible regions of ape genomes. In this regard, we highlight a few noteworthy findings.

### Orangutans show the greatest amount of recent SDs

Comparative analyses suggest that expansion of SDs occurred in the common ancestor of the great ape lineage as opposed to the African great ape lineage as we originally proposed based on sequence read-depth analyses back to the human reference genome[94,96]. This discrepancy highlights the importance of ab initio genome assemblies of related lineages that are comparable in quality and contiguity. The assembly of the acrocentric chromosomes (of which orangutans have the maximum up to ten) and the resolution of massive (>10 Mb) tandem SDs in the orangutan species account for the increase in SD content among the Asian great apes. The African great ape lineage still stands out for having the largest fraction of interspersed SDs—a genomic architectural feature that promotes recurrent rearrangements that facilitate syndromic disease associated with autism and developmental delay in the human species[100]. A complete sequence resolution of NHP interspersed SDs provides a framework for understanding disease-causing copy number variants in these other NHP lineages[101].

### Large-scale differences in acrocentric chromosomes

The short arms of NOR⁺ ape acrocentric chromosomes seemed specialized to encode rRNA genes. On the autosomes, ape NORs existed exclusively on the acrocentric chromosomes, embedded in a gene-poor and satellite-rich short arm. On the Y chromosome, NORs occurred occasionally towards the end of the chromosome and adjacent to satellites shared with other acrocentric chromosomes. Previous analyses of the human pangenome suggested that heterologous recombination between chromosomes with NORs is a mechanism for concerted evolution of the rRNA genes[77,102]. Our comparative analysis of ape genomes provides further support for this hypothesis. For example, the uniform direction of all rDNA arrays in a species would permit crossover recombination between heterologous chromosomes without substantial karyotypic consequence. However, rare translocations, mediated by the large SDs that commonly surround the NORs, have occurred during ape evolution, which has resulted in a different complement of NOR⁺ acrocentric chromosomes and possibly creating reproductive barriers associated with speciation[103].

### Lineage-specific gene family expansions, explosions and rearrangements

The number of lineage-specific duplications that encode transcripts and potential genes is now estimated at hundreds per ape lineage, often occurring at sites of evolutionary chromosomal rearrangements that have been historically difficult to resolve through sequencing (Supplementary Table XXI.81). Our analysis uncovered hundreds of fixed inversions frequently associated with the formation of these lineage-specific duplications. These findings challenge the predominant paradigm that subtle changes in regulatory DNA[104] are the major mechanism underlying ape species differentiation. Rather, the expansion, contraction and restructuring of SDs lead to not only dosage differences but concurrent gene innovation and chromosomal structural changes[105]. Indeed, in the case of humans, four such gene family expansions, namely *NOTCH2NL*[32], *SRGAP2C*[29,30], *ARHGAP11* (ref. 31) and *TBC1D3* (refs.106,107), have been functionally implicated over the past decade in the expansion of the frontal cortex of the human brain[29,30,32,106] as well as human-specific chromosomal changes[31]. Detailed characterization of the various lineage-specific expansions in NHPs will be more challenging; however,

it is clear that such SDRs are an underappreciated genic source of interspecific difference and potential gene neofunctionalization.

## Bonobo minicentromeres

We identified several idiosyncratic features of centromere organization and structure that characterize the different ape lineages, significantly extending earlier observations based on the characterization of five select centromeres[82]. Perhaps the most notable is the bimodal distribution of centromere HOR length in the bonobo lineage, with 19 out of the 48 bonobo centromeres being less than 100 kb in size. Given the estimated divergence of the *Pan* lineage, such 300-fold reductions in size must have occurred recently, perhaps in the last million years. These bonobo minicentromeres seem to be fully functional, with a well-defined CDR (encompassing all of the α-satellite DNA). Thus, their discovery may provide a roadmap for the design of smaller, more streamlined artificial chromosomes for the delivery and stable transmission of new genetic information in human cells[108].

## Epigenetic architecture of subterminal heterochromatin

Our analyses suggested that the subterminal chromosomal caps of chimpanzees, gorillas and siamangs have evolved independently to create multi-megabases of heterochromatin in each species. In chimpanzees and gorillas, we defined a common organization of a subterminal spacer (about 30 kb in size) that is hypomethylated and flanked by hypermethylated heterochromatic satellites with a periodicity of one spacer every 335–536 kb of satellite sequence (pCht satellite in hominoids and α-satellite in hylobatids). In each case, the spacer sequence differed in its origin but has arisen as an ancestral SD[87] that has become integrated and expanded in the subterminal heterochromatin. In contrast to the ancestral sequences located in euchromatin, the spacer sequences embedded in the subterminal caps acquired more distinct hypomethylation signatures, which implicates an epigenetic feature. This subterminal hypomethylation pocket is reminiscent of the CDRs identified in centromeres that define the sites of kinetochore attachment[109] as well as a methylation dip region observed among some acrocentric chromosomes[110]. It is tempting to speculate that the subterminal hypomethylation pocket may represent a site of protein binding or a punctuation mark, perhaps facilitating ectopic exchange and concerted evolution[111]. Although the function of these regions is unknown, we confirmed previous observations that subterminal heterochromatic caps are very late replicating (Supplementary Figs. XVII.56–XVII.57). A previous study[111] of carboxy-terminal heterochromatic subterminal associations found evidence of ectopic exchanges and 'post-bouquet' structure formation in sperm during the pachytene stage of meiosis. Late replication may help promote such exchange events, buffering the proximal unique sequence while simultaneously promoting concerted evolution among nonhomologous chromosomes.

Although the ape genomes sampled here are nearly complete, some limitations remain. Sequence gaps still exist in the acrocentric centromeres and a few other remaining complex regions where the largest and most identical tandem repeats reside. This is especially the case for the Sumatran orangutan centromeres, for which only 27% were completely assembled. The length (nearly double the size of the other apes) and the complex compound organization of orangutan α-satellite HOR sequence will require specialized efforts to completely order and orient these large tandem arrays[82]. The combination of HiFi reads to generate the high-quality assembly backbone and UL-ONT sequencing for scaffolding across the largest repeats was crucial for closure of most repeats, and it may be that even longer UL-ONT data coupled with new advances in ONT error correction[112] would enable the longest tandem repeats in the Sumatran orangutan genome to be fully resolved. Nevertheless, with the exception of these and other large tandem repeat arrays, we estimate that about 99.5% of the content of each genome has been characterized and is correctly placed. Second, although we completed the genomes of a representative individual, we sequenced and assembled only two haplotypes from each of the 6 species, and more than 15 species or subspecies of apes remain[113]. Sampling more closely related species that diverged within the last million years will provide an opportunity to understand the evolutionary processes shaping the most dynamic regions of our genome. High-quality assemblies of all chimpanzee species[114], as well as the numerous gibbon species[115], will provide important insight into selection, effective population size and the rapid structural diversification of ape chromosomes at different time points. Finally, although high-quality genomes help eliminate reference bias, they do not eliminate annotation biases towards humans. This will be particularly crucial for understanding both the regulatory DNA that have rapidly diverged between the species and the function of the ape-specific genes that have emerged as a result of duplication.

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

[1]Department of Genome Sciences, University of Washington School of Medicine, Seattle, WA, USA. [2]Genome Informatics Section, Center for Genomics and Data Science Research, National Human Genome Research Institute, National Institutes of Health, Bethesda, MD, USA. [3]UC Santa Cruz Genomics Institute, University of California, Santa Cruz, Santa Cruz, CA, USA. [4]Department of Biosciences, Biotechnology and Environment, University of Bari, Bari, Italy. [5]Department of Genetics, Epigenetics Institute, Perelman School of Medicine, University of Pennsylvania, Philadelphia, PA, USA. [6]Computer Science and Engineering Department, Huck Institutes of Life Sciences, Pennsylvania State University, State College, PA, USA. [7]Institute of Genomics, University of Tartu, Tartu, Estonia. [8]Department of Molecular Genetics and Microbiology, Duke University Medical Center, Durham, NC, USA. [9]Research Unit for Evolutionary Immunogenomics, Department of Biology, University of Hamburg, Hamburg, Germany. [10]German Primate Center, Primate Genetics Laboratory, Goettingen, Germany. [11]Institute for Systems Genomics, University of Connecticut, Storrs, CT, USA. [12]Department of Biology, Penn State University, University Park, PA, USA. [13]Computer Science and Artificial Intelligence Laboratory, Massachusetts Institute of Technology, Cambridge, MA, USA. [14]The Broad Institute of MIT and Harvard, Cambridge, MA, USA. [15]Genetics Training Program, Harvard Medical School, Boston, MA, USA. [16]Stowers Institute for Medical Research, Kansas City, MO, USA. [17]The Jackson Laboratory for Genomic Medicine, Farmington, CT, USA. [18]Department of Genetics and Genome Sciences, University of Connecticut Health Center, Farmington, CT, USA. [19]Department of Biology, Johns Hopkins University, Baltimore, MD, USA. [20]NIH Intramural Sequencing Center, National Human Genome Research Institute, National Institutes of Health, Bethesda, MD, USA. [21]Department of Molecular and Cell Biology, University of Connecticut, Storrs, CT, USA. [22]Department of Medicine, KCVI, Oregon Health Sciences University, Portland, OR, USA. [23]Division of Genetics, Oregon National Primate Research Center, Beaverton, OR, USA. [24]PSU Medical School, Penn State University School of Medicine, Hershey, PA, USA. [25]The Translational Genomics Research Institute, City of Hope National Medical Center, Phoenix, AZ, USA. [26]Department of Biochemistry and Molecular Genetics, School of Medicine, University of Louisville, Louisville, KY, USA. [27]Department of Molecular Biology and Genetics, Cornell University, Ithaca, NY, USA. [28]Vertebrate Genome Laboratory, The Rockefeller University, New York, NY, USA. [29]San Diego Biomedical Research Institute, San Diego, CA, USA. [30]Department of Biomolecular Engineering, University of California, Santa Cruz, Santa Cruz, CA, USA. [31]Department of Genetics, Genomics and Informatics, University of Tennessee Health Science Center, Memphis, TN, USA. [32]Department of Computer Science and Engineering, University of California, San Diego, San Diego, CA, USA. [33]National Center for Biotechnology Information, National Library of Medicine, National Institutes of Health, Bethesda, MD, USA. [34]Bio-X Institutes, Key Laboratory for the Genetics of Developmental and Neuropsychiatric Disorders, Ministry of Education, Shanghai Jiao Tong University, Shanghai, China. [35]LOEWE Centre for Translational Biodiversity Genomics, Frankfurt, Germany. [36]Senckenberg Research Institute, Frankfurt, Germany. [37]Institute of Cell Biology and Neuroscience, Faculty of Biosciences, Goethe University Frankfurt, Frankfurt, Germany. [38]San Diego Zoo Wildlife Alliance, Escondido, CA, USA. [39]Department of Computer Science, Rice University, Houston, TX, USA. [40]Laboratory of Neurogenetics of Language, The Rockefeller University, New York, NY, USA. [41]Laboratory of Bioinformatics and Population Genetics, Interdisciplinary Program in Bioinformatics, Seoul National University, Seoul, Republic of Korea. [42]Bioengineering Program, Faculty of Engineering, Bar-Ilan University, Ramat Gan, Israel. [43]Department of Computer Science, Johns Hopkins University, Baltimore, MD, USA. [44]Department of Genetics and Biochemistry, Clemson University, Clemson, SC, USA. [45]Center for Human Genetics, Clemson University, Greenwood, SC, USA. [46]Neuroscience Research Institute, University of California, Santa Barbara, Santa Barbara, CA, USA. [47]Ray and Stephanie Lane Computational Biology Department, School of Computer Science, Carnegie Mellon University, Pittsburgh, PA, USA. [48]Center for Genomic Research, International Institutes of Medicine, Fourth Affiliated Hospital, Zhejiang University, Yiwu, China. [49]Shanghai Jiao Tong University Chongqing Research Institute, Chongqing, China. [50]Department of Computational Biology, Cornell University, Ithaca, NY, USA. [51]Faculty of Informatics and Data Science, University of Regensburg, Regensburg, Germany. [52]Department of Integrative Biology, University of California, Berkeley, Berkeley, CA, USA. [53]Department of Ecology, Evolution and Marine Biology, Neuroscience Research Institute, University of California, Santa Barbara, Santa Barbara, CA, USA. [54]School of Life Sciences, Westlake University, Hangzhou, China. [55]National Laboratory of Crop Genetic Improvement, Huazhong Agricultural University, Wuhan, China. [56]Howard Hughes Medical Institute, Chevy Chase, MD, USA. [57]Department of Primate Behavior and Evolution, Max Planck Institute for Evolutionary Anthropology, Leipzig, Germany. [58]Department of Molecular, Cellular and Developmental Biology, Neuroscience Research Institute, University of California, Santa Barbara, Santa Barbara, CA, USA. [59]Center for Computational Biology, University of California, Berkeley, Berkeley, CA, USA. [60]Department of Computer Science, University of Maryland, College Park, MD, USA. [✉]e-mail: kdm16@psu.edu; adam.phillippy@nih.gov; ee3@uw.edu

## Reporting summary

Further information on research design is available in the Nature Portfolio Reporting Summary linked to this article.

## Data availability

The raw genome sequencing data generated during this study are available under NCBI BioProject identifiers PRJNA602326, PRJNA976699–PRJNA976702 and PRJNA986878–PRJNA986879, and transcriptome data are deposited under BioProject identifiers PRJNA902025 (UW Iso-Seq) and PRJNA1016395 (UW and PSU Iso-Seq and short-read RNA-seq). The genome assemblies are available from GenBank under the following accessions: GCA_028858775.2, GCA_028878055.2, GCA_028885625.2, GCA_028885655.2, GCA_029281585.2 and GCA_029289425.2. Genome assemblies can be downloaded from the NCBI (https://www.ncbi.nlm.nih.gov/datasets/genome/?accession=GCF_028858775.2,GCF_029281585.2,GCF_028885625.2,GCF_028878055.2,GCF_028885655.2,GCF_029289425.2). Convenience links to the assemblies and raw data are available on GitHub (https://github.com/marbl/Primates) along with a UCSC Browser hub (https://github.com/marbl/T2T-Browser). The UCSC Browser hub includes genome-wide alignments, CAT annotations, methylation and various other annotation and analysis tracks used in this study. The T2T-CHM13v2.0 and HG002v1.0 assemblies used here are also available through the same browser hub, and from GenBank through accessions GCA_009914755.4 (T2T-CHM13), GCA_018852605.1 (HG002 paternal) and GCA_018852615.1 (HG002 maternal). The alignments are publicly available to download or browse in HAL118 MAF and UCSC Chains formats (https://cglgenomics.ucsc.edu/february-2024-t2t-apes). Additional public data include 22 genome assemblies summarized in Supplementary Table II.13, as well as the human reference genome, GRCh38. Previous genome annotation files include gorilla (GCF_029281585.2), bonobo (GCF_029289425.2), chimpanzee (GCF_028858775.2), Sumatran orangutan (GCF_028885655.2) and Bornean orangutan (GCF_028885625.2) from the NCBI. For the pangenome graph, public human genome assemblies (https://github.com/human-pangenomics/HPP_Year1_Data_Freeze_v1.0) were used. The short-read genome sequencing data for great ape species were obtained from SRP018689, ERP001725, ERP016782 and ERP014340. The public methylation data used in this study include the raw PacBio HiFi reads with kinetics and methylation tags for humans (https://humanpangenome.org/data/) and for chimpanzees, gorillas, Bornean orangutans, Sumatran orangutans and siamang gibbon (https://www.genomeark.org/t2t-all/). Original data underlying Supplementary Fig. XVIII.59 and Supplementary Table XVIII.78 are accessible from the Stowers original data repository (https://www.stowers.org/research/publications/libpb-2504).

## Code availability

All code used for the reported analyses is available from our project's GitHub repository (https://github.com/marbl/Primates). The following additional GitHub repositories are available: polishing of the genome (https://github.com/arangrhie/T2T-Polish); assembly quality control (https://github.com/EichlerLab/assembly_eval); implicit graph and pangenome (https://github.com/T2T-apes/ape_pangenome); Cactus alignment (https://cglgenomics.ucsc.edu/february-2024-t2t-apes); assessment of ancestral sequence (https://github.com/shanksc/ancestral_state); AQER analysis (https://github.com/vertgenlab/vglDocumentation/tree/master/primateT2T); population genome processing and selection analysis (https://github.com/aabiddanda/haplotype-phasing); acrocentric and rDNA analyses (https://github.com/jouyun/2024_Primate_rDNA, https://github.com/borcherm/primate_rdna_cn); species-specific MEI analysis (https://github.com/Markloftus/t2t-ape-MEIs); non-B DNA annotation and NUMT detection (https://github.com/makovalab-psu/T2T_primate_autosomes);

and transcript comparison (https://github.com/canzarlab/apes_transcriptome_analysis).

**Acknowledgements** We thank R. Buggs for suggestions and T. Brown, A. Antunes and M. Emam for editing the manuscript and supplementary note; V. Shivakumar for performing additional quality control on the ape genomes; I. A. Alexandrov and F. Ryabov for contributions to the centromere annotation track of the genome browser; members of the Genome in a Bottle Consortium for sharing preliminary RNA-seq data for HG002; and staff at the Frozen Zoo at San Diego Zoological Society for providing the fibroblast cell lines. This research was supported in part by the Intramural Research Program of the National Human Genome Research Institute, National Institutes of Health (NIH), and extramural NIH grants R35GM151945 (to K.D.M.), R01HG002385, R01HG010169, U24HG007497 (to E.E.E.), R35GM146926 (to Z.A.S.), R35GM146886 (to C.D.H.), R35GM142916 (to P.H.S.), R01HG012416 (to P.H.S. and E.G.), R35HG011332 (to C.B.L.), R01GM123312 (to R.J.O.), U24HG010263 (to M.C.S.), R35GM133747 (to R.C.M.), U41HG007234 (to P.H., M.D., B.P.), R01HG010329 (to P.H., M.D.), R01MH120295 (to M.D.), 1P20GM139769 (to M.K.K. and M.L.), R35GM133600 (to C.R.B. and P.B.), and 1U19AG056169-01A1, UH3AG064706 and U19AG023122 (to N.J.S.) as well as a Vallee Scholars Award to P.H.S. and Weill Neurohub Family Foundation support to E.E.E. We also acknowledge financial support by the Deutsche Forschungsgemeinschaft (DFG, German Research Foundation; 437857095, 444810852 to T.L.L.), a Verne M. Willaman Endowment Professorship (to K.D.M.) and a John and Donna Krenicki Endowment Professorship (to R.J.O.). Sequencing was partially supported by the NIH Intramural Sequencing Center. This work used the computational resources of the NIH HPC Biowulf cluster (https://hpc.nih.gov), the HPC in the Computational Biology Core in the Institute for Systems Genomics at UConn, Clemson University's HPC Palmetto Cluster, and the HPC in the Genomics Institute at UCSC. RNA-seq was performed at the PSU Genomics Core facility, and Hi-C sequencing was performed at the Genome Sciences core facility at the Penn State College of Medicine. This work was supported by the National Library of Medicine Training Program in Biomedical Informatics and Data Science (T15LM007093 to B.K.) and in part by the National Institute of Allergy and Infectious Diseases (P01-AI152999 to B.K.). We acknowledge financial support under the National Recovery and Resilience Plan (NRRP), Mission 4, Component 2, Investment 1.1, Call for tender No. 104 published on 2 February 2022 by the Italian Ministry of University and Research (MUR), funded by the European Union—NextGenerationEU—Project Title 'Telomere-to-telomere sequencing: the new era of Centromere and neocentromere eVolution (CenVolution)', CUP H53D23003260006, grant assignment decree no. 1015 adopted on 7 July 2023 by the Italian MUR; Project Title 'SUDWAY: Substance Use Disorders through Whole genome, psychological and neuro-endophenotypes AnalYsis', CUP H53D2300331, grant assignment decree no. 1015 adopted on 7 July 2023 by the Italian MUR. F.M. was supported by Fondazione con il Sud (2018-PDR-01136). This work was supported by the Italian MUR grant PRIN 2020 (project code 2020J84FAM, CUP H93C20000040001) to F.A. The work of M.H. was supported by the LOEWE-Centre for Translational Biodiversity Genomics (TBG) funded by the Hessen State Ministry of Higher Education, Research and the Arts (LOEWE/1/10/519/03/03.001(0014)/52). E.E.E. and E.D.J. are investigators of the Howard Hughes Medical Institute. The work of D.H., P.M. and F.T.-N. was supported by the National Center for Biotechnology Information of the NIH National Library of Medicine (NLM). This article is subject to HHMI's Open Access to Publications policy. HHMI laboratory heads have previously granted a nonexclusive CC BY 4.0 licence to the public and a sublicensable licence to HHMI in their research articles.

**Author contributions** Individual analysis leads are indicated with an asterisk. L. Carbone, L. Carrel, O.A.R., C. Steiner, M.L.H., B.M. and K.D.M. managed sampling. B.M. and A.P.L. performed transcriptome data generation. K.H., G.G.B., S.Y.B. and J.C. generated ONT long-read data. J.C., R.E.G. and S. Sacco provided Illumina sequencing data. G.H.G., K.M.M., P.H.S. and J.L.R. generated HiFi sequencing data. R.E.G. and S. Sacco made Hi-C libraries that were later sequenced by B.M. *B.D.P. and A.R. managed data, processed data submissions and coordinated administrative tasks. W.T.H., J.W., A.R. and B.D.P. performed assembly quality control. *D.A. and S.K. assembled the genomes. *A.R. performed polishing and created the genome browsers. Assembly generation was supervised by S.K. S.M. performed chromosome recognition and M.V. led definitions of chromosome nomenclature. L.S., K.K. and K.D.M. analysed non-B DNA. B.K., W.W., A.G., E.M., E.G., G.F. and P.H.S. created pangenome graph alignments. *G.H. generated the Cactus alignments. P.H.S., R.S.H., S.K.M., B.K., W.W., A.G., E.M., E.G., G.F. and K.D.M. performed divergence analyses. *J.M.S., P.B., C.R.B., C.F., P.Z., G.A.H. and R.J.O. analysed repeat content. E.T.-G. and K.D.M. investigated NUMTs. M.L. and M.K.K. investigated specifically for species-specific MEIs. P.B. and C.R.B. performed ORF analysis on species-specific FL-L1s. R.J.O. supervised and integrated the results. *A.N.S., A. Biddanda, Q.L., M.C.S., M.G.T., Z.A.S., C.D.H., R.C.M. and K.D.M. analysed population data and investigated selective sweeps. *Y.S., A. Bankevich, E.E., I.G., W.L., M.P., P.A.P., S. Saha, Z.Z., Y. Zhu and C.T.W. analysed immunoglobulin loci. Y.S. and C.T.W. led the analyses and integrated the results. J. Malukiewicz., B.S.M. and T.L.L. performed annotation of MHC genes. P.H. supported and validated the MHC annotations. M.T. performed phylogenetic tree analysis across MHC loci. Y.H.E.L., D.R.S. and S.V.Y. analysed epigenetic data, focusing on methylation and gene expression. *J. Ma, M.Y., Y. Zhang, G.H. and T.S. analysed replication timing. R.J.O., G.H., E.B., D.G. and T.S. generated Repli-seq data. *F.A., M.V., L.d.G. and D.Y. analysed inversions and large-scale chromosome rearrangements. F.A., D.Y. and D.P. visualized the data. *J.L., J.H., S.Z. and Y.M. performed SDR analyses. A.P.C., M.H. and N.J.S. performed TOGA analyses. A.P.C. integrated the results. *Y. Luo, *R.J.M., M.K., S.A.Z. and C.B.L. analysed divergent regions of the genome by predicting AQERs. C.B.L. supervised the analysis and summarized the results. C.L., Y. Lee and E.D.J. investigated further into candidate genes using AQERs. *S.J.S., A.P.S., J.L.G., T.P., G.M.A. and M.B. analysed acrocentric regions. T.P. and G.M.A. performed NOR chromosome imaging and quantification. S.J.S. and M.B. analysed rDNA. A.P.S. generated dot plots. J.L.G., M.V. and A.M.P. supervised the analyses. *G.A.L., K.H.M. and H.L. investigated centromeres. G.A.L. integrated the section. *D.Y. and E.E.E. investigated subterminal heterochromatin. D.Y. performed the analyses and E.E.E. supervised

the analyses. *D.Y. and E.E.E. analysed SDs. E.E.E. supervised the SD analyses. H.J. optimized the pipeline and D.P. and D.Y. visualized the data. P.H. analysed novel genes and curated gene annotation across the region. E.E.E., D.Y. and A.M.P. wrote and edited the manuscript with input from all authors. E.E.E., A.M.P. and K.D.M. initiated and supervised the project, acquired the funding along with other senior authors. E.E.E. and A.M.P. coordinated the study.

**Competing interests** E.E.E. is a scientific advisory board member of Variant Bio. C.T.W. is a co-founder and Chief Scientific Officer of Clareo Biosciences. W.L. is a co-founder and Chief Technology Officer of Clareo Biosciences. The other authors declare no competing interests.

**Additional information**
**Correspondence and requests for materials** should be addressed to Kateryna D. Makova, Adam M. Phillippy or Evan E. Eichler.

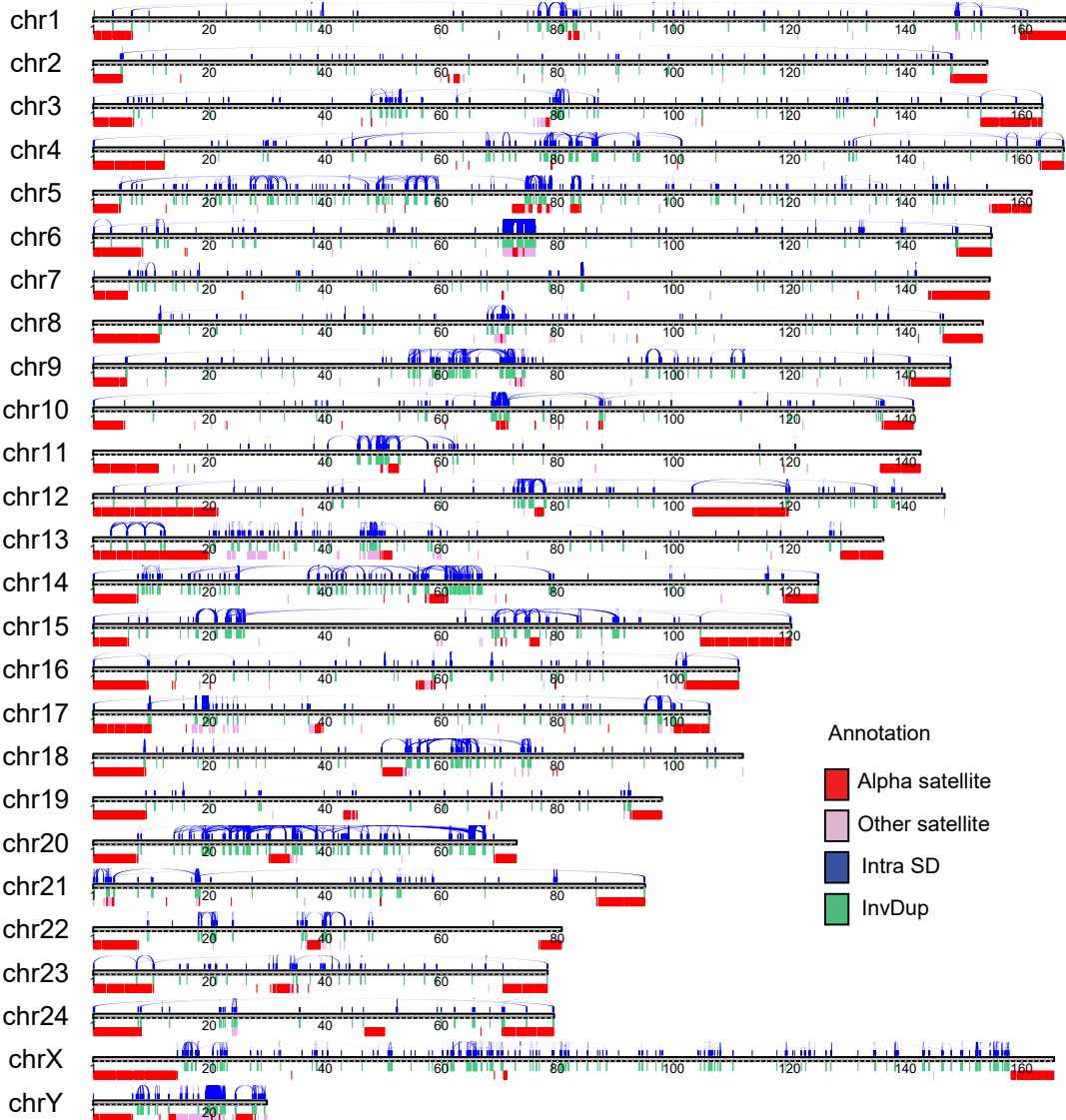

**Extended Data Fig. 1 | The siamang genome.** A schematic of the T2T siamang genome highlighting segmental duplications (Intra SDs; blue), inverted duplications (InvDup; green), centromeric, subterminal and interstitial α-satellites (red), and other satellites (pink). Note the large blocks of alpha-satellite defining the siamang subterminal heterochromatic caps.

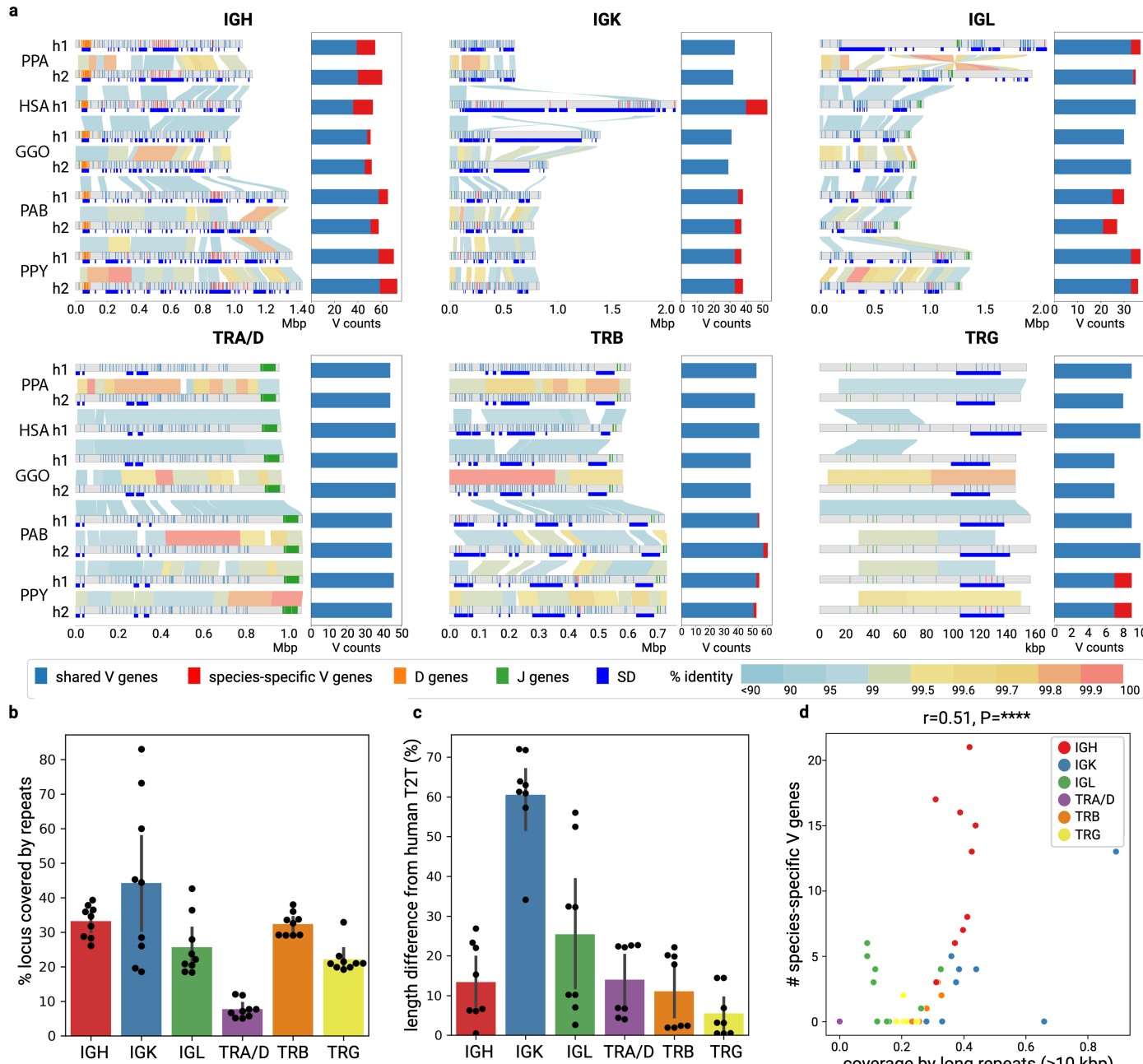

**Extended Data Fig. 2 | IG and TR genome organization in apes. a)** Annotated haplotypes of IGH, IGK, IGL, TRA/D, TRB, and TRG loci across four primate species and one human haplotype (HSA.h1 or T2T-CHM13). Each haplotype is shown as a line in the genome diagram where the top part shows positions of shared V genes (blue), species-specific V genes (red), D genes (orange), and J genes (green) and the bottom part shows segmental duplications (SDs) that were computed for a haplotype pair of the same species and depicted as dark blue rectangles. Human SDs were computed with respect to the GRCh38.p14 reference. Alignments between pairs in haplotypes are shown as links colored according to their percent identity values: from blue (< 90%) through yellow (99.5%) to red (100%). The bar plot on the right from each genome diagram shows counts of shared and species-specific V genes in each haplotype. **b)** Barplots showing the mean percentage of base pairs for each IG/TR locus covered by SDs computed between haplotypes of the same species and collected across five ape species; n = 9 haplotypes including 8 haplotypes of nonhuman ape species and the human T2T haplotype. Here and in panel c, error bars represent the 95% confidence intervals. **c)** Barplots showing length differences (%) computed for all ape IG/TR loci with respect to the corresponding human T2T locus; n = 8 haplotypes of nonhuman ape species. **d)** Counts of species-specific V genes vs. fractions of locus covered by long (≥ 10 kbp) repeats computed for IG/TR loci across five great ape species. Pearson's correlation and p-value ($p = 6.95 \times 10^{-5}$) are shown on the top of the plot.

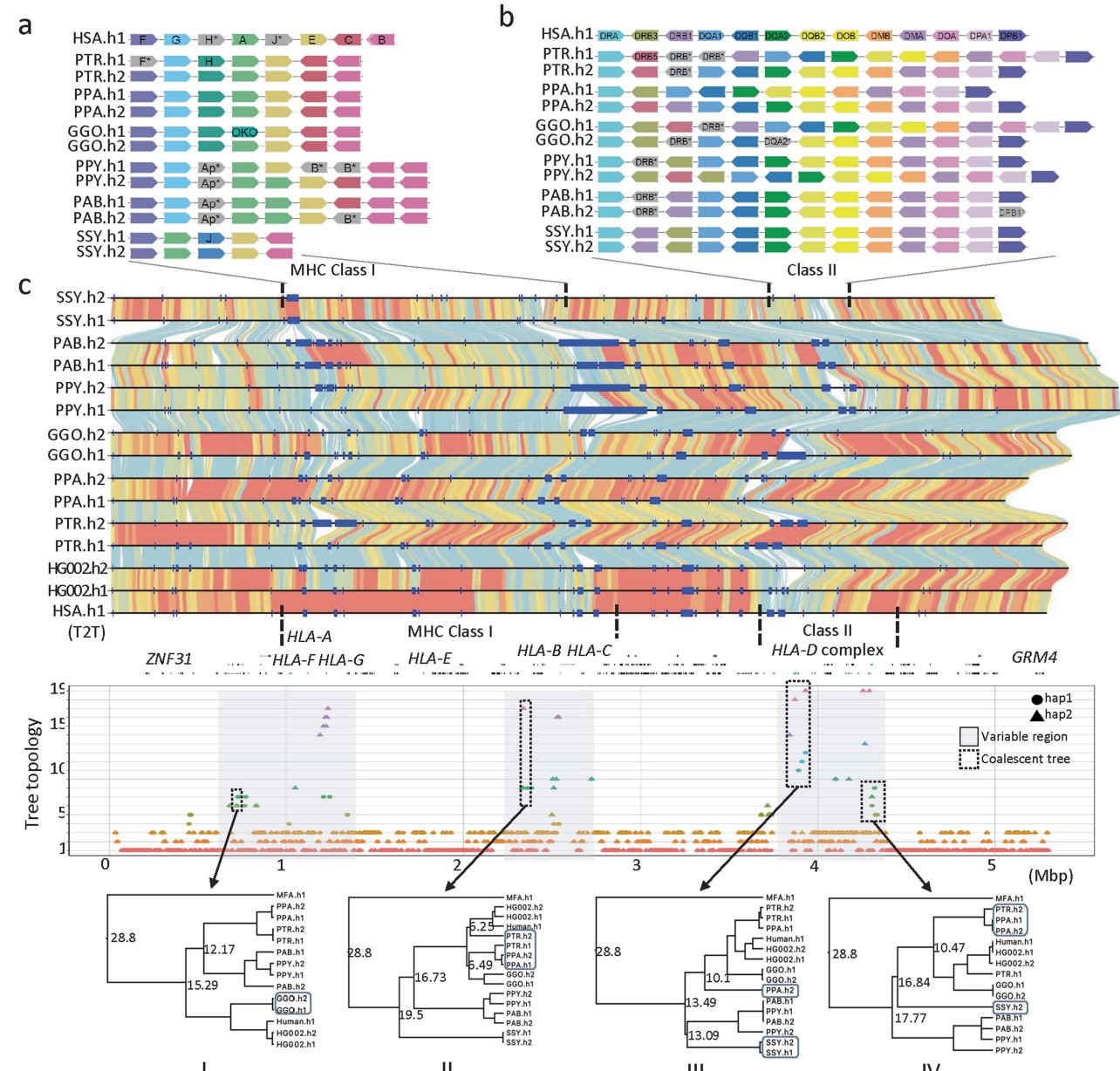

**Extended Data Fig. 3 | Ape MHC organization. a** and **b**) show schematic representation of MHC locus organization for MHC-I and MHC-II genes, respectively, across the six ape haplotypes (PTR.h1/h2, PPA.h1/h2, GGO.h1/h2, PPY.h1/h2, PAB.h1/h2, SSY.h1/2) and human (HSA.h1). Only orthologs of functional human HLA genes are shown. Loci naming in apes follows human HLA gene names (HSA.h1) with the exception of Gogo-OKO that does not have a human homolog, and orthologs are represented in unique colors across haplotypes and species. Orthologous genes that lack a functional coding sequence are grayed out and their name marked with an asterisk. Two human HLA class I pseudogene (HLA-H, HLA-J) are shown, because functional orthologs of these genes were identified in some apes. **c**) Pairwise alignment of the 5.31 Mbp MHC region in the genome, with human gene annotations and MHC-I and MHC-II clusters. Below is the variation in phylogenetic tree topologies according to the position in the alignment. The x-axis is the relative coordinate for the MHC region and the y-axis shows topology categories for the trees constructed. The three prominent subregions with highly discordant topologies are shown through shaded boxes. Four subregions (1-4) used to calculate coalescence times are shown with dashed boxes. The phylogenetic tree includes the macaque genome (MFA)[95] to better estimate the deep coalescence time observed at this locus. Numbers indicate time estimated in millions of years.

**Extended Data Table 1 | Overview of repeat content (Mbp and percentage) in the ape genomes**

| | Human | | Chimpanzee | | Bonobo | | Gorilla | | Sumatran orangutan | | Bornean orangutan | | Siamang | |
|---|---|---|---|---|---|---|---|---|---|---|---|---|---|---|
| | Mbp | % | Mbp | % | Mbp | % | Mbp | % | Mbp | % | Mbp | % | Mbp | % |
| DNA | 108.59 | 3.48 | 110.6 | 3.48 | 110.74 | 3.47 | 111.82 | 3.15 | 111.2 | 3.41 | 110.83 | 3.44 | 106.76 | 3.27 |
| LINE | 633.55 | 20.33 | 636.62 | 20.04 | 638.42 | 19.99 | 649.75 | 18.33 | 686.42 | 21.06 | 683.56 | 21.22 | 631.83 | 19.37 |
| PLE | 0.06 | 0 | 0.07 | 0 | 0.07 | 0 | 0.07 | 0 | 0.07 | 0 | 0.07 | 0 | 0.07 | 0 |
| LTR | 272.91 | 8.75 | 274.9 | 8.65 | 275.67 | 8.63 | 277.85 | 7.84 | 279.36 | 8.57 | 278.28 | 8.64 | 256.49 | 7.86 |
| SINE | 393.51 | 12.62 | 391.01 | 12.3 | 393.08 | 12.3 | 397.19 | 11.2 | 396.85 | 12.17 | 395.47 | 12.28 | 402.73 | 12.34 |
| RC | 0.45 | 0.01 | 0.46 | 0.01 | 0.46 | 0.01 | 0.46 | 0.01 | 0.46 | 0.01 | 0.46 | 0.01 | 0.44 | 0.01 |
| Retroposon | 4.31 | 0.14 | 4.78 | 0.15 | 4.9 | 0.15 | 5.21 | 0.15 | 3.87 | 0.12 | 4.39 | 0.14 | 6.66 | 0.2 |
| Satellite | 161.77 | 5.19 | 251.68 | 7.92 | 270.19 | 8.46 | 462.5 | 13.04 | 191.45 | 5.87 | 159.2 | 4.94 | 376.7 | 11.55 |
| Simple/low | 108.38 | 3.48 | 69.17 | 2.18 | 107.68 | 3.37 | 181.36 | 5.11 | 134.09 | 4.11 | 138.98 | 4.31 | 54.29 | 1.66 |
| Other | 5.34 | 0.17 | 2.21 | 0.07 | 2.61 | 0.08 | 2.15 | 0.06 | 2.23 | 0.07 | 2.35 | 0.07 | 1.34 | 0.04 |
| RNA | 2.78 | 0.09 | 1.55 | 0.05 | 1.53 | 0.05 | 1.38 | 0.04 | 1.89 | 0.06 | 1.59 | 0.05 | 1.36 | 0.04 |
| Un-masked | 1,434.70 | 46.02 | 1,434.70 | 45.15 | 1,389.24 | 43.49 | 1,456.09 | 41.06 | 1,451.97 | 44.54 | 1,445.75 | 44.89 | 1,424.20 | 43.65 |
| Total masked | 1,691.65 | 54.27 | 1,743.04 | 54.85 | 1,805.35 | 56.51 | 2,089.74 | 58.94 | 1,807.88 | 55.46 | 1,775.19 | 55.11 | 1,838.69 | 56.35 |
| NUMT | 0.64 | 0.021 | 0.79 | 0.025 | 0.87 | 0.027 | 0.63 | 0.018 | 0.76 | 0.024 | 0.84 | 0.026 | 0.52 | 0.016 |

PLE: Penelope-like retroelements; LTR: long terminal repeats; RC: rolling circle repeats.

# Reporting Summary

## Statistics

For all statistical analyses, confirm that the following items are present in the figure legend, table legend, main text, or Methods section.

| n/a | Confirmed | |
|---|---|---|
| ☐ | ☒ | The exact sample size (*n*) for each experimental group/condition, given as a discrete number and unit of measurement |
| ☐ | ☒ | A statement on whether measurements were taken from distinct samples or whether the same sample was measured repeatedly |
| ☐ | ☒ | The statistical test(s) used AND whether they are one- or two-sided<br>*Only common tests should be described solely by name; describe more complex techniques in the Methods section.* |
| ☒ | ☐ | A description of all covariates tested |
| ☐ | ☒ | A description of any assumptions or corrections, such as tests of normality and adjustment for multiple comparisons |
| ☐ | ☒ | A full description of the statistical parameters including central tendency (e.g. means) or other basic estimates (e.g. regression coefficient) AND variation (e.g. standard deviation) or associated estimates of uncertainty (e.g. confidence intervals) |
| ☐ | ☒ | For null hypothesis testing, the test statistic (e.g. $F$, $t$, $r$) with confidence intervals, effect sizes, degrees of freedom and $P$ value noted<br>*Give P values as exact values whenever suitable.* |
| ☐ | ☒ | For Bayesian analysis, information on the choice of priors and Markov chain Monte Carlo settings |
| ☒ | ☐ | For hierarchical and complex designs, identification of the appropriate level for tests and full reporting of outcomes |
| ☒ | ☐ | Estimates of effect sizes (e.g. Cohen's *d*, Pearson's *r*), indicating how they were calculated |

*Our web collection on statistics for biologists contains articles on many of the points above.*

## Software and code

Policy information about availability of computer code

| Data collection | The software used to collect sequencing data include Pacific Biosciences Sequel IIe instrument, using version 2.0 sequencing reagents (control software version 10.1.0.119549), SMRT Link v12. |
|---|---|
| Data analysis | The custom scripts used in this study are available by the following:<br><br>Overall scripts: (https://github.com/marbl/Primates), polishing of the genome (https://github.com/arangrhie/T2T-Polish), assembly QC (https://github.com/EichlerLab/assembly_eval), implicit graph & pangenome (https://github.com/T2T-apes/ape_pangenome), Cactus alignment (https://cglgenomics.uc.sc.edu/february-2024-t2t-apes), assessment of ancestral sequence (https://github.com/shanksc/ancestral_state), AQER analysis (https://github.com/vertgenlab/vglDocumentation/tree/master/primateT2T), population genome processing and selection analysis (https://github.com/aabiddanda/haplotype-phasing), acrocentric/rDNA analyses (https://github.com/jouyun/2024_Primate_rDNA, https://github.com/borcherm/primate_rdna_cn), species-specific MEI analysis (https://github.com/Markloftus/t2t-ape-MEIs), non-B DNA annotation and NUMT detection, (https://github.com/makovalab-psu/T2T_primate_autosomes), and transcript comparison (https://github.com/canzarlab/apes_transcriptome_analysis).<br><br>In addition to the custom scripts, the following codes were used: Alignment (https://github.com/arangrhie/T2T-Polish/tree/master/winnowmap (v2.03/v1.0), lastz (v1.04), minimap2(v2.24/v2.26/v2.28), blastn (v2.12.0), blastp (v2.12.0), https://github.com/waveygang/wfmash, https://github.com/marbl/MashMap v3.1.1, NUCMER), alignment processing (https://github.com/AndreaGuarracino/paf2chain, http://github.com/wjwei-handsome/wgatools, https://github.com/mrvollger/rustybam v0.1.29, https://github.com/sstadick/perbase), conservation score calculation (PhastCons v1.5), pan-genome graph (https://github.com/pangenome/impg), further assembly QC (https://github.com/mobinasri/flagger v0.3.3), non-B DNA annotation (https://github.com/abcsFrederick/non-B_gfa), gene annotation (https://github.com/ComparativeGenomicsToolkit/Comparative-Annotation-Toolkit, IgDetective, Digger, Exonerate v2.4), repeat annotation |

(Repeatmasker v4.1.0/v4.1.5/v4.1.6, TRF v4.1.0, ULTRA, windowmasker v2.2.22, http://doua.prabi.fr/software/one-code-to-find-them-all), transcriptome data alignment (StringTie2 v2.2.1), phylogenetic tree (IQ-TREE v2.1.2) ILS (https://github.com/rivasiker/trails, https://github.com/stschiff/msmc2), selection signature scans (Sweepfinder2, saltiLASSI), TOGA (https://github.com/hillerlab/TOGA), replication timing (https://github.com/ma-compbio/Phylo-HMGP), structural variation calling (https://github.com/schneebergerlab/syri v1.6.3, https://github.com/EichlerLab/pav v2.3.2), segmental duplication (https://github.com/vpc-ccg/sedef v1.1), alpha satellites higher order array prediction (https://github.com/fedorrik/HumAS-HMMER_for_AnVIL), data visualization (https://github.com/daewoooo/SVbyEye, https://github.com/mrvollger/StainedGlass v0.5).

For manuscripts utilizing custom algorithms or software that are central to the research but not yet described in published literature, software must be made available to editors and reviewers. We strongly encourage code deposition in a community repository (e.g. GitHub). See the Nature Portfolio guidelines for submitting code & software for further information.

## Data

Policy information about availability of data

All manuscripts must include a data availability statement. This statement should provide the following information, where applicable:
- Accession codes, unique identifiers, or web links for publicly available datasets
- A description of any restrictions on data availability
- For clinical datasets or third party data, please ensure that the statement adheres to our policy

The raw genome sequencing data generated by this study are available under NCBI BioProjects, PRJNA602326, PRJNA976699–PRJNA976702, and PRJNA986878–PRJNA986879 and transcriptome data are deposited under BioProjects, PRJNA902025 (UW Iso-Seq) and PRJNA1016395 (UW and PSU Iso-Seq and short-read RNA-seq). The genome assemblies are available from GenBank under accessions: GCA_028858775.2, GCA_028878055.2, GCA_028885625.2, GCA_028885655.2, GCA_029281585.2 and GCA_029289425.2. Genome assemblies can be downloaded via NCBI (https://www.ncbi.nlm.nih.gov/datasets/genome/?accession=GCF_028858775.2,GCF_029281585.2,GCF_028885625.2,GCF_028878055.2,GCF_028885655.2,GCF_029289425.2). Convenience links to the assemblies and raw data are available on GitHub (https://github.com/marbl/Primates) along with a UCSC Browser hub (https://github.com/marbl/T2T-Browser). The UCSC Browser hub includes genome-wide alignments, CAT annotations, methylation, and various other annotation and analysis tracks used in this study. The T2T-CHM13v2.0 and HG002v1.0 assemblies used here are also available via the same browser hub, and from GenBank via accessions GCA_009914755.4 (T2T-CHM13), GCA_018852605.1 (HG002 paternal), and GCA_018852615.1 (HG002 maternal). The alignments are publicly available to download or browse in HAL118 MAF and UCSC Chains formats (https://cglgenomics.ucsc.edu/february-2024-t2t-apes).

Additional public data also include 22 genome assemblies summarized in Table ASM.S11, as well as the human reference genome, GRCh38. Previous genome annotation files include gorilla (GCF_029281585.2), bonobo (GCF_029289425.2), chimpanzee (GCF_028858775.2), Sumatran orangutan (GCF_028885655.2), and Bornean orangutan (GCF_028885625.2) from NCBI. For the pangenome graph, public human genome assemblies (https://github.com/human-pangenomics/HPP_Year1_Data_Freeze_v1.0) were used. The short-read genome sequencing data for great ape species were obtained from SRP018689, ERP001725, ERP016782, and ERP014340. The public methylation data used in this study include the raw PacBio HiFi reads with kinetics and methylation tags for human (https://humanpangenome.org/data/) and for chimpanzee, gorilla, Bornean orangutan, Sumatran orangutan, and siamang gibbon (https://www.genomeark.org/t2t-all/).

## Research involving human participants, their data, or biological material

Policy information about studies with human participants or human data. See also policy information about sex, gender (identity/presentation), and sexual orientation and race, ethnicity and racism.

| Reporting on sex and gender | N/A |
|---|---|
| Reporting on race, ethnicity, or other socially relevant groupings | N/A |
| Population characteristics | N/A |
| Recruitment | N/A |
| Ethics oversight | N/A |

Note that full information on the approval of the study protocol must also be provided in the manuscript.

# Field-specific reporting

Please select the one below that is the best fit for your research. If you are not sure, read the appropriate sections before making your selection.

☒ Life sciences ☐ Behavioural & social sciences ☐ Ecological, evolutionary & environmental sciences

For a reference copy of the document with all sections, see nature.com/documents/nr-reporting-summary-flat.pdf

# Life sciences study design

All studies must disclose on these points even when the disclosure is negative.

| | |
|---|---|
| Sample size | One cell line per species was used (out of six species: chimpanzee, bonobo, gorilla, B. orangutan, S. orangutan and siamang). To determine polymorphic status, we further used one additional near-T2T assembly per species, summarized in Table ASM.S11. |
| Data exclusions | No data were excluded. |
| Replication | For the FISH experiment to validate inversion (hsa16 inversion; Fig. 3b), we repeated for three times and 10 metaphase spreads with the corresponding fluorochromes captured for each experiment. |
| Randomization | N/A; randomization was not applicable as the samples were obtained from previous research (for additional data generation). Also, with n=1 for each species, makes it impossible to control selection bias. |
| Blinding | N/A; sample size equal to one technically makes blinding ineffective to control for bias. |

# Reporting for specific materials, systems and methods

We require information from authors about some types of materials, experimental systems and methods used in many studies. Here, indicate whether each material, system or method listed is relevant to your study. If you are not sure if a list item applies to your research, read the appropriate section before selecting a response.

## Materials & experimental systems

| n/a | Involved in the study |
|---|---|
| ☒ | ☐ Antibodies |
| ☐ | ☒ Eukaryotic cell lines |
| ☒ | ☐ Palaeontology and archaeology |
| ☒ | ☐ Animals and other organisms |
| ☒ | ☐ Clinical data |
| ☒ | ☐ Dual use research of concern |
| ☒ | ☐ Plants |

## Methods

| n/a | Involved in the study |
|---|---|
| ☒ | ☐ ChIP-seq |
| ☒ | ☐ Flow cytometry |
| ☒ | ☐ MRI-based neuroimaging |

## Eukaryotic cell lines

Policy information about cell lines and Sex and Gender in Research

| | |
|---|---|
| Cell line source(s) | KB8711 or PR00251 (San Diego Zoological Society), AG18354 (Coriell), AG06213 (Coriel), AG05252 (San Diego Zoological Society), KB3781 or Jim (San Diego Zoological Society), Jambi (Oregon Health and Science University) (detailed in Table Assembly.S1). |
| Authentication | Each cell line was authenticated as described in Makova et al. 2024 (Note S1 - for orangutan and S2 - for chimpanzees). In addition to the documentation available (Studbook number), multiple analyses were performed to authenticate our samples. The previous study by Makova et al., observed species-specific genomic signatures, including the nucleolar organizing region of Y chromosome in Sumatran orangutan, and the mitochondrial genome analyses suggesting the Bornean or Sumatran orangutan origins in the respective samples.<br><br>Between chimpanzee and bonobo, the previous study also authenticated the respective species by mitochondrial genome. In addition to this, we also further validated species by comparing with the short-read genome-wide SNP PCA (Fig. Sequencing.S1-S5). We further found average genome-wide sequence identity between two Sumatran orangutan assemblies (one previous and T2T) and one Bornean orangutan assembly (T2T), to identify greater sequence identity among the respective species compared to between two species, as expected (Fig. Sequencing S6). |
| Mycoplasma contamination | The cell lines were tested negative for mycoplasma contamination. |
| Commonly misidentified lines (See ICLAC register) | No commonly misidentified cell lines were used. |

## Plants

Seed stocks

N/A; no plants were used in this study.

Novel plant genotypes

N/A

Authentication

N/A

