## [Peer Review file · Nature]

Complete sequencing of ape genomes

Corresponding Author: Professor Evan Eichler

Version 0:

Reviewer comments:

Referee #1

(Remarks to the Author)

In this manuscript, (virtually) complete genomes from 6 apes (chimpanzee, bonobo, gorilla, two orangutans and siamang) are presented and analysed. It represents an incredible amount of analysis and results, and many new findings are interspersed with confirmation or smaller increments of previous findings. How to find out which is which was a real challenge.

General comments

First, I will applaud the quality of the genomes and, not least, the very careful annotation. These genomes should immediately replace previous reference genomes of these species, and they will be a very valuable resource for other researchers, who will also be able to exploit this data much more than is possible, even in this very long manuscript. The supplement is long but nicely organised into sections authored by different subgroups of authors, and they can be read independently. Most of the main points of the supplement are referred to in the main text. The assembly strategy etc is also well laid out.

The main text is close to 14000 words with nine densely packed, multipanel Figures. I think the manuscript suffers severely from information overload; it reads like eight incomplete papers in one. Many Figure panels are very complex, yet they are not always referred to in the text, and when they are, it is often very cursory. Thus, figures often need to be understood from the short legends alone, and it is left a bit to the reader to make her/his interpretations and explain her/his own reasons for why they were included. The sheer amount of information makes it difficult for this reviewer to check whether individual claims are well supported, not least because inferences are often qualitative from the Figures rather than quantitative and supported by statistics. It would be nice if there were less of such claims throughout.

Many analyses are very descriptive and unrelated to any hypothesis or prior expectation. While this is very understandable for this kind of data, it would still be very useful if, for each new pattern analysed, there is first a clear motivation, analysis and then a clear conclusion (which we sometimes have to wait for the discussion to get).

Thus, while I believe analyses are generally very competently carried out, the manuscript could be much more impactful if many analyses were removed and many Figures were simplified. These are the analyses that repeat (with better genomes) previous findings with marginal gains such as sweep finding, ILS, and structural variation findings (see below for details). In between are very interesting results that almost "drown" and they would benefit from the addition of more detail and interpretation. It is difficult to see how the findings relate to other recent studies by some of the same authors, for example, the human T2T papers and the recent SD paper in Cell. In the cell paper, female individuals are sequenced with long reads (not quite as complete, though), but there is no comparison with the findings of this paper, and there is no joint analyses. That could perhaps help delineate how much variation in different types of repetitiveness (and structural variants) is found within species as compared to between. This is done in some cases here when the two haplotypes can be compared but it is not done consistently and this makes it hard to evaluate which patterns are truly divergent between species. A short discussion on this would be great and could perhaps replace some of the summaries of results that are now included in the discussion. Also, what differences are found on the autosomes and the sex chromosomes, which were recently published in Nature in a paper much easier to read than the present manuscript

I find that the most novel and interesting results concern what has been very inaccessible until now. These results could be

discussed more in light of previous research and more care could be made to illustrate specific points of greatest novelty.

They are

MHC: two full haplotypes per species are impressive, but mining the Cell paper data could add two more for each species. Then, a more detailed evolutionary analysis of dn/ds, the divergence between haplotypes and coalescent times of pairs of alleles could be estimated. A more detailed TRAILS analysis could be made on the alignable fractions as well. How do the results relate to the large knowledge about human MHC/HLA, including known GWAS findings?

NOR+/acrocentric chromosomes: To me this is the highlight of the manuscript. The dynamics of rDNA evolution is very impressive and can also be very important for both some syndromes (in humans) and for reproductive isolation as suggested. I would like more details on this including interindividual variations and a comparison to previous results such as the T2T human genomes

Centromeres: Also very intriguing with same comments as for NOR+. More about evolutionary rates and species polymorphisms would be very valuable as would comparisons to human T2T results. Can more be said about the bonobo short centromeres for some chromosomes. Is that also found in other mammals/primates?

Subterminal repeats: This is also very interesting. I know little about these but they were noticed also on the X in the recent X/Y primate paper. Could more be speculated on their function and could a hypothesis be formulated for future research?

Sections on segmental duplications are very interesting. However, I lack considerations of which are polymorphic within species and which have shared polymorphisms between closely related species such as the two oranges and the chimp and bonobo. Any sign of adaptive introgression between species for MHC as could be expected due to balancing selection? The transposable element section is also interesting but again, I would like a consideration of the number of intraspecies polymorphisms as a way to measure recent transpositions more directly.

I am less intrigued by the results on

Pan-genome graph: What is it supposed to tell us, and how can it be used?

ILS: What new insights have been gained? Is ILS of repetitive regions important and if so, why? Can we say anything about the selective neutrality of these?

Selection and diversity. Is it important to have more putative sweeps identified? Is there a difference in the newly versus previously identified sweeps? Is the difference due to the better genomes or differences in methods?

Immunoglobulin genes: I find the results very hard to evaluate as they are now explained. I would like to have either more motivation and details or less emphasis on these results. What are the expected error rates in orthology inference?

Methylation: It is unclear to me if two different cell line types are very interesting for methylation inference. It is also unclear what we learn from these

Replication timing: Again, what do we learn?, is the tissue relevant? Are differences between species relatable to other genome features, I remember claims that recombination is important. GC- content as well

Accelerated regions. These could be interesting but the discussion is hard to follow - would be great with a simpler rewrite of this. That could include statistical tests for the suggestions/claims put forward.,

NUMTS: What makes them interesting here? Why is 8% difference between orang species interesting

Specific comments and questions

Assemblies: Is it possible to give good advice for optimal strategies for more T2T primate genomes going forward? How important is nanopore compared to pacbio and what should be ideal coverages of each?

L 236 states that contiguity is an order of magnitude improved; how should that be understood (Cell paper has N50 contig size of 50Mb or above)

315 "We infer ILS for an average of 39.5% of the autosomal genome and 24% of the X chromosome, representing an increase of approximately 7.5% compared to recent reports from less complete genomes²⁴ in part due to inclusion of more repetitive DNA"

It is not clear what the 7.5% refers to

L 353 "Similarly, we observed 33,032 (0.7%) soft-clipped reads (>200 bp) in the gorilla T2T assembly in contrast to 89,498 (2%) soft-clipped reads"

Why so many soft-clipped reads left even with the new annotation? Is it technical?

L 548: "We identified 20 states with different patterns of replication timing. Overall, the replication timing program is largely conserved, with 53.1% of the genome showing conserved early and late replication timing across primates, while the remaining regions exhibit lineage-specific patterns..."

I am not sure what states refer to and how to evaluate these differences

980 These include an improved and more nuanced understanding of species divergence, human ancestral alleles, incomplete lineage sorting, gene annotation, repeat content, divergent regulatory DNA, and complex genic regions as well as species-specific epigenetic differences involving methylation

This quote from the discussion also points to these sections as perhaps not the most important to report on in this manuscript since more novel results can “drown” in the large body of information

Referee #2

(Remarks to the Author)

This paper reports haplotype-resolved T2T assemblies for 6 ape species, including an outgroup to the great apes. The included clades are key to studies of human evolution, and although highly contiguous assemblies for them were previously available the authors set out to bridge the remaining gap in quality to human sequence, enabling a comparison across several structurally complex sequence classes, for many of which this was previously not possible genome-wide. The manuscript includes an extensive characterization of the resources across numerous analyses, which the authors broadly group into a description of the assemblies, improvements of analyses based on them, and a description of newly resolved sequences. The methods to bring the whole genomes to near completion are state of the art, I have no concerns or comments regarding their quality, not least as the same approach applied to the human genome has previously been reported by some of the authors, and the improvements are widely recognized. It's wonderful to see the assemblies of some key NHP species being brought to the same quality, a commendable effort that will undoubtedly be of wide use across several fields.

The manuscript aims to place the new assemblies in the context of previous analyses, while also describing the previously inaccessible sequences. The improvement of the genomic resources in the latter is self-evident. For the former, some analyses might sometimes provide more nuance, but this is in part difficult to assess or follow given the somewhat frequent lack of context. In the best case, the reader is then unable to appreciate the refinements and improvements, but it might also give the impression of novelty for results that are at times rather confirmatory (a fact that also in part acknowledged and does not diminish the importance of the work in any way). I've listed examples below, given the tradeoff between clearly communicating improvements and the length of the text, the authors might want to reconsider the presentation of some of these sections to make the paper more accessible, although this is ultimately their choice. The same applies to those reporting some one-offs that are named in the text but don't seem to be used elsewhere, or whose use is at least not immediately evident, e.g. several different alignments or 3D maps, among others.

While I have no concerns regarding the assemblies themselves, I do for a subset of the more focal analyses, which I hope the authors can clarify. They are listed together with additional questions and comments below.

###

It is difficult to follow the provenance and identity of different cell lines or other samples. The available information is currently scattered across e.g. Table 1, Table AssemblyS1, the supplementary with references to prior work, or the reporting summary. The tables include species-level aggregates, the identifiers for the parental samples seem to be completely missing. The authors state that the journals' policy considerations regarding “Research animals and/or animal-derived materials that require ethical approval” do not apply, a statement that is challenging to evaluate. Please include a table reporting uniquely identifiable sample-level information for all samples used throughout this manuscript, including at least information on the source biorepository, biosample id, cell type, primary or immortalized, immortalization type if relevant, all generated data types, and provenance of the samples where available, particularly for samples that are not in public biobanks. I would also appreciate any additional information the authors might be able to retrieve and deem potentially relevant (e.g. number of passages prior to expansion, year the cell line was established, information on the origin, e.g. from an individual in a Zoo, Sanctuary, their habitat, wild-born / first generation captive, captive bred, etc.). I have no reason to question the legitimacy of the source of sampling, but the numerous ethical considerations and resulting sensitivities surrounding the research use of endangered animals are probably best preventively addressed with as much information as possible. It might be possible to compile this information in part via the referenced prior work, but I believe the high visibility I expect this manuscript to receive warrants not leaving this task to the reader.

I did not find any mention of the potential impact of using immortalized LCLs, and maybe other cell lines, which are prone to numerous alterations affecting the whole spectrum of variation described in the manuscript (beyond somatic rearrangements in VDJ genes), as a result of the transformation process itself and cell propagation. It seems karyotype normality was assessed and the authors describe EBV contamination that required filtering, were any additional measures taken to ensure other potential alterations do not affect the described sequences and conclusions? I believe a convincing assessment of this matter is very difficult to do, especially for most newly described sequence classes or variant types, it might be possible to at least increase confidence indirectly via established markers. Where do the great-ape samples fall relative to the well-characterized ranges of variation of the diversity panel the authors used, e.g. in a PCA? I don't necessarily expect the samples to behave like natural populations or think they are required to, but it seems important to ascertain they don't behave completely differently where possible, and this potential limitation should be clearly communicated. On a similar note, were any measures taken to ensure correct species labels, particularly the Siamang?

L160: I have strong reservations about the use of “definitive” here. Disregarding remaining gaps and what should be considered a “complete” assembly, science is always subject to revision, and thus a result can hardly, if ever, be considered definitive.

L259: The authors have generated numerous alignments for this manuscript (at least 5 mentioned in this section, but there seem to be more referenced in other places). On the one hand, it is sometimes challenging to follow which of them was used for which analyses. On the other, it is not evident why this was deemed necessary, as several seem to be redundant, e.g. the various cactus alignments appear to be subsets of one, nor what the consequence of using a particular one is beyond the few cases of comparison.

L280 & Sup. P25: Please explicitly state that phastCons was used to calculate the scores and report the model parameters. The legend of Fig PanGenomeS1 D claims protein-coding sequences as most conserved, 4 other classes with higher average scores are shown and IG_V genes are presumably also protein-coding. The text states that lncRNA are among the least conserved, but they show among the highest scores in the figure. The authors need to include a length-matched baseline of scores for all analyses, as Figure SeqDiv S3 suggests the distribution of these values to be in line with what can be expected from sampling them randomly across the genome. The interpretation "Conservation scores were strongly skewed towards 1 (mean 0.964), as expected for such closely related species." is flawed and the presented results suggest that ~96% of the human genome is under purifying selection, when the high similarity of ape genomes and corresponding short branch lengths is largely the result of their shallow divergence. The authors further report a strong correlation with pLI, but do not provide a correlation coefficient for what seems to be a surprisingly modest effect (PanGenomeS1 E). They describe genes off the linear trendline with apparently differing effects of selection in either metric, their number and function are not reported but the figure suggests such cases are frequent. These results seem unexpected by themselves given the inclusion of the human genome, but also in the light of recent large-scale reports on selective constraint across mammals and primates and should warrant further investigation. Taken together they do not instill confidence in the scores' utility, which are likely severely underpowered. The analyses are buried in the supplementary text, but it seems all data has already been released, calling for caution given their potential usage in other projects.

L296 / Fig 2b: The reported gap-divergence metric is very difficult to interpret, as it combines multiple sources of biological and technical variability, is not symmetrical, and appears highly subject to choices of the deployed alignment strategy or its filters, as evidenced by its distribution. If a region is single-copy in species 1 and duplicated in species 2, should the two copies be aligned to the same region in species 1 or one considered as counting towards gap-divergence? Should this scenario contribute in the same way to gap divergence as e.g. a deletion or failure to reliably align a centromere or any other satellite array? Which haplotype should be chosen, Table Div1 shows substantial differences between them, but Figures SeqDiv S1-2 only report distributions for one. Do the others look the same? While a discussion on the proportion of the assemblies that fails to reliably align for whatever reason seems adequate, it's not clear why this should be elevated to a metric that is comparable to the biologically much more clearly defined SNV divergence in orthologous regions in Fig 2b. I appreciate the desire to communicate that divergence assessed via SNVs doesn't capture all differences between species, if the authors wish to do so this should be done with more nuance in my opinion.

L303: The genome of the Bornean Orang has been sequenced on several occasions (e.g. Prado et al., 2013, Nater et al., 2017), and a genome assembly for this species has been reported in Shao et al., 2024, referenced here.

L307: The referenced table SeqDiv. S2 does not seem to exist.

L310, Sup. P39: The methods state that "ILS estimation on the following four-species (ABCD) phylogenies" was performed. Do the estimated percentages of ILS represent aggregates of all possible topologies different from the canonical tree (which seems to be ((A, B), C), D), or do they test a specific alternative? Please include a description of the rationale for choosing the specific quartets. The authors also ran msmc and chose a constant mutation rate across all species to scale the results of both analyses, which might not be realistic for all species, particularly the Siamang. Could they please elaborate on the potential impact this might have, which reported estimates are based on trails or msmc (this is only mentioned in the figure legend), and how well these estimates agree for nodes reconstructed in both? Is there any range of uncertainty for the Ne estimates, like they seem to provide for time? The methods mention a multiz alignment which doesn't seem described elsewhere.

I miss a contextualization of the results reported here (other than the % increase of ILS) with prior estimates for the same nodes, despite diverging estimates even when based on the same method. For example, the Ne estimate for the African ape ancestor appears to be 25% higher than that reported in Rivas-Gonzales et al., 2023, and the estimates for the pan ancestor are 33% higher. The authors state to take the latter estimate with caution, but it is neither clear why, nor why confidence is higher for the deeper nodes. These differences seem substantial but might not be appreciated by the reader if unmentioned, nor does it seem sufficiently justified to simply consider them improvements rather than differences. The supplement mentions bp-resolution, suggesting it should be possible to subset the resulting estimates by regions without the requirement to rerun the analyses, if that is the case it would seem like a good opportunity to better understand these differences.

L314: What do the ranges represent?

L317: What is the basis for the claim that increased ILS is due to repetitive regions? If this refers to observations on the HLA, do the authors observe a difference in genome-wide estimates after excluding it, or other previously unresolved regions?

L343: Could the authors please clarify how the non-SD-based differences were defined, i.e. if they were detected based on the alignment or a lack of detectable orthologous sequence for the NHP-based gene model? Can anything be stated about the function in either category (lineage-specific SD or others, presumably human losses?)

L361: "We now estimate.." do these estimates differ from prior ones across repeat categories, outside satellite/VNTRs?

L379: Did the authors observe any differences in methylation status of these lineage-specific MEIs compared to similar shared ones, is there any evidence for differences in transcriptional activity?

L391: Is there any evidence for competition, i.e. that the difference between L1 and Alu activity is related to each other?

L434: The scales for Ne and time are missing.

L405, L505, supp P60: The authors mapped a diversity panel on NHP to the new assemblies. As most of the improvements within them are due to structurally complex regions such as duplications that are now resolved, it is not evident how they can be reliably genotyped with short reads. Calls that are the result of ambiguous mappings from e.g. duplications have the potential to result in biased estimates of diversity and allele frequencies that the selection scans are based on. The description of the filtering criteria is sparse and vague, e.g. 'applying standard quality control to SNPs and indels', 'bi-allelic sites that fell within high-confidence regions', but don't provide definitions of standard quality control or high-confidence regions. Without knowing how the authors ensured correct variant calls and associated summaries in structurally challenging/multicopy genes, and how callable regions were defined in the assemblies, it is not possible to evaluate if the results they present can be considered reliable.

L467: The definition of "ape-specific" to exclude human is awkward. It might be more appropriate to call them 'genes absent in human' or something else, as they are neither present across apes, nor does the description suggest that specificity to apes was tested via an outgroup. Though there seems to be a macaque in the lower panels of Fig 3d, a description of its inclusion and treatment and reference seems completely absent.

L498: How do the coalescent times estimated for these loci compare to those estimated by the trails analysis for the same region?

L499: How was missing data in the alignment for this analysis dealt with (both row/haplotype-wise and position-wise), given the high degree of polymorphism in this region? Which 3 topologies represent 96% of the region, and how well are they supported?

L533: What does first-generation mean here?

L543: There does seem to be no mention of 3D chromatin organization in Supplementary Notes XII-XIII, or anywhere else (other than phasing the haplotypes).

L537 & Sup. P96: The main text states that the authors distinguish between hyper- and hypomethylated promoters, but the methods only mention a distinction between consistently and variably methylated promoters, regardless of whether methylation levels are high or low. For the analyses, the authors establish a threshold of 15% divergence in methylation status to report that ~17% of promoters are variably methylated across species. Neither the rationale for the threshold nor the resulting number of differentially methylated promoters they expect to observe by chance are evident. The absence of a statistical framework or its description does not enable the reader to contextualize the results. The same applies to the 25% threshold established to define lineage-specific methylation. The referenced BioProject PRJNA101639 appears to be associated with a study on gene expression in honeybee antennae. How was callability across complex regions dealt with for the short-read mappings?

L573: While previously undescribed events are reported, the recall rate for previously described inversions across the sources they compare is not. If there are differences, what do they attribute them to?

L593: The authors do not report what the r^2 value corresponds to, but to set an expectation for the number of inversions on the human lineage some kind of phylogenetic comparative method probably needs to be applied, rather than an OLS linear model.

L617, Sup P108: The methods description on the AQER analyses is difficult to follow, leaving many questions that result in an incomplete understanding of the analyses. I suspect the diploid 16-way cactus alignment was used here, but this is not evident from the text, which suggests that previously unnamed pairwise human-other species alignments were generated for this analysis. If that is the case, please include a full description. For example, what does the scoring threshold refer to, how was the ancestral sequence reconstructed and how were base probabilities calculated, was the ancestral sequence defined individually for each species as suggested by the statement regarding including the extant species' allele? I'm also curious to know how the diploid haplotypes were dealt with when counting variants, and how gaps/missing data were dealt with, particularly for the larger windows.

Somewhat alarmingly, the methods suggest that a hard threshold of >29 substitutions / 500bp window between the ancestral genome and focal extant species was used to define windows with significantly more divergence than expected, without explaining what this expectation or claims of significance are based on, and that this threshold was used across windows and species. If correct, this raises serious concerns about the validity of the analyses, as the expected number of substitutions will be strongly influenced by the sequence composition of the window, the branch length between the ancestral node and extant species, but likely also large-scale regional differences in substitution rates along chromosomes (see e.g. Mikkelsen et al. 2005, Fig 2), all of which need to be considered. While the chosen threshold might happen to be conservative enough to capture AQERS across the different scenarios, this would probably yield differing detection sensitivities across species rendering any quantitative claims on enrichment resulting from subsequent comparative analyses invalid.

L620: Why was the orang excluded?

L621: Could the authors please provide the full comparison with previously defined HAQERS, e.g. a Venn diagram?

L624 / Fig. 2a: The statement does not seem to be supported by Figure 2a, which suggests there is an enrichment across NHP AQERS for bivalent promoters, albeit less than for HAQERS. Could the authors please provide summaries for the enrichments? What is the difference between this figure and Fig. AQER.S1c, which appears to provide different results for the same analyses?

It might be possible to ascertain if differences are due to human-based chromatin states by comparing them with NHP-based states (e.g. Garcia-Perez et al., 2021), should the authors wish to corroborate the claim. It might also be interesting to differentiate between preexisting CREs with a newly acquired function versus wholly new CREs by looking at their sequence constraint across other species.

L645: The authors report 770-1420 protein-coding genes as specific to the NHPs and absent in humans in L342, but only 6 as restricted to humans here. I'm having difficulties reconciling these numbers, at face value it appears there should be a comparable rate of such events in human. In the current presentation, it is not clear if the different methods target conceptually different scenarios, as they appear to aim at the same thing. Is there any reason the authors report lineage-specific events only for NHPs in the annotation section, but human-specific events only for toga?

L806: 'evolutionary periods of time': I suggest rephrasing this, the current statement is also true for non-congeneric species

L848: Panels d-f present variations of a similar plot that use the same color scheme, but the value ranges of panels d and f have been stretched by 2 or 3 times for different ranges, which seems visually misleading by exaggerating the differences in the former or compressing them in the latter. It is not clear which scale was used for panel d, nor what the additional effect of the various choices of range compressions are compared to a monotonic scale. A scale like those used in Fig. 8. b-c might be more appropriate.

L854: Please specify all tests used.

L905: This statement is certainly true for a comparison with WSSD, but part of the authors have additionally widely used alignment and assembly-based approaches to yield much higher estimates in prior releases. This seems like a more appropriate baseline, though both comparisons are absent. The methods state that calls with up to 70% of satellite content were kept, could they please explain the rationale behind this choice, as it seems this might not have been included in prior approaches, together with the impact it has on the numeric comparison across species (including the monkeys) given the substantial increase in resolution and variation reported in Fig. 2c for this sequence class. I do not doubt that the claimed increase is true regardless of the analytical setup, but it seems important to understand if it is in part also driven by choices driving a change in the author's definitions of SDs. The method section on the definition of lineage-specific SDs would benefit from distinguishing between orthologs and paralogs, given the ambiguity of the term homology in this context. What does "SDs with sequence content (coverage >20%) changed" refer to regarding query and target sequence? Is my understanding correct that a 300bp MEI of one species in a 1kb ancestral SD would result in classifying it as lineage-specific. If so, this seems to be quite different from other scenarios such as an expansion. I appreciate that the classification of these events is challenging, it would be helpful if the authors could provide summaries of the different scenarios they describe in Fig. SD.S1 across species and internal nodes. The annotations in the supplementary table are incomplete for SD status and lineage status and are missing the genotypes. Does SD diversity scale with SNV diversity, or do they behave differently?

L908: The definition is redundant, the Siamang is not a great ape.

L911, Supp. P137: The builds used for monkeys are not referenced, the referenced table SD2 does not exist. It seems the SD estimates will be largely conflated with the potentially much lower assembly quality for non-ape genomes. Unless the authors reference unpublished T2T builds, the correct conclusion is that the assemblies of great apes have the highest SD content.

L923, Fig. SD.S4: Assuming the r^2 value refers to an OLS linear model, it is not clear what criteria were used to conclude that the siamang or the unnamed ancestral nodes do not follow the correlation. Given the phylogenetic non-independence, the authors need to apply a phylogenetic comparative method to measure any deviation from the expected number of SDs in a lineage.

Referee #3

(Remarks to the Author)

The authors have generated T2T haplotype resolved assemblies for six ape species: chimpanzee, bonobo, gorilla, Bornean orangutan, Sumatran orangutan, and siamang. They resolve challenging regions, such as the major histocompatibility complex and immunoglobulin loci. Comparative analyses, including human enables them to investigate the evolution and diversity of regions previously uncharacterized or incompletely studied. These studies alleviate the previous reference bias from studies based on the human referenced genome. This includes lineage-specific segmental duplications, centromeric DNA, acrocentric chromosomes, and subterminal heterochromatin as well as regions under selection in the different species. These genomes should serve as a resource for studies of the human and ape relatives for all future studies.

The paper is sorted into three sections:

1. The genome assemblies
2. Highlights of analysis based on these assemblies
3. What is in positions in the genome previously not studied (ie assembled).

All sections are summarized in the main paper but have related supp info sections giving more information for each analysis. For the most part there is a good balance between the information given in the main text and the supp info. Both texts are overall well written.

For some of the analysis of segmental duplications, gene family expansion and regions under selection a bit more info could be given in the main text. If there are some examples of these events that can be validated experimentally that might strengthen the paper.

The results presented in the manuscript is novel and the six genomes will be resources valuable for a very long time.

In many places in the manuscript and supp info numbers are given with 4 significant digits. That is too much, please use two or three significant digits both in text and display items (for example extended data table 2). This should be fixed throughout the main and supp info text as well as in tables and figures .

Main text comments:

L 219 collapse -> collapsed sequence

L224). Overall, we estimate 99.2–99.9% of each genome is completely and accurately assembled, including heterochromatin. Define how this was calculated.

L239 Table 1. Define the numbers within parenthesis in columns six and seven

L255-257 “Schematic of the T2T siamang genome highlighting segmental duplications (Intra SDs; blue), inverted duplications (InvDup; green), centromeric, subterminal and interstitial α -satellites (red), and other satellites (pink).” Is the siamang genome compared to human here or to all of the other apes?

L269-270 We annotated over 18million base pairs for chromosome Y. What does “annotate” mean here?

L462. Including ORF genes means what?

L503 “The remaining 4% are discordant topologies that cluster within 200–500 kbp regions.” Are these in chunks?

L 620 We identified 14,210 AQER sites (Table AQER) across four primate lineages, including 3,268 on the human branch. So these are divergent sites, but how do you know if they are functional or just random changes happening on the different lineages? How sensitive to faulty alignments is this analysis? Having experimental support would be great but maybe not trivial.

L672 New genomic regions – maybe say “newly identified”, “newly examined” or “newly assayed” regions

L815 “over <2mya” is confusing. Please explain/rephrase

L841panel a – should it be marked that the figure is based on HSA1, HSA2 etc (not just labelled chrom 1)

L859 In total, these account for 1.05 Gbp of subterminal satellite sequences (642 Mbp or 18.2% of the siamang genome). How to these numbers match with the total genome size?

Comments on supplemental notes:

Page 9 Read mapping “The X was included to the paternal and Y to the maternal haploid genome.” What does this mean? Why?

Page 23: “Autosome SNV divergence between human and NHPs was lowest for human-chimpanzee and human-bonobo (0.15-0.16%), then human-gorilla (0.19-0.20%) and lastly human-orangutan (0.36%)”. Which orangutan?

Page 39: “We used the same mutation rate of TRAILS analysis and the following generation times:” Where/how did you get generation times?

Page 46: “The effective population time through time is shown in Fig. ILS.S5” Do you mean population size?

Page 49: “This gene undergoes copious expansion across chromosomes, all of these falling in lineage-specific SD regions” Any biological interpretation if this does something or is just a random event? If it does something maybe add to main text.

Page 51: “Then for a given gene in T2T, either protein-coding (Fig. Gene.S3f) or member of a multicopy gene family (Fig. Gene.S3g), we find...” Unclear what the difference between protein-coding and gene family is.

Page 59: Figure Species-Specific MEI Classifications. Color should be red not magenta in label

Page 115: “For primate lineages (i.e., PTR, PAB, GGO, PAB, PPY, SSY), sequences that were not aligned or aligned of identity.” PAB listed twice, incorrect?

Page 135: "In addition to the subterminal satellites, spacer SDs interrupting the satellites were investigated were investigated." repeated were investigated

Page 136: Add more info to figure legends for both figures

Version 1:

Reviewer comments:

Referee #1

(Remarks to the Author)

I find that the manuscript has improved immensely by careful rewriting during the revision. It now reads very well, and it is very clear that it contains a large amount of novel findings. My comments and suggestions have all been addressed satisfactorily, with several additional analyses. I appreciate the carefully crafted supplement and the detailed way the genomic resources have already been made available. I see no reason for not immediately adopting these new genomes as the reference genomes.

The paper represents a monumental effort. I believe it will have an immediate impact on the field and a future impact once we understand the many new insights provided into acrocentric chromosomes, subtelomeric repeats, centromere evolution, and duplications better.

Thus, I have no further comments.

Referee #2

(Remarks to the Author)

I thank the authors for the effort they took to answer my questions concerns. The restructured manuscript they provide in this revision has greatly improved in clarity and enables the reader to much better appreciate the importance of the presented work via the stronger focus on analytical aspects in which the T2T assemblies are able to truly shine. While this has unfortunately required to move a large portion of the analyses to the supplement, I appreciate that this has not resulted in a lack of detail in the authors responses to comments on those sections.

While most of my concerns were addressed, I have remaining questions on the AQER, together with some minor in addition comments.

The authors have greatly expanded the details of their method to defined AQERS, which are now clearly explained. They also provided additional analyses to address some of my previous concerns, such as accounting for differences in branch-lengths, most of which are sensible. However, while the authors aimed to provide clarity on the impact of local variation in mutation rates on the expected number of changes by using repeats and regions with ILS as proxies, this does not address concerns raised by the lack of consideration of a windows' sequence composition when establishing the expected number of substitutions. It seems the counts required to establish an AQER are agnostic to the underlying base-composition, which is surprising given the substantially variation in frequencies at which different mutation types occur, resulting in different expectations for e.g. GC-rich vs GC-poor windows. A single threshold that averages these differences is thus likely inadequate to set an expectation of what should be consider "quickly evolving" and will potentially lead to GC-content driving the resulting AQERs, while also undercalling regions at the other end of the composition spectrum despite exhibiting an excess of substitutions. This would be consistent with e.g. their observed enrichment in (generally CpG-rich) promotor regions. The authors reference additional confounding factors, such as the non-independence between mutation rates and selection, or selective pressures in regions with high mutation rates that have been deprioritized in other studies. While accounting for all confounders is probably not feasible anyway, it is not evident why this should justify not accounting for any confounder. Doing could similarly deprioritize certain regions, but still make the calls subject to the pitfalls the authors bring up, and. Additionally, it is neither evident why the authors chose to base the thresholds on substitution rates estimated from 10mb windows, a size at which the dispersion of expected number of changes and GC-content is likely to substantially differ from those at the 500bp mainly used to define AQERs, nor what influence this cutoff has on the calls.

While the approach described in the methods is ultimately consistent with the authors' subsequent definition of AQERs as the "most divergent regions of the human genome, agnostic to the underlying force, or forces, of divergence", their current naming suggests they evolve more quickly than expected, which I believe is currently incompletely accounted for. Should the authors therefore choose to not consider nucleotide composition, the potentially resulting biases need to be explicitly communicated somewhere.

Additional questions on AQERs:

- The calls for larger window sizes use a multiplier of the same 10Mb substitution rate cutoffs, which may result in differences in sensitivity across window-sizes. Given the dependency of the observed enrichments on AQER window sizes, this might warrant consideration.

- I appreciate the comparison to hg38 AQERs, but am surprised by the low intersection of calls between assemblies. These differences attributed to different haplotypes in either assembly and newly resolved sequence, but it is unclear how this conclusion was reached, particularly for the T2T calls. Are the distributions of mutation counts comparable for the different sets?

Additional questions:

L134: Please drop the "all" or tone down the sentence in some other way.

L231, Figure PanGenome.S1D/S2: I appreciate the authors response to my previous comments by establishing a more appropriate background model. Could they please clarify what might cause the increase counts around scores of ~.3 in S2. The (averaged?) values of different classes are -at least to me- not necessarily expected as claimed. Several putatively

functional sequence classes appear to be less constrained than the presumably neutrally evolving background in S2, which I suspect is the result of a lack of power over short branch-lengths of apes.

L280: Please rephrase “biomedical genes” and provide an example of the biomedical relevance for a general audience. Please clarify what tissue comparisons the claim of tissue-specificity is based on in the supplement.

L460-475/throughout: Please specify the test statistic for all p-values and other coefficients throughout the main text.

Supp. P52: Please provide definitions of “standard quality control” and the specific filters applied to SNVs and indels.

Referee #3

(Remarks to the Author)

Overall the authors have done a good job at addressing the reviewers comment. I have only two smaller comments.

Line 300 We now estimate that the autosomes of the ape genomes contain 53.21–57.99% detectable repeats,... Still has four significant digits. Please fix throughout paragraph/manuscript

Line 325 L1s with intact ORFs (more than 500 in both orangutan species compared to 203 in gorilla). Humans and gorillas fall in between this spectrum. “Gorilla mentioned twice is that correct?”

Version 2:

Reviewer comments:

Referee #2

(Remarks to the Author)

I thank the authors for their diligent efforts in responding to my remaining comments, all of which have been thoroughly addressed. I greatly appreciate the enormous amount of work that went into this manuscript, and the invaluable resources the authors provide to the community therein. Congratulations, and thank you!

Referee #1 (Remarks to the Author):

In this manuscript, (virtually) complete genomes from 6 apes (chimpanzee, bonobo, gorilla, two orangs and siamang) are presented and analysed. It represents an incredible amount of analysis and results, and many new findings are interspersed with confirmation or smaller increments of previous findings. How to find out which is which was a real challenge.

General comments

First, I will applaud the quality of the genomes and, not least, the very careful annotation. These genomes should immediately replace previous reference genomes of these species, and they will be a very valuable resource for other researchers, who will also be able to exploit this data much more than is possible, even in this very long manuscript. The supplement is long but nicely organised into sections authored by different subgroups of authors, and they can be read independently. Most of the main points of the supplement are referred to in the main text. The assembly strategy etc is also well laid out.

The main text is close to 14000 words with nine densely packed, multipanel Figures. I think the manuscript suffers severely from information overload; it reads like eight incomplete papers in one. Many Figure panels are very complex, yet they are not always referred to in the text, and when they are, it is often very cursory. Thus, figures often need to be understood from the short legends alone, and it is left a bit to the reader to make her/his interpretations and explain her/his own reasons for why they were included. The sheer amount of information makes it difficult for this reviewer to check whether individual claims are well supported, not least because inferences are often qualitative from the Figures rather than quantitative and supported by statistics. It would be nice if there were less of such claims throughout.

We appreciate the detailed and constructive comments from the reviewer. We significantly revised the manuscript reducing the detail (information overload) and simplifying panels—especially those that are not called out in the main and moving them to the supplement or creating new extended display items as suggested. We opted to retain for main display the more quantitative versus qualitative information and kept an eye to remove claims that were not as strongly supported.

Many analyses are very descriptive and unrelated to any hypothesis or prior expectation. While this is very understandable for this kind of data, it would still be very useful if, for each new pattern analysed, there is first a clear motivation, analysis and then a clear conclusion (which we sometimes have to wait for the discussion to get).

We have introduced motivations for some of the major shifts in the descriptions followed by a sentence where appropriate regarding the conclusion to improve flow and clarity.

Thus, while I believe analyses are generally very competently carried out, the manuscript could be much more impactful if many analyses were removed and many Figures were simplified. These are the analyses that repeat (with better genomes) previous findings with marginal gains such as sweep finding, ILS, and structural variation findings (see below for details). In between are very interesting results that almost “drown” and they would benefit from the addition of more detail and interpretation. It is difficult to see how the findings relate to other recent studies by some of the same authors, for example, the human T2T papers and the recent SD paper in Cell. In the cell paper, female individuals are sequenced with long reads (not quite as complete, though), but there is no comparison with the findings of this paper, and there is no joint analyses. That could perhaps help delineate how much variation in different types of repetitiveness (and structural variants) is found within species as compared to between. This is done in some cases here when the two haplotypes can be compared but it is not done consistently and this makes it hard to evaluate which patterns are truly divergent between species. A short discussion on this would be great and could perhaps replace some of the summaries of results that are now included in the discussion. Also, what differences are found on the autosomes and the sex chromosomes, which were recently published in Nature in a paper much easier to read than the present manuscript

We significantly restructured the text either reducing or moving to the supplement sections that the referee flagged below as “marginal” or “incremental.” Per the referee’s suggestion, we performed a more direct analysis with the additional individuals sequenced as part of Mao et al., Cell, 2024¹. As noted, those genomes are less complete and do not resolve centromeres, acrocentric DNA, or subterminal heterochromatic caps so they cannot be used for any genome-wide analysis involving these previously inaccessible regions, which are a focus of the paper. Those genomes, however, do provide the opportunity to comment more directly on fixed lineage-specific differences. We, thus, focus on a comparison of MEIs as well as an examination of the coalescence of the MHC locus using these additional genomes (see below). We add some of these findings to the discussion. We also describe the comparison with sex chromosomes and autosomes as follows:

“We now estimate that the autosomes of the ape genomes contain 53.21–57.99% detectable repeats, which include transposable elements (TEs), various classes of satellite DNA, variable number tandem repeats (VNTRs), and other repeats (**Fig. Repeat.S1**), significantly fewer than the sex chromosomes (X [61.79–66.31%] and Y [71.14–85.94%])¹¹. Gorilla, chimpanzee, bonobo, and siamang genomes show substantially higher satellite content driven in large part by the accumulation of subterminal heterochromatin through lineage-specific satellite and VNTR expansions (**Fig. 1, Fig. Repeat.S1, Extended data Table 1**).”

Additionally, as we state in the paper:

“Compared to the data reported for sex chromosomes (8.6 and 7.4 Mbp), autosomal subterminal caps (7.3 and 7.7 Mbp) are comparable in length for gorilla and siamang, respectively. In contrast, for chimpanzee and bonobo, both the distribution and length of the heterochromatic caps differs significantly between autosomes and sex chromosomes. For example, we previously detected pCht subterminal arrays on the p-arm of chimpanzee and q-arm of bonobo chrX, but in autosomes, we find that the caps are much less

prevalent in q-arms. In addition, within the *Pan* lineage the average length of the subterminal repeat arrays are overall much longer on the autosomes (5.3 and 6.6 Mbp) when compared to the sex chromosomes (0.66 and 2.2 Mbp) for both chimpanzee and bonobo, respectively.”

“We also find NORs on both orangutan and siamang Y chromosomes¹¹, and partial DJ fragments on the chimpanzee and bonobo chrY (Fig. 5b, Fig. Acro.S2), suggesting their ancestral chrY may have been NOR+.”

We find a comparable number of SD bases in autosomes compared to sex chromosomes (Fig. SD.S2).

Fig. SD.S2. Comparison of SD content in sex chromosomes and autosomes.

I find that the most novel and interesting results concern what has been very inaccessible until now. These results could be discussed more in light of previous research and more care could be made to illustrate specific points of greatest novelty.

We thank the referee for providing specific suggestions as to what and what not to emphasize. In general, we dedicate significantly more text to the inaccessible and reduce the more “incremental” resource section and attempt to highlight the novelty in the context of previous research.

They are

MHC: two full haplotypes per species are impressive, but mining the Cell paper data could add two more for each species. Then, a more detailed evolutionary analysis of dn/ds, the divergence between haplotypes and coalescent times of pairs of alleles could be estimated. A more detailed TRAILS analysis could be made on the alignable fractions as well. How do the results relate to the large knowledge about human MHC/HLA, including known GWAS findings?

Based on the referee's suggestion, we examined the assembled contigs for MHC in the additional ape genomes¹ (**Fig. MHC.S9**) and found that the corresponding ~4.5-5.7 Mbp was complete, essentially doubling the number of haplotypes for four species: chimpanzee, bonobo, Western gorilla, and Sumatran orangutan. As a result, we performed a phylogenetic analysis with these additional genomes (**Fig. MHC.S9**). Expanding the MHC analysis with the additional ape genomes, we observe deep coalescence (i.e., topologies inconsistent with the species tree), similar to the results based on T2T genomes for the four regions that we originally highlighted (**Fig. MHC.S10**). In general, the new haplotypes, however, do cluster by species but the within-species divergence time were very variable depending on the genomic regions. For example, variable regions I, II and IV among bonobo were <1 mya but that of region III showed a divergence of 14.00 mya. Also, the within-species divergence times were found to be 7.17 and 11.61 mya for chimpanzee haplotypes, for region II and IV, in contrast to <1 mya for regions I and III. These data suggest that much deeper sampling of many more haplotypes will be required across the apes to understand the full extent of genetic diversity.

We also revisited the TRAILS analysis. We find that 100, 96, 51 and 99.9% of the narrow variable regions I, II, III and IV were consistently called out by TRAILS, respectively (1 kbp bins with >50% of ILS fraction).

We provided the posterior decoding of the HLA region in Figure ILS.S4 and found that there are genomic regions departing from the modal phylogeny. We also focused on the four variable HLA regions considering the tree topology and the coalescent level. We added the following paragraph:

“Moreover, we focused on the four HLA variable regions at a base-pair scale considering the tree topology and the first coalescent event (**Fig. ILS.S5**). Three of the variable regions (I, II, and III) show elevated levels of ILS, with the only exception of region IV, confirming their high variability. For region I, 88% of the analyzed sequence was compatible with a topology in which human and gorilla are sister groups, and the first coalescent is shallow, consistent with the reconstructed phylogeny in **Extended Fig. 3c** and **Fig. MHC.S10**, suggesting a possible balancing selection event. In region II, 36% of the sequence is consistent with an HC topology, with the remaining regions equally distributed. In the remaining two regions, we observed 55% (region III) and 71% (region IV) of the region topology consistent with HCGO.”

Figure ILS.S5. Treemaps of the topology inferred for the four HLA variable regions. The topology is indicated in the middle of each rectangle. We show four possible topologies: human-chimp (HG recent) with shallow first coalescent, human-chimp with deep first coalescent (HG ancient), chimp-gorilla (CG) and human-gorilla (HG). Labels on the right corner refer to the first coalescent deepness.

Figure MHC.S9. Alignment of MHC regions with the addition of genomes from previous studies¹¹⁵.

Figure MHC.S10. The tree topology and divergence time estimates of the narrow variable regions (I, II, III and IV) detected in **Extended Fig. 3c**, after addition of genomes¹¹⁵.

Intersecting with GWAS catalog (v1.0-e110_r2023-09-25²), we find the four regions intersect with 253 SNPs associated with 238 traits including COVID-19 severity, asthma, autoimmune neurological syndromes, HIV, tuberculosis, etc., suggesting the putative function of the variable sequence that exists in the MHC region.

We also added the following to the main:

“We estimate coalescence times of these regions to range from 10–24 mya **between species, and <1-14 mya within species (Extended Fig. 3c; Fig. MHC.S10).**”

Furthermore, we annotated the MHC genes in these additional eight haplotypes (new **Table MHC.S6-9**). They confirm the variation in gene organization already observed in the original 12 haplotypes. We added the following to the main text:

“Annotation of additional MHC haplotypes from previously published genomes of chimpanzee, bonobo, gorilla and Sumatran orangutan⁴¹ revealed comparable structural variation among the MHC genes (**Table MHC.S6-9**).”

NOR+/acrocentric chromosomes: To me this is the highlight of the manuscript. The dynamics of rDNA evolution is very impressive and can also be very important for both some syndromes (in humans) and for reproductive isolation as suggested. I would like more details on this including interindividual variations and a comparison to previous results such as the T2T human genomes

We thank the referee for these thoughtful comments and appreciate the referee's interest in this part of the genome. Because these regions of the genome are still difficult to assemble and annotate, their characterization in each genome is a painstaking labor of love. Routine sequence and assembly of this portion of the acrocentric genome remains the last frontier with respect to automated assembly. We added supplemental figures to demonstrate the variability in satellite content, as well as rDNA structure and variation. Indicating the satellite content for each chromosome and haplotype individually demonstrates the more fine-grained intraindividual variability that exists between haplotypes and chromosomes, complementing the main figure that shows this data at the aggregate species level. Using representative rDNA units for each species, we demonstrated interspecies variability by creating and visualizing a multiple sequence alignment between these units, showing the conservation of the coding regions relative to the intergenic spacers. Aligning all species-specific rDNA HiFi reads to each of these units demonstrates the intraindividual variation in the rDNA units that we are unable to assemble. Again, the genes look relatively conserved compared to the spacer region, where more structural and small variation exists. We notice differences in the amount of variation that exists in each species, which is likely explained by the fact that rDNA units in the same array tend to appear more similar than units in different arrays. This is a result of the increased prevalence of unequal crossing over between sister chromatids homogenizing units within an array compared to gene conversions or crossovers homogenizing units between arrays. As a result, gorilla appears to have much less rDNA variation than orangutan given it has four total arrays rather than 19.

In addition, we developed methods to estimate the copy number and several companion pieces are in preparation, with one posted to bioRxiv that examines the activity of individual arrays across humans and nonhuman primates³. The bioRxiv study documents tremendous variation in copy number, sequence, and NOR activity across a panel of human individuals, including the T2T-CHM13 and HG002 genomes. The bioRxiv study reports haplotype-specific NOR activity in the ape genomes. Based on the haplotype variation in apes shown in this study, and the variation observed across multiple humans, we speculate chromosome-unique patterns of copy number, sequence, and gene activity exist in nonhuman primates for the rRNA genes, but also for the copy number of the satellite DNA in the region. Among the apes, copy number of rRNA genes ranges from 1-287 copies per chromosome, depicted in the heatmap in **Figure 5b** and **Fig. Acro S2**, and differences in satellite DNA between the haplotypes are apparent in **Fig. Acro S3**. These ranges are similar to the interindividual ranges reported in our analysis of 10 human genomes where we estimated a range from 33-260 rDNA copies per chromosome.

We added some of these details regarding interindividual variation to the text as follows:

“Estimated rDNA copy number for ape arrays varies from 1 on chrY of Bornean orangutan to 287 on HSA21 of chimpanzee; total diploid rDNA copy number similarly varies from 343 in siamang to 1,142 in chimpanzee (Methods, **Fig. 5b**, **Fig. Acro.S2**, **Table Acro.S1**), with total rDNA copy number varying widely between individual NOR+ haplotypes of the same species, as expected, **but within the range of human variation. For example, a recent analysis of 10 human genomes estimated total rDNA copy number from 532 to 683 with individual chromosomes showing more than 10-fold variation (e.g., 22 to 260 rDNA copies on chromosome 14)**⁷⁸. Heterozygous NOR loss was observed in bonobo (HSA21)

and Sumatran orangutan (HSA13), **both** of which were mediated by a truncation of the chromosome prior to the typical NOR location (**Fig. 5b, Fig. Acro.S2**). The structure and composition of both satellites and SDs varies considerably among the apes, **as well as between chromosomes and haplotypes within individuals** (**Fig. 5a, c. Fig. Acro.S3**). The orangutan acrocentrics are dominated by HSat3 and α -satellite, compared to the more balanced satellite composition of the other apes. Gorilla is notable for the presence of double NORs on both haplotypes of HSA22, with the additional NORs inverted relative to the first and including a complete DJ but only a single, inactive rDNA unit (**Fig. Acro.S4**).

...

We extracted representative rDNA units from each assembly to serve as a reference for each species and confirmed a similar sequence structure, including the presence of a central microsatellite region within the intergenic spacer sequence for all species (**Fig. Acro.S5**), but with relatively high nucleotide substitution rates outside of the >99% identical 18S and 5.8S coding regions (**Fig. Acro.S6**). **Noticeably, certain species contain more variation in the rDNA unit than others, which is explained by the fact that units in the same array tend to appear more similar than units in different arrays, and species have differing numbers of NORs** (**Fig. 5b, Fig. Acro.S7**). This is likely due to the increased prevalence of **unequal crossing over homogenizing units within an array compared to gene conversions or crossovers homogenizing units between arrays**^{79,80}.

...

The DJ sequence was found to be conserved across all great apes and present as a single copy per NOR, including the palindromic structure typical of the human DJ, with the exception of siamang, which contains only one half of the palindrome on each haplotype but in opposite orientations **on both haplotypes of chr21** (**Fig. Acro.S8**).”

Updated figures: (numbers below images)

Figure 5. ... Sum of satellite and rDNA sequence across all short arms where one haplotype is NOR+ in each species. “Unlabeled” indicates sequences without a satellite annotation, which mostly comprise SDs. ...

Figure Acro.S3. Satellite content of all short arms. Sum of satellite and rDNA sequence across each short arm where one haplotype is NOR+ in each species. “Unlabeled” indicates sequences without a satellite annotation, which mostly comprise SDs. Total SD bases are given for comparison, with some overlap between regions annotated as SDs and satellites. Total satellite content on haplotypes of the same chromosome can vary as much as 22.5 Mbp (PPA HSA18), or as little as 100 kbp (PPY HSA21). Of the two truncated short arms, one appears much shorter than the other (PAB HSA13, 1.2 Mbp vs. PPA HSA21, 7.5 Mbp) mainly due to large HSAT1A expansions on the bonobo truncated HSA21.

Fig. Acro.S3 becomes Fig. Acro.S4.

Fig. Acro.S4 becomes Fig. Acro.S5.

Figure Acro.S6. Interspecies variation within the rDNA. Multiple sequence alignment of the representative rDNA units for each species generated with MAFFT shows high conservation within the genic regions (particularly the 18S and 5.8S) relative to the intergenic spacer, which shows significant structural variation.

Figure Acro.S7. Intraspecies variation within the rDNA. rDNA-specific HiFi reads were extracted for each species and aligned to the representative rDNA unit. Again, the genic regions, particularly the 18S and 5.8S, show higher conservation relative to the intergenic spacer. Certain SNVs and SVs appear relatively common in the population of rDNA units, as does variation in the size of CT-microsatellites.

Fig. Acro.S5 becomes Fig. Acro.S8.

Centromeres: Also very intriguing with same comments as for NOR+. More about evolutionary rates and species polymorphisms would be very valuable as would comparisons to human T2T results. Can more be said about the bonobo short centromeres for some chromosomes. Is that also found in other mammals/primates?

We compare differences in α -satellite HOR array lengths between haplotypes from the same species and congeneric species (**Fig. CEN.S2**) and find that four out of five species (bonobo, chimpanzee, and both orangutans) have α -satellite HOR arrays lengths that are similar between haplotypes (mean difference ~ 1), while gorilla has a greater difference in α -satellite HOR array length (mean ~ 1.37 ; **Fig. CEN.S2a**). This suggests that gorilla centromeres may be undergoing more rapid evolution contributing to greater rates of intraspecies centromeric variation. However, we will need to confirm this observation with additional sequencing of other individuals from these species, which is currently underway.

Figure CEN.S2. Haplotypic and cross-species variation in centromeric α -satellite HOR array lengths. a) Fold difference in α -satellite HOR array lengths between haplotypes for each nonhuman primate. b) Differences in α -satellite HOR array lengths for each chromosome among human and five nonhuman primates. Chromosomes are named according to HSA nomenclature to permit cross-species comparisons, and the lengths of the α -satellite HOR arrays are shown on a log scale.

While our sampling of other nonhuman primates is limited, we have sequenced the genomes of owl monkey and macaque. Our preliminary assessment of these centromeres indicates that they are, on average, larger than those presented in this study and some are composed of a mixture of different repeat sequences (unlike the pure α -satellite HOR arrays present in the great apes). While we have not yet found a primate species that has half of their centromeres that are less than 700 kbp (like the bonobo), we do see evidence of smaller centromeres in some species. For example, in human, we consistently observe smaller α -satellite HOR array lengths for the Y chromosome (mean \pm SD: 664 \pm 375 kbp, $n=16^4$). In this study, we also observed small α -satellite HOR arrays in gorilla chromosome 12 (HSA chromosome 2B), which are 104 and 106 kbp long for each haplotype. We also observed smaller arrays on Sumatran

orangutan chromosome 10 (HSA chromosome 12) that are 116 and 163 kbp long for each haplotype (**Fig. CEN.S2b**). A recent report of the Eastern Hoolock gibbon also suggests a small chromosome X centromere that spans ~112 kbp⁵. Thus, while an abundance of small centromeres in a single primate species is relatively uncommon, we do see evidence of functional smaller centromeres in other human and nonhuman primate species. **No primate species, however, shows the distribution of small α -satellite HOR arrays as observed for the bonobo.**

Subterminal repeats: This is also very interesting. I know little about these but they were noticed also on the X in the recent X/Y primate paper. Could more be speculated on their function and could a hypothesis be formulated for future research?

The average length of subterminal heterochromatic caps is very similar between autosomes (7.3 and 7.7 Mbp) and the X chromosomes (8.6 and 7.4 Mbp) for gorilla and siamang, respectively. In contrast, for chimpanzee and bonobo, both the distribution and length of the heterochromatic caps differ significantly between autosomes and sex chromosomes. For example, we previously detected pCht subterminal arrays on the p-arm of chimpanzee and q-arm of bonobo chrX, but in autosomes, we find that the caps are much less prevalent in q-arms. In addition, within the Pan lineage the average length of the subterminal repeat arrays are overall much longer on the autosomes (5.3 and 6.6 Mbp) when compared to the sex chromosomes (0.66 and 2.2 Mbp) for both chimpanzee and bonobo, respectively.

With respect to the function or consequences of subterminal satellites, there is evidence⁶ in chimpanzee and gorilla that the subterminal caps are late replicating. Integrating replication-timing data, we confirm that the subterminal caps in chimpanzee and gorilla are very late replicating and report a significant difference in terms of replication for their chromosomes with and without subterminal heterochromatic caps (see below; **Fig. RT rebuttal 1**; **Fig. RT.S3-4**). There are also reports that in gorilla telomeric RNA accumulates extrachromosomally in mitotic cells and that telomere and subterminal satellite repeats may be co-transcribed. Hirai and Koga studied C-terminal heterochromatic subterminal associations and found evidence of ectopic exchanges and “post-bouquet” structure formation in sperm during pachytene⁷. These exchanges may serve to protect variation proximally while simultaneously promoting concerted evolution among nonhomologous chromosomes.

We added the late replication finding to the results and speculate further on their potential functional role in the discussion:

“It is tempting to speculate that the subterminal hypomethylation pocket may represent a site of protein binding or a “punctuation” mark perhaps facilitating ectopic exchange and concerted evolution¹¹⁰. While the function of these regions is unknown, we confirm earlier observations that subterminal heterochromatic caps are very late replicating (**Fig. RT.S3-4**). Hirai and Koga studied C-terminal heterochromatic subterminal associations and found evidence of ectopic exchanges and “post-bouquet” structure formation in sperm during pachytene stage of meiosis¹¹⁰. Late replication may help promote such exchange events buffering the proximal unique sequence while simultaneously promoting concerted evolution among nonhomologous chromosomes.”

Sections on segmental duplications are very interesting. However, I lack considerations of which are polymorphic within species and which have shared polymorphisms between closely related species such as the two oranges and the chimp and bonobo.

We performed two analyses to address this query. First, we compared the two haplotypes to determine homozygous (potentially fixed) versus heterozygous (potentially polymorphic) within each species; note in humans, we used three T2T or near-T2T genomes for these comparisons (T2T-CHM13 and h1/2 of HG002) as opposed to two. We also assessed the variability of SDs across closely related species in the *Pan* and *Pongo* genera via four-way comparisons. We focused on the autosomes excluding the sex chromosomes (male samples with X&Y) and the

acrocentric portions of the short arms where synteny and heterozygosity are challenging to assess because of ectopic exchange events (**Fig. SD rebuttal. 1**).

Figure SD rebuttal. 1. SDs analyzed for polymorphism. We analyzed 145.4 Mbp of SDs per haplotype after filtering chromosomes X&Y and the acrocentric SDs (blue).

Using the remaining sequences, we considered four different categories (**Fig. SD.S7a**): 1) fixed-homozygous for the SD sequence, 2) heterozygous or haplotype-specific, where there is no corresponding SD in alternative haplotype, 3) variable length, where the two haplotypes differ in length more than 10%, and 4) highly variable where the two haplotypes differ by at least twofold across the SD. Overall, we find that close to or over two-thirds of the SDs are fixed in African great apes as opposed to one-half in Asian apes (**Fig. SD7a**). Predictably, the proportion of variable SDs is greater for lineage-specific SDs (by on average 16%), which are expected to be of more recent origin, except for siamang that have relatively longer divergence to the most recent common ancestor (19.5 mya) (**Fig. SD.S7b**). In the four-way comparisons in the *Pan* and *Pongo* genera, we identified the same variable SD categories treating the four haplotypes as the same species. We find that the largest fraction of the SDs corresponds to a highly variable length category (**Fig. SD.S7c**). As expected, combining the three variable SD categories, we observe 85-92% of total variability, which far exceeds within-species inter-haplotype variability of 30-50%.

Figure SD.S7. Within-species variability of SDs observed between two haplotypes.

a) Classification of “fixed”, “haplotype-specific”, “highly variable length”, and “variable length” SDs with respect to the alternative haplotype. b) Average inter-haplotype variability within each species observed for the lineage-specific SDs. Within species-specific SDs that correspond under the *Pan* and *Pongo* genera are summarized separately due to relatively short divergence time between the two. c) Four-way comparison among haplotypes of the *Pan* and *Pongo* lineages to quantify SD variability between haplotypes of less diverged species.

As a second approach, we estimated the copy number of each of the four categories using Illumina WGS obtained from 10 individuals⁸ from each species. Briefly, SD copy number was estimated by mapping short-read data using fastCN⁹ back against T2T ape genomes. For each SD category, we estimated the copy number and its associated standard deviation focusing on the genes mapping to each SD. Predictably, those SDs associated with variable SD showed significantly greater variability in copy number than those classified as fixed SDs (Fig. SD.S8), confirming that variable SDs found in our data are also variable in the population.

Figure SD.S8. Distribution of $\log(\text{standard deviation} + 1)$ of copy number across 10 additional individuals of gorilla, Sumatran orangutan, bonobo and chimpanzee for the four categories of variable status. The standard deviation of copy number found for SD genes flagged as fixed between two haplotypes are significantly smaller (two-tailed Wilcox ranked sum test, indicated on the top ***: $p < 0.001$ ****: $p < 0.0001$) than variable SDs.

We added descriptions as follows:

“In general, the number of SDs assigned to different lineages correlates with branch length ($r^2=0.927$; **Fig. SD.S6**) with the exception of siamang and some ancestral nodes reflecting the great ape expansion of SDs⁹⁵. We further assessed the variability of SDs within species by comparing two haplotypes and find that 30-37% are variable or polymorphic with the exception of gorilla, which shows reduced variability (26.5%). For Sumatran orangutan and siamang, we find that 42-50% are predicted to be variable based on the comparison of the two haplotypes (**Fig. SD.S7-8**). A much larger survey of individuals from each species, however, will need to be assessed to more accurately classify fixed versus polymorphic SDs in each species.”

We also added descriptions of the method used in the supplementary notes as follows:

“Homozygous and heterozygous genotypes were determined by comparing the two haplotypes. Polymorphic SDs within congeneric species were also investigated by making four-way comparisons for *Pan* and *Pongo* lineages. To check if the copy number of genes located in polymorphic SDs are also variable in additional ape genomes, we utilized Illumina short-read whole-genome sequencing data² processed by a previous study¹¹⁵.”

Any sign of adaptive introgression between species for MHC as could be expected due to balancing selection?

Distinguishing adaptive introgression from trans-species polymorphism is very challenging for the MHC region, which we have shown here have a remarkably deep coalescent and are subject to balancing selection for multiple alleles^{10,11}. Given the complex and deep phylogeny, we believe a much larger sampling of haplotypes at the population level will be required.

The transposable element section is also interesting but again, I would like a consideration of the number of intraspecies polymorphisms as a way to measure recent transpositions more directly.

Building upon our CACTUS unaligned species-specific mobile element insertion (MEI) dataset, for which we provided allelic information (homozygous vs. heterozygous; **Fig. Repeat rebuttal 1**), we attempted to distinguish polymorphic from fixed events using the Mao et al. genomes. Through genotyping the species-specific MEIs in additional genomes, we find that the fraction of polymorphism increases by 2.1-9.4% in African great apes. The genotyping in Sumatran orangutan identified a 20% increase in polymorphic MEIs, which is also in line with the trend we observe by comparing just the two haplotypes.

Table Repeat.S14. Summary of genotyping of species-specific MEIs in additional genomes.

Lineage-specific MEIs		# of MEIs in T2T (two genomes; only mappable ones)				# of MEIs by consensus (four genomes)		
Reference	MEI_type	Total	Hom	Het	Fraction-Poly	Fixed	Polymorphic	Fraction-Poly
PPA.T2T	LINE/L1	1,140	804	336	29.5%	778	362	31.8%
PPA.T2T	Retroposon/ SVA	715	432	283	39.6%	416	299	41.8%
PPA.T2T	SINE/Alu	2,073	1,708	365	17.6%	1,664	409	19.7%
PTR.T2T	LINE/L1	1,236	921	315	25.5%	805	431	34.9%
PTR.T2T	Retroposon/ SVA	644	434	210	32.6%	385	259	40.2%
PTR.T2T	SINE/Alu	1,927	1,584	343	17.8%	1,429	498	25.8%
GGO.T2T	LINE/L1	3,698	2,663	1,035	28.0%	2,461	1,237	33.5%
GGO.T2T	Retroposon/ SVA	1,975	1,375	600	30.4%	1,241	734	37.2%
GGO.T2T	SINE/Alu	5,950	4,838	1,112	18.7%	4,323	1,627	27.3%
PAB.T2T	LINE/L1	3,123	934	2,189	70.1%	332	2,791	89.4%
PAB.T2T	Retroposon/ SVA	403	110	293	72.7%	31	372	92.3%
PAB.T2T	SINE/Alu	85	33	52	61.2%	15	70	82.4%
Total		15,234	9,953	5,281	34.7%	8,403	6,831	44.8%

We also added “**Table Repeat.S15. Full table of genotypes identified for the mappable lineage-specific MEIs in additional genomes**” to the supplementary tables.

Table Repeat.S13. becomes Table Repeat.S16.

As the CACTUS unaligned callsets were composed of species-specific MEIs called from only one haplotype (primary), we performed a separate analysis utilizing the phased assembly

variant (PAV)-generated callset with human (T2T-CHM13) as the reference to identify polymorphic MEIs from both haplotypes (haplotype 1 and haplotype 2) for each primate species with the exception of human. We screened structural variants identified by PAV for the presence of MEIs using the pipeline developed for the CACTUS unaligned structural variant dataset to call species-specific MEIs. All MEIs located on sex chromosomes were removed from the following analyses as these insertions cannot be easily classified as polymorphic. Inclusion of the second haplotype resulted in an additional 2,628 heterozygous elements across the five species (PPA, PTR, GGO, PAB, and PPY).

Due to the relatively recent radiation of bonobo and common chimpanzee (~1.6 mya) and divergence of the two orangutan species (1 mya), it is conceivable that some lineage-specific insertions are polymorphic within these species. To account for that, we screened MEIs specific to the chimpanzee lineage as well as MEIs specific to the orangutan lineage for evidence of polymorphic MEIs. Toward this, we utilized the genotyping information provided by PAV. In addition, we determined genotypes using the previously described BLAT-based pipeline. Heterozygous or homozygous designations were only made when both pipelines were in agreement. Comparing MEIs per millions of years between the PAV and CACTUS unaligned callsets showed good congruence for elements deemed homozygous (**Fig. Repeat rebuttal 1a**). As expected, we observe a greater total number of heterozygous MEIs by including the second haplotype assembly (**Fig. Repeat rebuttal 1a**).

Figure Repeat rebuttal 1. a) A scatterplot comparing the total number of MEIs per millions of years for each active non-LTR retrotransposon family (Alu, L1, SVA) in each species (PTR, PPA, GGO, PAB, and PPY) between the PAV species-specific MEI callset (Hap1 and Hap2) and the CACTUS unaligned species-specific MEI callset (Hap1 only). Orange and blue dots represent the total number of MEIs per millions of years that are homozygous or heterozygous, respectively, in the genome. b) Barplots showing the total number (non-normalized) of intraspecies polymorphic MEIs (species-specific + lineage-specific) within the PAV callsets. c) Stacked barplots showing the proportion of the total number of all full-length (≥ 5900 bp) with intact open reading frames (ORF1 and ORF2) L1s within each species (entire PAV Structural Variation Callset) that are either polymorphic (blue; species-specific + lineage-specific) or non-polymorphic (gray). All elements located on sex chromosomes were removed for these analyses.

Utilizing the MEI callsets generated from the PAV dataset, we determined the number of polymorphic MEIs for each species (i.e., heterozygous elements) based on genotyping

information generated by PAV and our BLAT approach. In addition, we screened for polymorphic MEIs of PAV insertion calls present in both chimpanzee species as the divergence of these species is of recent origin. The speciation estimate of Sumatran and Bornean orangutans is even more recent. Hence, we also surveyed shared orangutan events for the presence of heterozygous MEIs in Bornean and/or Sumatran orangutans. All heterozygous lineage-specific events were combined with the respective species-specific MEIs (**Fig. Repeat rebuttal 1b**). In agreement with our CACTUS unaligned species-specific analyses, our results confirm that orangutans contain a greater total number of polymorphic L1 insertions (PAB: 4262, PPY: 3250) compared to the other great apes (GGO: 842, PTR: 421, PPA: 388) (**Fig. Repeat rebuttal 1b**). In contrast, the other great apes harbor a greater total number of polymorphic *Alu* insertions (GGO: 1708, PTR: 523, PPA: 590) compared to both orangutan species (PAB: 123, PPY: 108), and we further support the quiescence of *Alu* mobilization in orangutans with barely more than 100 polymorphic elements in each orangutan species (**Fig. Repeat rebuttal 1b**).

Finally, we determined the abundance of full-length (defined as L1s ≥ 5900 bp in length) L1 insertions with intact open reading frames within each. To estimate this, we identified all full-length L1s within a species from the MEI callsets generated from PAV, and then determined the open-reading frame (ORF) status of ORF1 and ORF2 at the nucleotide level. We intersected this dataset with our polymorphic callset and determined the number of full-length L1 insertions with both ORFs intact for each great ape species (**Fig. Repeat rebuttal 1c**). These results indicate that both orangutan species contain over four times (PAB: 5.82/4.79, PPY: 5.67/4.66) as many full-length L1s with intact ORFs each compared to PTR and PPA, respectively, and over three times compared to GGO (PAB: 3.19, PPY: 3.11). Furthermore, on average, 63.4% of the orangutan-specific full-length functional L1s (PAB: 67.9%, PPY: 58.9%) are polymorphic MEIs. These results provide further evidence for the recent increase in L1 mobilization within both orangutan species. Based on our findings, orangutans appear to have the highest transposable element mobilization rate, primarily driven by L1 of all great ape species.

I am less intrigued by the results on Pan-genome graph: What is it supposed to tell us, and how can it be used?

The pangenome provides us, in principle, an approach unbiased by the (human) reference to both discover and distinguish ancestral from derived alleles in any ape lineage. We also used it to develop a conservation score for every base pair in the genome. We made this clearer in the text and moved some of the details of the description to the supplement. The text has been reorganized into a single paragraph as follows:

“Ape pangenome. Most previous genome-wide comparative studies of apes have been limited by mapping inferior assemblies to a higher quality human genome and, as a result, introduce reference biases. Because the assemblies are comparable in quality and contiguity, we sought to mitigate this bias by creating an ape pangenome as an unbiased framework to understand evolution. First, we employed Progressive Cactus¹⁵ to construct several pangenomes, using different parameters, which included human and ape haplotypes (Supplementary Note III; Data Availability). Predictably, the resulting interspecies graphs were more complex than the recently released human pangenome for 47 individuals¹⁶—one ape pangenome, for example, was 3.38 Gbp with three times the number of edges and nodes. The resulting alignments allowed us to annotate genes and repeats as well as the ancestral state of more base pairs of

human GRCh38. Applying the parsimony-like method used by the 1000 Genomes Project and Ensembl¹⁷, we observe a genome-wide increase of 6.25% in the total ancestrally annotated base pairs over the existing Ensembl annotation (release 112), with the greatest autosomal increase for chromosome 19 (21.48%; Fig. Ancestral.S1). As a second approach, we also applied pangenome graph builder (PGGB)¹⁸ to construct all-to-all pairwise alignments for all 12 primate haplotypes along with three T2T human haplotypes (T2T-CHM13v2.0 and T2T-HG002v1.0). We used these pairwise alignment data to construct an implicit graph (Methods) of all six species and computed a conservation score using phastCons v1_5 for every base pair in the genome (Fig. PanGenome.S1; Methods). The approach is transitive without a reference bias and considers both assembled haplotypes for each genome, as well as unique and repetitive regions, identifying the most rapidly evolving regions in each primate lineage, including the major histocompatibility complex (MHC) and the chromosome 8p23.1 inversion (Fig. PanGenome.S1).”

ILS: What new insights have been gained? Is ILS of repetitive regions important and if so, why? Can we say anything about the selective neutrality of these?

Leveraging novel algorithms for ILS inference with high genome resolutions allowed us to identify a higher proportion of the genome in ILS when compared with previous surveys. Moreover, we explored the role of repetitive regions in our ILS inference. We iteratively removed 1000 bp windows overlapping with chm repeats in HCGO quartet inference. When all the repeats-containing windows are excluded, the inferred ILS is 37.6%, consistent with those estimated in Rivas-Gonzales et al. 2024 using the alignments from Shao et al. 2023. When we excluded repeats longer than 100 bp and 1000 bp, the inferred ILS proportion is 37.6% and 38.5%, respectively, increasing to 39.4 when repeats longer than 10⁵ bp are removed (**Fig. ILS.S6**). These results suggest that repetitive regions may play a role in the increased ILS proportion inference.

We added the following paragraph in the Supplementary Note:

“Furthermore, we explored the role of repetitive regions in increased ILS estimation in HCGO, when compared with previous estimates²⁹. In doing so, we iteratively excluded 1000 bp windows overlapping with the T2T-CHM13 repeat track, considering different repeat sizes. When all the repeats-containing windows are excluded, the inferred ILS is 37.7%, which is consistent with those estimated in Rivas-Gonzales et al. 2023²⁹ using the alignments from Shao et al. 2023²⁶. When we excluded repeats longer than 100 bp and 1000 bp, the inferred ILS proportion is 37.6% and 38.5%, respectively, increasing to 39.4 when repeats longer than 10⁵ bp are removed (**Fig. ILS.S6**). These results suggest that repetitive regions account for ~2% of the ILS increase consistent with selectively neutrally evolving DNA.”

Figure ILS.S6. Estimates of ILS proportion (x-axis) in HCGO when excluding 1000 bp windows overlapping with repetitive regions of different sizes (y-axis). Numbers in the y-axis refer to the lower bound of the excluded repeats. Confidence intervals refer to standard deviations inferred across 1000 bp windows.

To explore the association between inferred ILS in HCGO and adaptation we intersected ILS proportion and selective sweeps at the gene level for bonobo, chimpanzee, and gorilla. We observed that genes putatively under selection have significantly lower ILS proportion than those with no evidence of selective pressure (Wilcoxon rank sum test, $p\text{-value}=5.8^{-12}$). This signal is driven by the putatively selected genes identified by SweepFinder2, which is designed to detect predominantly distorted Site Frequency Spectrum pattern due to hard sweeps. In fact, the ILS proportion for selected genes with saltiLASSI, which is tuned to detect both hard and partial sweeps, is significantly larger than those with no evidence of selection ($p\text{-value}=6.04^{-6}$). These results confirm that ILS patterns are directly affected by selective trajectories.

Figure ILS.S7. ILS proportion and inferred selective pressure. We intersected ILS proportion and selection scans results at gene level. Names showing an excess or defect of ILS (exceeding 1.5-fold the interquartile range) are annotated.

Selection and diversity. Is it important to have more putative sweeps identified? Is there a difference in the newly versus previously identified sweeps? Is the difference due to the better genomes or differences in methods?

Yes, the fact that many of the new sites of selective sweeps map to duplicated genes provides a more balanced view of genes important in species adaptation. Although it is difficult to directly compare whether the selective sweep region is completely new with the completed genome, we indicated whether each gene was previously reported in **Table Selection.S3**. In addition, we also examined the selective sweep of genes that are nested inside an SD, using 100 kbp unique flanking sequences of each segmental duplication. The new SD gene models have been added to the **Table Selection.S3**. Overall, we find that 36 genes are new gene models and seven of these genes (*ZN682*, *HYES*, *IPSP*, *TRAC*, *MSTO1*, *CL18A*, *NPB13*) are located in SDs with sweep signature in both up and downstream unique sequences.

“Among the ape species, we identify 143 and 86 candidate regions for hard and partial selective sweeps (**Table Selection.S3**), along with seven duplication-overlapping genes showing signatures of selection on both unique flanking sequences of the duplications.”

Immunoglobulin genes: I find the results very hard to evaluate as they are now explained. I would like to have either more motivation and details or less emphasis on these results. What are the expected error rates in orthology inference?

We added additional motivation for the inclusion of IG and TR gene analysis in this manuscript. This centers primarily on the fact that accurate characterizations of the IG and TR loci have remained elusive in many previous genome assemblies, plagued mostly by technical errors or limitations using B-lymphocytes. Likewise, the complexity of these loci has long hindered the use of short-read sequencing datasets. This is in stark contrast to MHC genes, which have been studied extensively across vertebrate species. Thus, the high-quality T2T assemblies generated by the current effort provide the first opportunity to 1) generate robust curations of the IG and TR genes for ape species, but 2) also shed initial light on locus divergence patterns among closely related species. Specifically, the results demonstrate that germline structural variation is a major contributor to interspecific and intraspecific diversity. Further, our analyses highlight contrasting features between IG and TR loci, as well as differences between other critical genes in the adaptive immune system, including HLA/MHC. Our revised text has attempted to draw out some of these points more clearly alongside the results and observations made from the analysis of these loci.

With respect to orthology, we explicitly chose not to assign direct orthologs of IG and TR genes between species. As we note in the supplemental results, the extreme structural differences and divergence observed between haplotypes within and between species will necessitate the characterization of additional haplotypes from all species before orthologs can be defined. To be clear, the haplotype alignments shown in Figure 3 do not necessarily represent orthologous regions but simply represent the best aligned sequence blocks between each respective haplotype comparison and show the distribution of percent identity values within IG and TR loci and in comparison with MHC loci.

“Immunoglobulin/T-cell receptors and major histocompatibility complex (MHC) loci. Complete ape genomes make it possible to investigate more thoroughly structurally complex regions known to have a high biomedical relevance, especially with respect to human disease. In addition, four of the primate genomes sequenced and assembled here (bonobo, gorilla and two orangutans), as well as the human T2T reference, are derived from fibroblast instead of lymphoblastoid cell lines. The latter, in particular, has been the most common source of most previous ape genome assemblies limiting characterization of loci subject to somatic rearrangement (e.g., VDJ immunoglobulin genes)³⁴. Thus, we specifically focused on nine regions associated with the immune response or antigen presentation that are subjected to complex mutational processes or selective forces.

Immunoglobulin and T-cell receptor loci. Antibodies, B-cell receptors, and T-cell receptors mediate interactions with both foreign and self-antigens and are encoded by large, expanded gene families that undergo rapid diversification both within and between species^{35,36}. We conducted a comparative analyses of the immunoglobulin heavy chain (IGH), light chain kappa (IGK), and lambda (IGL) as well as T-cell receptor alpha (TRA), beta (TRB), gamma (TRG), and delta (TRD) loci in four ape species (**Supplementary Note X**) for which two complete intact haplotypes were constructed (**Extended Fig. 2a, Fig.IG.S1a**). We identify an average of 60 (IGHV), 36 (IGKV), 33 (IGLV), 46 (TRAV/TRDV), 54

(TRBV), and 8 (TRGV) putatively functional IG/TR V genes per parental haplotype per species across the seven loci (**Extended Fig. 2a, Additional file. Table IG.S1**). As previously noted for human haplotypes, the ape IG/TR gene loci are characterized by large structural differences within and between species, accounting for as much as 33% of inter-haplotype length differences in IG and up to 10% in the TR loci; this suggests IG loci undergo more rapid divergence than the TR loci, possibly due to evolutionary and functional constraints placed on TR loci by required interactions with MHC (**Extended Fig. 2b-c**). Additionally, IG loci show the most pronounced structural variation, including gene-spanning expansions of tandem duplications across all IGH haplotypes, long-range rearrangements in both gorilla IGK haplotypes, and a notable 1.4 Mbp inversion distinguishing the two IGL haplotypes of bonobo (**Extended Fig. 2a; Fig. IG.S1a-b**). These large-scale differences frequently correspond to species-specific genes (those that comprise phylogenetic clades of genes unique to either bonobo, gorilla or both orangutan species) (**Extended Fig. 2aa; Fig. IG.S1c**). We observe the greatest number of species-specific genes within IGH (**Fig. 3a; Fig.IG.S1d**), where we also note a greater density of SDs longer than 10 kbp relative to the other six loci (**Extended Fig. 2d**), indicating that IG genomic structure is likely a key driver of intra-species evolution.”

Methylation: It is unclear to me if two different cell line types are very interesting for methylation inference. It is also unclear what we learn from these

Replication timing: Again, what do we learn?, is the tissue relevant? Are differences between species related to other genome features, I remember claims that recombination is important. GC- content as well

The referee is correct that the methylation and replication timing data are limited with the former being restricted to underlying fibroblast and lymphoblast cultures used to construct the genomes. The data, however, reveal patterns that are shared between these two divergent cell types, and potentially generalizable when data from a broad array of cell types become available. Moreover, these data are particularly relevant and important for the characterization of the inaccessible regions with respect to general genome-wide features for each species. For example, the methylation data was used to discover a hypomethylation pocket association with almost all spacer regions as part of the subterminal caps, which are flanked by heavily methylated satellite regions. Repli-seq data showed that these particular regions were among the latest replicating regions and that in the case of chimpanzee chromosomal arms that contained these structures were later replicating than chromosomes that did not (see below). In addition, the CpG methylation data was used to assess the CDRs (centromere dip regions). We used the methylation data to show that the CDRs generally correlated with HOR length in bonobos but there was a minimal size below which CDRs were not observed—suggesting a functional limit. Finally, the observation that the patterns of DNA methylation are conserved in the majority of promoters and that variably methylated promoters share similar genomic, methylation, and transcriptome features in both fibroblast and lymphoblast cell lines argues that methylation data may be a powerful approach to distinguish potential functional genes versus pseudogenes.

Figure RT. rebuttal 1. a) Box plots comparing the log₂ ratio of replication timing in the telomeric regions (1 Mbp) of each chromosomal arm and centromeres. The replication timing is calculated as the log₂ ratio between early and late S-phase cells using BrdU immunoprecipitation¹². Chromosomal arms are categorized into three groups based on the presence of subterminal pCht satellites, other types of subterminal pCht satellites, or the absence of repeats. Centromere repeats are shown as a comparison. **b)** Replication timing profile for three chromosomes in chimpanzee and two chromosomes in gorilla. The y-axis represents the log₂ ratio of replication timing at a resolution of 50 kbp per bin. The colored bars below indicate the location of CenSat repeats. The significance of p-values from the Wilcoxon test are shown as stars: * denotes p < 0.05, ** denotes p < 0.01, *** denotes p < 0.001, and NS indicates nonsignificant results.

Figure RT.S3. Box plots showing the replication timing of telomeric regions of each chromosomal arm at varying distances from the chromosome ends (i.e., 50 kbp, 100 kbp, 250 kbp, 500 kbp, and 1 Mbp). The significance of p-values from the Wilcoxon test is indicated: *

denotes $p < 0.05$, ** denotes $p < 0.01$, *** denotes $p < 0.001$, and NS indicates nonsignificant results.

Figure RT.S4. The dot plot on the left shows the replication timing of each subterminal pCht satellite in chimpanzee. The horizontal line indicates the standard error of replication timing. Subterminal pCht satellites are categorized into subterminal pCht satellites (top) and interstitial pCht satellites (bottom). The bar plot shows the length of each subterminal pCht satellite, ordered by the size. The scatterplot on the right shows the relationship between the average replication timing and the length of the subterminal pCht satellites. The blue curve and gray shaded area are calculated using the LOESS smoothing method.

In response to the reviewer’s comment, we minimized the replication timing and methylation section (moving much of the details to the supplement) and instead emphasized its application in understanding epigenetic properties of complex regions of ape genomes throughout the manuscript. We now briefly describe both aspects in a section summarizing additional genome annotations available:

“Genome browser and other annotation features. To facilitate downstream analyses with these genomes, we developed a UCSC Browser hub (<https://github.com/marbl/T2T-Browser>) that includes the underlying genome-wide alignments, gene annotations, accelerated regions, repeats, etc. (Data availability). We augmented this browser with additional tracks to be used in future studies and follow-up papers. For example, we mapped Repli-seq, including previously collected NHP datasets⁷¹, to investigate evolutionary patterns of replication timing and identified 20 distinct states of replication timing that can

now be correlated with different genomic features (**Fig. RT.S1, S2**). Because they can interfere with mitochondrial DNA annotation, we created a curated set of integrated NUMTs (nuclear sequences of mitochondrial DNA origin) in ape genomes (**Methods**). We observe a substantial gain in the number (3.7-10.5%) and total length of NUMTs (6.2-30%) (**Table Repeat.S11**) over non-T2T assemblies. We also identified and annotated a nearly complete catalog of sequence motifs capable of forming non-canonical (non-B) DNA (G-quadruplexes, A-phased, direct, mirror, inverted, and short tandem repeats in particular), especially in previously inaccessible regions that have been historically difficult to sequence⁷² (**Table AssemblyS10; Supplementary Note XV**). We overlay these data with a multiscale epigenomic map of the apes, including DNA methylation and replication timing (**Supplementary Note XVI-XVII**). Using 5-methylcytosine (5mC) DNA signature from long-read sequencing data, for example, we distinguish hypomethylated and hypermethylated promoters associated with gene expression and demonstrate that in each cell type, the majority (~88%) of promoters are consistently methylated (8,174 orthologous ape genes assessed) (**Fig. MET. S2; Table MET.S1-2**). The analysis identifies on average ~12% of promoters that are variably methylated between species (1,035 for fibroblast and 922 lymphoblast cell lines) as candidates for regulatory differences among the species (**Table MET.S1-2**). Consistently methylated promoters were associated with more highly expressed genes and had a higher density of CpG sites compared to variably methylated promoters ($P < 10^{-16}$ two-sided Mann–Whitney U test, **Fig. MET.S2**), highlighting the association between evolution of CpG sequences, methylomes, and transcriptomes. These annotations will facilitate future investigations to understand genome evolution and epigenetic changes associated gene expression changes in ape genomes^{73,74}.”

Accelerated regions. These could be interesting but the discussion is hard to follow - would be great with a simpler rewrite of this. That could include statistical tests for the suggestions/claims put forward.

To improve clarity, we simplified our discussion of accelerated regions. In particular, we rewrote this section to narrow our claims to those that are well supported by statistical evidence.

“Regions of rapid divergence. Because rapidly evolving regions can help pinpoint genes under positive selection⁴¹ or cis-regulatory elements (CREs) undergoing functional changes⁶⁶, we applied a mutation-counting approach to identify ancestor quickly evolved regions (AQERs) that diverged during ape evolution⁶⁶ (**Methods; Supplementary Note XIII**). We identified 13,218 AQER sites (**Table AQER**) across four primate lineages, including 3,268 on the human branch (i.e., HAQERs). Our analysis more than doubles the number of HAQERs identified from previous gapped primate assemblies ($n=1,581$) (**Fig. 4a, Fig. AQER.S1**). Such elements are highly enriched in repetitive DNA (e.g., 5-fold for SDs $p < 1e-30$, 2-fold for simple repeats $p < 1e-30$). HAQERs also exhibit a significant enrichment for bivalent chromatin states (repressing and activating epigenetic marks) across diverse tissues, with the strongest enrichment being for the bivalent promoter state ($p < 1e-35$) (**Fig. 4a; Table AQER.S1**)—a signal present to a lesser degree on the chimpanzee, bonobo, and gorilla lineages ($p < 1e-19$, $p < 1e-30$, $p < 1e-12$ respectively). This lesser enrichment may be due to the chromatin states being called from human cells and tissues (**Supplementary Note**). An example of a human-specific HAQER change includes an exon and a potential CRE in the gene *ADCYAP1*, in the layer 5 extratelencephalic neurons of primary motor cortex. This gene shows convergent downregulation in human speech motor cortex and the analogous songbird vocal learning layer 5 type extratelencephalic neurons necessary for speech and song production^{67,68}. We find here downregulation in layer 5 neurons of humans relative to macaques (RNA-seq) and an associated unique human epigenetic signature (hypermethylation and decreased ATAC-seq) in the middle HAQER

region of the gene that is not observed in the same type of neurons of macaque, marmoset, or mouse (Fig. 4b, Fig. AQER.S2-S3).”

NUMTS: What makes them interesting here? Why is 8% difference between orang species interesting

The statement has been removed in the main document. In response to reviewer’s comments on these sections, we also significantly reduced or modified these sections, including moving ILS, selective sweeps, methylation, replication time and NUMTS to the supplement. Instead, we mention these analyses in the beginning and point referees to SI for more detail. As suggested, we significantly revised (and reduced) the IG section and accelerated regions moving some of the associated display items to extended display or SI.

Specific comments and questions

Assemblies: Is it possible to give good advice for optimal strategies for more T2T primate genomes going forward? How important is nanopore compared to pacbio and what should be ideal coverages of each?

Given the rapid advances in LRS technologies, most advice would be temporal in nature and soon obsolete. What we can say now is that both technologies are critical to achieving T2T genomes—HiFi providing the sequence accuracy necessary for the backbone of the assembly, and UL-ONT providing the scaffolding information necessary to traverse repeats, trio or Hi-C for phasing haplotypes and highly accurate short-reads for polishing. Herro-corrected ONT reads were recently suggested as an alternative for HiFi reads to get near T2T genomes with slightly lower base-level accuracy as shown in Koren et al., 2024¹³. We added the following sentence to the discussion:

“The length (nearly double the size) and the complex compound organization of orangutan α -satellite HOR sequence will require specialized efforts to completely order and orient **these large tandem arrays**⁸¹. **The combination of HiFi reads to generate the high-quality assembly backbone and UL-ONT sequencing for scaffolding across the largest repeats was critical for closure of most repeats and it may be that even longer UL-ONT data coupled with new advances in ONT error-correction**¹¹¹ **would allow for the longest tandem repeats in the Sumatran genome to be fully resolved.** Nevertheless, with the exception of these and other large tandem repeat arrays, we estimate that ~99.5% of the content of each genome has been characterized and is correctly placed.”

L 236 states that contiguity is an order of magnitude improved; how should that be understood (Cell paper has N50 contig size of 50Mb or above)

This is a good point. The genomes that we were comparing were the standard references generated in the past and not some of the new LRS genomes that, while still incomplete, represent significant improvements in contiguity. The statement has been removed.

315 “We infer ILS for an average of 39.5% of the autosomal genome and 24% of the X chromosome, representing an increase of approximately 7.5% compared to recent reports from less complete genomes²⁴ in part due to inclusion of more repetitive DNA” It is not clear what the 7.5% refers to

The 7.5% refers to ILS fractions reported in citation 24. We revised the sentence to read:

“We infer ILS for an average of 39.5% of the autosomal genome and 24% of the X chromosome. Compared to recent reports³⁰, this represents a 7.5% increase in autosomal ILS, in part due to inclusion of more repetitive DNA.”

L 353 “Similarly, we observed 33,032 (0.7%) soft-clipped reads (>200 bp) in the gorilla T2T assembly in contrast to 89,498 (2%) soft-clipped reads

Why so many soft-clipped reads left even with the new annotation? Is it technical?

Yes, we believe this is a technical issue related to sequencing error and library preparation. The plot below compares base quality scores and k-mer content (reported by fastqc) between soft-clipped reads that could or could not be rescued by the gorilla T2T assembly. The latter tend to have lower quality scores and higher content of polyA sequences, suggesting that soft-clipping of reads mapped to the gorilla T2T assembly was due to low-quality read (sub-)sequences. For chimpanzee, bonobo, and Sumatran orangutan, the two sets of soft-clipped reads showed less differences in base quality scores, but a very similar difference in polyA sequence bias (data not shown).

L 548: “We identified 20 states with different patterns of replication timing. Overall, the replication timing program is largely conserved, with 53.1% of the genome showing conserved early and late replication timing across primates, while the remaining regions exhibit lineage-specific patterns...”

I am not sure what states refer to and how to evaluate these differences

We removed this from the main but the definition of “states” is already explained in the supplement as follows:

“

Evolutionary patterns of replication timing

To characterize changes in replication timing across the primate species, we applied Phylo-HMGP¹⁷⁷, focusing on lymphoblastoid cell lines from human, chimpanzee, Bornean orangutan, and siamang. Phylo-HMGP identified 20 states with distinct evolutionary patterns (**Fig. RT.S1A**). These states were classified into five categories based on the replication timing signal values and the variability across the species: early (E), weakly conserved early (WE), late (L), weakly conserved late (WL), and non-conserved (NC). We observed that 31.5% of the genome is in conserved early state and 21.5% in the conserved late state. Lineage-specific states were mainly reflected by weakly conserved and NC states. For example, State 13

has a unique early replication timing specific to chimpanzees, while State 4 shows a distinct late replication timing unique to Bornean orangutans. The distribution of replication timing signals for each state, organized by category, is displayed in **Fig. RT.S1B**. **Fig. RT.S1C** shows an example of replication timing profiles and Phylo-HMGP states on the genome browser.

Correlation between SDs and replication timing evolutionary patterns

We analyzed the relationship between SDs and replication timing, focusing on lineage-specific, nonhomologous sequence elements. The genome was divided into 5 kbp bins, and replication timing signals were calculated for SDs and the genome-wide background. A Wilcoxon rank-sum test assessed the differences between SDs and the background. We found that ancient SDs shared across many species (e.g., BCHGO and BCHGOS) tend to replicate early ($p\text{-value} < 1e\text{-}20$), while more recent SDs (e.g., O and BC) and species-specific SDs tend to replicate late ($p\text{-value} < 1e\text{-}50$). Examples of the replication timing distribution are shown in **Fig. RT.S2**.”

Figure RT.S1. Phylo-HMGP identifies 20 states with distinct evolutionary patterns of replication timing in lymphoblastoid cells from four primary species. (A) The top panel shows the proportion of each state across the entire genome, while the bottom panel displays the average replication timing signals in each state, with columns representing different states ordered by the average human replication timing signals. **(B)** The 20 Phylo-HMGP states are categorized into five groups: early (denoted as E), weakly conserved early (WE), late (L),

weakly conserved late (WL), and non-conserved (NC). Violin and box plots show the replication timing distributions for each state, organized by category. **(C)** Visualization of replication timing patterns and state annotations in the genome browser (chr1:50,000,000-75,000,000 in human T2T genome version 2.0).

Figure RT.S2. Correlation between segmental duplications (SDs) and replication timing (RT). The blue density curves represent the distribution of RT signals in SD regions, and the

gray density curves represent the background RT signals. **(A)** Distribution of RT signals in more ancestral SDs, including BCHGO (shared among bonobo, chimpanzee, human, gorilla, and orangutan) and BCHGOS (shared among bonobo, chimpanzee, human, gorilla, orangutan, and siamang). **(B)** Distribution of RT signals in more recent SDs, including Specific (species-specific), BC (shared between bonobo and chimpanzee), and O (shared between two types of orangutans).

L 980 These include an improved and more nuanced understanding of species divergence, human ancestral alleles, incomplete lineage sorting, gene annotation, repeat content, divergent regulatory DNA, and complex genic regions as well as species-specific epigenetic differences involving methylation

This quote from the discussion also points to these sections as perhaps not the most important to report on in this manuscript since more novel results can “drown” in the large body of information

We agree and significantly reduced these sections in an effort to focus on the major new findings especially related to the inaccessible regions.

Referee #2 (Remarks to the Author):

This paper reports haplotype-resolved T2T assemblies for 6 ape species, including an outgroup to the great apes. The included clades are key to studies of human evolution, and although highly contiguous assemblies for them were previously available the authors set out to bridge the remaining gap in quality to human sequence, enabling a comparison across several structurally complex sequence classes, for many of which this was previously not possible genome-wide. The manuscript includes an extensive characterization of the resources across numerous analyses, which the authors broadly group into a description of the assemblies, improvements of analyses based on them, and a description of newly resolved sequences. The methods to bring the whole genomes to near completion are state of the art, I have no concerns or comments regarding their quality, not least as the same approach applied to the human genome has previously been reported by some of the authors, and the improvements are widely recognized. It's wonderful to see the assemblies of some key NHP species being brought to the same quality, a commendable effort that will undoubtedly be of wide use across several fields. The manuscript aims to place the new assemblies in the context of previous analyses, while also describing the previously inaccessible sequences. The improvement of the genomic resources in the latter is self-evident. For the former, some analyses might sometimes provide more nuance, but this is in part difficult to assess or follow given the somewhat frequent lack of context. In the best case, the reader is then unable to appreciate the refinements and improvements, but it might also give the impression of novelty for results that are at times rather confirmatory (a fact that also in part acknowledged and does not diminish the importance of the work in any way). I've listed examples below, given the tradeoff between clearly communicating improvements and the length of the text, the authors might want to reconsider the presentation of some of these sections to make the paper more accessible, although this is ultimately their choice. The same applies to those reporting some one-offs that are named in the text but don't seem to be used elsewhere, or whose use is at least not immediately evident, e.g. several different alignments or 3D maps, among others.

While I have no concerns regarding the assemblies themselves, I do for a subset of the more focal analyses, which I hope the authors can clarify. They are listed together with additional questions and comments below.

We appreciate the referee's overall enthusiasm for the release of the new assemblies in addition to their time and thoughtful review. Based on their comments and that of Referee #1, we significantly streamlined the main text to focus more on the novel findings as opposed to the more nuanced improvements. This entailed consolidating some sections and moving some of the detailed analyses such as the pangenome alignments, methylation, and replication timing analyses to the supplement.

###

It is difficult to follow the provenance and identity of different cell lines or other samples. The available information is currently scattered across e.g. Table 1, Table AssemblyS1, the supplementary with references to prior work, or the reporting summary. The tables include species-level aggregates, the identifiers for the parental samples seem to be completely

missing. The authors state that the journals' policy considerations regarding "Research animals and/or animal-derived materials that require ethical approval" do not apply, a statement that is challenging to evaluate. Please include a table reporting uniquely identifiable sample-level information for all samples used throughout this manuscript, including at least information on the source biorepository, biosample id, cell type, primary or immortalized, immortalization type if relevant, all generated data types, and provenance of the samples where available, particularly for samples that are not in public biobanks. I would also appreciate any additional information the authors might be able to retrieve and deem potentially relevant (e.g. number of passages prior to expansion, year the cell line was established, information on the origin, e.g. from an individual in a Zoo, Sanctuary, their habitat, wild-born / first generation captive, captive bred, etc.). I have no reason to question the legitimacy of the source of sampling, but the numerous ethical considerations and resulting sensitivities surrounding the research use of endangered animals are probably best preventively addressed with as much information as possible. It might be possible to compile this information in part via the referenced prior work, but I believe the high visibility I expect this manuscript to receive warrants not leaving this task to the reader.

We expanded the table including as many additional details as possible on the samples, including biosample ID, whether the cell line is primary or immortalized, Studbook number, ISIS ID, age, source breeding history and whether EBV was detected. Please note in the case of the parents there are no cell lines but rather DNA was obtained from blood that was collected as opportunistic health checks.

Supplementary Table Assembly.S1. Description of the ape samples and genome data.

Common name	Scientific name	Name	Desc.	Tissue	Sex	Biosample ID	Primary or immortalized	Studbook number	ISIS ID	Age (yrs)	Source	Breeding history	EBV detected
Chimpanzee (PTR)	Pan troglodytes	mPanTro3	n/a	Lymphoblastoid	M	SAMN30216104	Immortalized	n/a	n/a	n/a	Coriell	n/a	EBV
Bonobo (PPA)	Pan paniscus	mPanPan1	Child	Fibroblast/ Lymphoblastoid	M	SAMN13935689	Fib. primary / lym immortalized	171	GAN 7225479	n/a	San Diego Zoo Wildlife Alliance, San Diego, CA, USA	Captive Born	EBV
		mPanPan2	Mother	Fibroblast	F	SAMN35877942	DNA only, not a cell line	59	ISIS174013	38	San Diego Zoo Wildlife Alliance, San Diego, CA, USA	Captive Born	n/a
		mPanPan3	Father	Fibroblast	M	SAMN35877943	DNA only, not a cell line	58	ISIS180343	15	Democratic Republic of the Congo	Wild Born	n/a
Gorilla (GGO)	Gorilla gorilla	mGorGor1	Child	Fibroblast	M	SAMN04003007	Primary	500	MIG12- 24804338	n/a	San Diego Zoo Wildlife Alliance, San Diego, CA, USA	Captive Born	n/a
		mGorGor2	Mother	Fibroblast	F	SAMN35877944	DNA only, not a cell line	195	ISIS24489	39	Democratic Republic of the Congo	Wild Born	n/a
		mGorGor3	Father	Fibroblast	M	SAMN35877945	DNA only, not a cell line	97	ISIS24485	46	Democratic Republic of the Congo	Wild Born	n/a
Bornean orangutan (PPY)	Pongo pygmaeus	mPonPyg2	n/a	Fibroblast	M	SAMN10521809	Primary	1127	GAN 23049074	n/a	San Diego Zoo Wildlife Alliance, San Diego, CA, USA	Captive Born	n/a
Sumatran orangutan (PAB)	Pongo abelii	mPonAbe1	n/a	Fibroblast	M	SAMN10521808	Primary	n/a	n/a	26	Laura Carrel (PSU), originally from Coriell	n/a	n/a
Siamang (SSY)	Symphalangus syndactylus	mSymSyn1	n/a	Lymphoblastoid	M	SAMN30216103	Immortalized	AZA484	n/a	14	Oregon Health and Science University	Captive Born	EBV

I did not find any mention of the potential impact of using immortalized LCLs, and maybe other cell lines, which are prone to numerous alterations affecting the whole spectrum of variation described in the manuscript (beyond somatic rearrangements in VDJ genes), as a result of the transformation process itself and cell propagation. It seems karyotype normality was assessed and the authors describe EBV contamination that required filtering, were any additional measures taken to ensure other potential alterations do not affect the described sequences and conclusions? I believe a convincing assessment of this matter is very difficult to do, especially

for most newly described sequence classes or variant types, it might be possible to at least increase confidence indirectly via established markers. Where do the great-ape samples fall relative to the well-characterized ranges of variation of the diversity panel the authors used, e.g. in a PCA? I don't necessarily expect the samples to behave like natural populations or think they are required to, but it seems important to ascertain they don't behave completely differently where possible, and this potential limitation should be clearly communicated. On a similar note, were any measures taken to ensure correct species labels, particularly the Siamang?

We performed three analyses to ensure the integrity and biological relevance of the genomes. As described, we 1) performed karyotype analysis on each cell line, 2) removed EBV from the assemblies for those two that were transformed, and 3) comparative analyses against the other haplotype and other samples from the same species. The latter is particularly powerful because somatic artifacts derived from cell lines are most likely heterozygous in nature and observing essentially the same structure in another independent assembly of a different haplotype confirmed the biological significance of most interspecific differences. Based on feedback from Referee #1, we also took advantage of LRS-assembled genomes from Mao et al. and generated the genome-wide alignment as follows (**Figs. Assembly.rebuttal 1-5**). The results confirm a high degree of collinearity with the exception of centromeres, subterminal heterochromatin, and acrocentric DNA.

Chimpanzee

T2T (top) vs Clint_PTR primary

T2T vs Clint_PTR alternative

Figure Assembly.rebuttal 1. Alignment of T2T chimpanzee versus the previous LRS-assembled genomes (GCA_030128855.1 & GCA_030128845.1).

Bonobo

T2T vs Mhudiblu_PPA primary (bottom)

T2T vs Mhudiblu_PPA alternative

Figure Assembly.rebuttal 2. Alignment of T2T bonobo versus the previous LRS-assembled genomes (GCA_030221875.1 & GCA_030221885.1).

Gorilla

T2T vs Kamilah_GGO primary

T2T vs Kamilah_GGO alternative

Figure Assembly.rebuttal 3. Alignment of T2T gorilla versus the previous LRS-assembled genomes (GCA_030174185.1 & GCA_030174155.1).

Sumatran orangutan

T2T vs Susie_PAB primary

T2T vs Susie_PAB alternative

Figure Assembly.rebuttal 4. Alignment of T2T Sumatran orangutan versus the previous LRS-assembled genomes (GCA_030170355.1 & GCA_030170345.1).

Bornean vs Sumatran orangutan

T2T (PPY) vs Susie_PAB primary

T2T (PPY) vs Susie_PAB alternative

Figure Assembly.rebuttal 5. Alignment of T2T Bornean orangutan versus the previous LRS-assembled genomes (GCA_030170355.1 & GCA_030170345.1).

We also performed multidimensional scaling (MDS) analysis using SNPs to see where the T2T samples fall relative to existing species and subspecies data. The MDS plots below based on pairwise SNP distances between samples suggest the T2T ape samples, indicated by blue datapoint (“Ref”), are not necessarily outliers at least among the samples used as part of the great ape genome diversity project (**Fig. Sequencing.S1-5**). Thus, we believe that each genome selected is a reasonable representative of the particular subspecies.

Figure Sequencing.S1. MDS plot using SNP distance matrix in bonobo (*Pan paniscus*).

Figure Sequencing.S2. MDS plot using SNP distance matrix in chimpanzee (*Pan troglodytes*), suggesting T2T chimpanzee genome is genetically related to western chimpanzee population.

Figure Sequencing.S3. MDS plot using SNP distance matrix in gorilla, suggesting the T2T gorilla is closest to western lowland gorilla (*Gorilla gorilla gorilla*).

Figure Sequencing.S4. MDS plot using SNP distance matrix in Bornean orangutan (*Pongo pygmaeus*).

Figure Sequencing.S5. MDS plot using SNP distance matrix in Sumatran orangutan (*Pongo abelii*).

To further illustrate the differences between Bornean and Sumatran orangutans, we computed the genome-wide average of pairwise identity between the genomes and found that between Bornean and Sumatran genomes, we observe an average identity of 98.9%, which was 0.3-4% lower than within-species identities (**Fig. Sequencing.S6**); among Sumatran orangutans, average pairwise identity was 99.2%, and between two haplotypes of Bornean orangutan pairwise identity was 99.3%.

Figure Sequencing.S6. Genome-wide average identity between orangutan genomes.

With respect to Siamang, we have a specimen report that came with the blood sample and the species (*Symphalangus syndactylus*) is defined in this report. The studbook number for this individual is AZA484 and the Global Accession Number (GAN) is MIG12-29960010.

Since these additional details constitute valuable information regarding authentication of biological material, we added these MDS and pairwise identity plots to Supplementary Note I.

L160: I have strong reservations about the use of “definitive” here. Disregarding remaining gaps and what should be considered a “complete” assembly, science is always subject to revision, and thus a result can hardly, if ever, be considered definitive.

Good point. We removed the term “definitive” and the sentence now reads:

“This resource should serve as a baseline for all future evolutionary studies of humans and our closest living ape relatives.”

L259: The authors have generated numerous alignments for this manuscript (at least 5 mentioned in this section, but there seem to be more referenced in other places). On the one hand, it is sometimes challenging to follow which of them was used for which analyses. On the other, it is not evident why this was deemed necessary, as several seem to be redundant, e.g. the various cactus alignments appear to be subsets of one, nor what the consequence of using a particular one is beyond the few cases of comparison.

The referee is correct. Different alignments were generated to optimize different pangenome constructions; in other cases, alignment parameters had to be tweaked especially when dealing with biologically complex regions. We agree this made the text cumbersome and, as a result, we moved much of this text to the supplement where we tried to explain when and why it was necessary. We simplify the main text to describe the two fundamentally different approaches and why they were used. Most of the analyses in the paper (i.e., gene and repeat annotation) are derived from the cactus alignments and we now make that clearer in the main text:

“First, we employed Progressive Cactus¹⁵ to construct several pangenomes, using different parameters, which included human and ape haplotypes. (**Supplementary Note III; Data Availability**). Predictably, the resulting interspecies graphs were more complex than the recently released human pangenome for 47 individuals¹⁶—one ape pangenome, for example, was 3.38 Gbp with three times the number of edges and nodes. The resulting **alignments allowed us to annotate genes, repeats, as well as the** ancestral state of more base pairs of human GRCh38.”

...

As a second approach, we also applied pangenome graph builder (PGGB)¹⁸ to construct all-to-all pairwise alignments for all 12 human primate haplotypes along with three T2T human haplotypes (T2T-CHM13v2.0 and T2T-HG002v1.0). We used these pairwise alignment data to construct an implicit graph

(Methods) of all six species and computed a conservation score for every base pair in the genome (Fig. PanGenome.S1; Methods).”

L280 & Sup. P25: Please explicitly state that phastCons was used to calculate the scores and report the model parameters.

We modified the main text and supplementary material to describe the version of phastCons that was used and the specific parameters.

“As a second approach, we also applied pangenome graph builder (PGGB)¹⁸ to construct all-to-all pairwise alignments for all 12 human primate haplotypes along with three T2T human haplotypes (T2T-CHM13v2.0 and T2T-HG002v1.0). We used these pairwise alignment data to construct an implicit graph (Methods) of all six species and computed a conservation score using phastCons v1_5 for every base pair in the genome (Fig. PanGenome.S1; Methods).

...

We generated per-base conservation scores and conserved elements tracks using PhastCons¹⁶ v1_5 with two approaches, following the PhastCons HOWTO (<http://compugen.cshl.edu/phast/phastCons-HOWTO.html>) as described in Secomandi et al¹⁷. A single random haplotype aligned to CHM13 was selected for each species/chromosome pair, including all sex chromosomes. In the first approach, we used default parameters (unsupervised EM learning), splitting alignments into 1 Mbp chunks and combining predictions. Since this approach estimated high conservation scores for the majority of the genome (mean 0.964), suggesting that the EM algorithm could not distinguish between constrained and neutrally-evolving regions for such closely-related species and relatively small sample size, we devised a second approach. For the second approach, we applied a fourfold degenerate site model, using entire alignments for all chromosomes except for chromosome 2, which was split into 1 Mbp chunks. In this approach, we inferred one model using all 4D sites across chromosomes and then applied it genome-wide. We provided strong priors to the model by fine-tuning PhastCons HMM model parameters --target-coverage and --expected-length using Grid Search to jointly maximize enrichment (Jaccard index) of CHM13 CDS sequences across all chromosomes. As the model was inferred genome-wide and the same final model parameters were applied to all chromosomes, conservation scores can be directly compared between chromosomes. The initial model was estimated with maximum likelihood and a tree from TimeTree (<https://timetree.org>). Exact software command and model parameters can be found in the github (https://github.com/T2T-apes/ape_pangenome/tree/main/conservation). “

The legend of Fig PanGenomeS1 D claims protein-coding sequences as most conserved, 4 other classes with higher average scores are shown and IG_V genes are presumably also protein-coding. The text states that lncRNA are among the least conserved, but they show among the highest scores in the figure.

This was a typo. We corrected the legend below. Also, we note that Fig. SeqDiv.S4 was a duplicate of Fig. PanGenome.S1D. The duplicate Fig. SeqDiv.S4 has been deleted.

“D) Conservation scores relative for specific gene annotation classes. Transcripts with retained introns, NMD transcripts, and protein-coding sequences are the most conserved, while pseudogenes and IGV genes are the among least conserved classes.”

The authors need to include a length-matched baseline of scores for all analyses, as Figure SeqDiv S3 suggests the distribution of these values to be in line with what can be expected from sampling them randomly across the genome. The interpretation “Conservation scores were strongly skewed towards 1 (mean 0.964), as expected for such closely related species.” is flawed and the presented results suggest that ~96% of the human genome is under purifying selection, when the high similarity of ape genomes and corresponding short branch lengths is largely the result of their shallow divergence.

We agree with the reviewer’s conclusion. The previously reported 96% came from the phastCons run that relied on unsupervised EM learning to infer the parameters, and this apparently did not work well. We now make it clear in the supplement that this was the reason why we additionally ran phastCons using a model generated from 4-fold degenerate sites. Results from the 4-fold degenerate sites make much more sense and are reported now.

“Conservation scores obtained with the four-fold degenerate site model were strongly skewed towards 0 (mean 0.276) as expected given most of the genome should not be under purifying constraint. Conservation scores differ among genomic features indicating that the elements that we expect to be under purifying selection (e.g. protein coding genes, rRNAs) are at a higher level of conservation than the background score (**Fig. PanGenomes.S1D**).”

The authors further report a strong correlation with pLI, but do not provide a correlation coefficient for what seems to be a surprisingly modest effect (PanGenomeS1 E). They describe genes off the linear trendline with apparently differing effects of selection in either metric, their number and function are not reported but the figure suggests such cases are frequent. These results seem unexpected by themselves given the inclusion of the human genome, but also in the light of recent large-scale reports on selective constraint across mammals and primates and should warrant further investigation. Taken together they do not instill confidence in the scores’ utility, which are likely severely underpowered. The analyses are buried in the supplementary text, but it seems all data has already been released, calling for caution given their potential usage in other projects.

We observed the correlation coefficient of $8.1e-164$, which is indeed very low. To be sure, because pLI is a population-based statistic, it is not unexpected to find genes lying off the trendline. This is, for example, the basis of the McDonald-Kreitman test, a difference in the inter-versus intra-species diversity. The panels E-H have been removed following the updates on conservation score.

L296 / Fig 2b: The reported gap-divergence metric is very difficult to interpret, as it combines

multiple sources of biological and technical variability, is not symmetrical, and appears highly subject to choices of the deployed alignment strategy or its filters, as evidenced by its distribution. If a region is single-copy in species 1 and duplicated in species 2, should the two copies be aligned to the same region in species 1 or one considered as counting towards gap-divergence? Should this scenario contribute in the same way to gap divergence as e.g. a deletion or failure to reliably align a centromere or any other satellite array? Which haplotype should be chosen, Table Div1 shows substantial differences between them, but Figures SeqDiv S1-2 only report distributions for one. Do the others look the same? While a discussion on the proportion of the assemblies that fails to reliably align for whatever reason seems adequate, it's not clear why this should be elevated to a metric that is comparable to the biologically much more clearly defined SNV divergence in orthologous regions in Fig 2b. I appreciate the desire to communicate that divergence assessed via SNVs doesn't capture all differences between species, if the authors wish to do so this should be done with more nuance in my opinion.

The reviewer makes several important points. It is indeed quite challenging to distill “gap divergence” into a single number for many of the aforementioned reasons. We chose to use wfmash precisely because it optimizes for the single longest alignment between two genomes, retaining the context around duplicated and deleted regions (other alignment tools are optimized for different use cases and may allow multiple mappings). The wfmash alignments are provided as a **track** in the UCSC Genome Browser and have also been useful for our team and others in confirming polymorphic TE insertions for example.

https://genome.ucsc.edu/cgi-bin/hgTracks?db=hub_4578304_mPanTro3_v2.0_pri&lastVirtModeType=default&lastVirtModeExtraState=&virtModeType=default&virtMode=0&nonVirtPosition=&position=chr10_hap1_hsa12%3A11682870%2D11699759&hgid=2307692016_K9ELdlqvIDbJNkH7ScdEtzqdtzEp

Gap divergence provides insight into how much of the genome can be aligned **and is one of the advantages of telomere-to-telomere genomes where this can be estimated for the first time**. For example, highly structurally divergent regions (e.g., the Y chromosome) exhibit substantially higher gap divergence. Nevertheless, we agree that “gap divergence” is a complex metric as it encompasses both biological as well as technical limitations of alignment and requires a much more detailed investigation. **Consequently, we dramatically shortened this part of the text and leave details regarding the advantage of using wfmash, for example, to the supplement. As suggested by the referee, we state what portion of the ape genome fails to reliably align and suggest that estimates of gap divergence are affected by mutation but also technical limitations of alignment.**

“Divergence and selection. Overall, sequence comparisons among the complete ape genomes reveal greater sequence divergence than previously estimated (**Supplementary Note III-IV**). This is because we now find that 12.5-27.3% of an ape genome fails to align or is inconsistent with a simple 1-to-1 alignment introducing gaps. Gap divergence estimates show a 5- to 15-fold difference in the number of affected Mbp when compared to SNVs due to rapidly evolving and structural variant regions of the genome **as well as technical limitations of alignment in repetitive regions (Fig. SeqDiv S1 & 2)**. We cataloged all structurally divergent regions (SDRs) among the ape genomes and found an average 327 Mbp of sequence (10%) per ape lineage (**Fig. 2a, Methods; Supplementary Note V**). Predictably, these include

centromeres, acrocentric short arms, and subterminal heterochromatic caps but also numerous gene-rich SD regions enriched at the breakpoints of large-scale rearrangements (Fig. SDR.S1).”

L303: The genome of the Bornean Orang has been sequenced on several occasions (e.g. Prado et al., 2013, Nater et al., 2017), and a genome assembly for this species has been reported in Shao et al., 2024, referenced here.

The statement has been removed from the text and we added these additional citations to the reference list:

“In short, these ape diploid genome assemblies represent an advance in terms of sequence accuracy and contiguity with respect to all prior ape genome assemblies⁴⁻⁸.”

L307: The referenced table SeqDiv. S2 does not seem to exist.

Thank you for pointing this out. We confirm that **Table SeqDiv.S2** was a typo and has been removed from the text.

L310, Sup. P39: The methods state that “ILS estimation on the following four-species (ABCD) phylogenies” was performed. Do the estimated percentages of ILS represent aggregates of all possible topologies different from the canonical tree (which seems to be ((A, B), C), D), or do they test a specific alternative? Please include a description of the rationale for choosing the specific quartets. The authors also ran msmc and chose a constant mutation rate across all species to scale the results of both analyses, which might not be realistic for all species, particularly the Siamang. Could they please elaborate on the potential impact this might have, which reported estimates are based on trails or msmc (this is only mentioned in the figure legend), and how well these estimates agree for nodes reconstructed in both? Is there any range of uncertainty for the Ne estimates, like they seem to provide for time? The methods mention a multiz alignment which doesn’t seem described elsewhere.

I miss a contextualization of the results reported here (other than the % increase of ILS) with prior estimates for the same nodes, despite diverging estimates even when based on the same method. For example, the Ne estimate for the African ape ancestor appears to be 25% higher than that reported in Rivas-Gonzales et al., 2023, and the estimates for the pan ancestor are 33% higher. The authors state to take the latter estimate with caution, but it is neither clear why, nor why confidence is higher for the deeper nodes. These differences seem substantial but might not be appreciated by the reader if unmentioned, nor does it seem sufficiently justified to simply consider them improvements rather than differences. The supplement mentions bp-resolution, suggesting it should be possible to subset the resulting estimates by regions without the requirement to rerun the analyses, if that is the case it would seem like a good opportunity to better understand these differences.

We thank the reviewer for their careful review of the ILS analysis and their suggestions.

The reported ILS percentages represent an aggregate of ((A,C),B),D) and ((C,B),A),D) phylogenies, harnessing D as outgroup. We amended the relevant paragraph as follows:

“In our analysis we considered any genomic region in ILS status when the reconstructed phylogeny is different from ((A,B),C),D).”

We choose the phylogenies in order to have at least one estimate for each branch across the analyzed species. We preferred to use *Homo sapiens* when possible, as in the case of HCOS and HGOS.

We now specify this in the Methods section of the Supplementary Note as follows:

“We performed the ILS estimation on the following four-species (ABCD) phylogenies, **selected in order to have at least one estimate for each branch across the analyzed species.**”

ILS is more likely when the ancestral population to two or three species is large; therefore, the inferred ILS proportion is dependent on N_e . The differences in ILS and N_e estimation between Rivas-González 2023 and our inferences are therefore correlated to N_e and may be the results of different harnessed methods, such as TRAILS and coalHMM, combined with a higher quality of the sequence data, which may have added additional sources of genomic variation. The estimated N_e for the human-chimp population is very similar across Rivas-González 2023, Rivas-González 2024, and our estimates (177×10^3 , 167.4×10^3 , 192×10^3 , respectively). The N_e of the African ape ancestor was higher than previous estimated (138×10^3 vs. 107×10^3 and 101×10^3) and could be due to an underestimation of NABC by coalHMM coupled with a slight overestimation in TRAILS as shown in Fig. 2 of Rivas-González et al. 2024 (below). Overall, despite some discrepancy on N_e estimates, the overall pattern is consistent across analyses.

We suggest caution when mentioning the *Pan* ancestor N_e because the inferred ILS is slightly higher than 5%, and therefore accompanied by some uncertainty, when estimating ILS, regardless the exploited methodology developed so far.

(Fig. 2 in Rivas-González et al. 2024)

We agree with the reviewer that using a constant mutation rate across species in the MSMC analysis could introduce some bias. Therefore, we are now also presenting the analysis using species-specific mutation rates in **Fig. ILS.S8**. The mutation rates were extracted from Rivas-González et al. 2023^{24,14} and Carbone et al. 2014^{14,15} (**Fig. ILS.S8**). The resulting demographic histories are only slightly different to those obtained using a constant mutation rate, with the only difference of a reduced population size for *Pongo abelii* and an increased one for *Symphalangus syndactylus*.

Figure ILS.S8. Effective population size trajectories through time as inferred by msmc2. A) Constant mutation rate 1.25×10^{-8} bp*gen. B) Species-specific mutation rates were used, as indicated in legend.

We added the following paragraphs:

“When species-specific mutation rates were used^{29,31}, a reduced population size for *Pongo abelii* and an increased one for *sympalangus syndactylus* were observed.”

We also shortened the ILS discussion to highlight some of the most relevant findings including putting these into context (i.e., differences that are significantly different from previous literature):

“Focusing on those segments that could be reliably aligned, we estimated speciation times and modeled incomplete lineage sorting (ILS) across the ape species tree (**Fig. 2b, Table.ILS.S1**)¹⁹. Our analyses date the human–chimpanzee split at 5.5–6.3 mya (minimum to maximum estimate of divergence), the African ape split at 10.6–10.9 mya, and the orangutan split at 18.2–19.6 mya. We infer ILS for an average of 39.5% of the autosomal genome and 24% of the X chromosome. Compared to recent reports²⁰, this represents a 7.5% increase in autosomal ILS, in part due to inclusion of more repetitive DNA (**Fig. ILS.S1**). Consequently, we estimate that the human–chimpanzee–bonobo ancestral population (average $N_e=198,000$) is larger than that of the human–chimpanzee–gorilla ancestor ($N_e=132,000$), consistent with an increase of the ancestral population 6–10 mya (**Supplementary Note VI**).”

L314: What do the ranges represent?

The ranges represent the minimum and maximum values across the HCGO phylogenies.

We rephrased the paragraph as follows:

“Our analyses date the human–chimpanzee split at 5.5–6.3 mya (minimum to maximum estimate of divergence), the African ape split at 10.6–10.9 mya, and the orangutan split at 18.2–19.6 mya (**Fig. 2a**).”

L317: What is the basis for the claim that increased ILS is due to repetitive regions? If this refers to observations on the HLA, do the authors observe a difference in genome-wide estimates after excluding it, or other previously unresolved regions?

In order to show the role of repetitive regions in our ILS inference, we iteratively removed 1000 bp windows overlapping with common repeat elements in HCGO quartet inference. When all the repeats-containing windows are excluded, the inferred ILS is 37.7%, which is consistent with those estimated in Rivas-González et al. 2023 using the alignments from Shao et al. 2023. When we excluded repeats longer than 100 bp and 1000 bp, the inferred ILS proportion is 37.6% and 38.5%, respectively, increasing to 39.4 when repeats longer than 10^5 bp are removed (**Fig. ILS.S6**). These results suggest that repetitive regions may play a role in the increased ILS proportion inference.

We added the following paragraph in the Supplementary Note:

“Furthermore, we explored the role of repetitive regions in increased ILS estimation in HCGO, when compared with previous estimates²⁹. In doing so, we iteratively excluded 1000bp windows overlapping with the T2T-CHM13 repeat track, considering different repeat sizes. When all the repeats-containing

windows are excluded, the inferred ILS is 37.7%, which is consistent with those estimated in Rivas-Gonzales et al. 2023²⁹ using the alignments from Shao et al. 2023²⁶. When we excluded repeats longer than 100 bp and 1000 bp, the inferred ILS proportion is 37.6% and 38.5% respectively, increasing to 39.4 when repeats longer than 10^5 bp are removed (**Fig. ILS.S6**). These results suggest that repetitive regions account for ~2% of the ILS increase consistent with selectively neutrally evolving DNA.”

Figure ILS.S6. Estimates of ILS proportion (x-axis) in HCGO when excluding 1000 bp windows overlapping with repetitive regions of different sizes (y-axis). Numbers in the y-axis refer to the lower bound of the excluded repeats. Confidence intervals refer to standard deviations inferred across 1000 bp windows.

L343: Could the authors please clarify how the non-SD-based differences were defined, i.e. if they were detected based on the alignment or a lack of detectable orthologous sequence for the NHP-based gene model? Can anything be stated about the function in either category (lineage-specific SD or others, presumably human losses?)

Non-SD-based differences included significant changes in the isoform or gene models, including transcripts that differed significantly in length from the human model. For example, we found that in 73% of cases, the non-SD differences involved differences over more than 50% of the

length of the predominant protein model in human. In the remaining 27% of cases, the predicted gene resembles a lncRNA and may in fact be spurious. We added more descriptions of the nature of the non-SD-based novel genes as follows:

“We identify a fraction (3.3–6.4%) of protein-coding genes present in the NHP T2T genomes that contain new transcript models relative to human annotation. This includes 770–1,482 novel gene models corresponding to 315–528 gene families in the NHPs with ~68.6% corresponding to lineage-specific SDs, all supported by Iso-Seq transcripts (**Table. Gene.S1, S2**). In this list, we also find biomedical genes such as *LRPAP1* whose tissue-specific expression is suggested in **Supplementary Note VIII**, as well as human-specific genes including *NOTCH2NL*²⁹, *SRGAP2C*^{30,31}, and *ARHGAP11*³² (**Table. Gene.S7**). The non-SD overlapping gene copies consist of 73% of transcript models that are changed in their sequence by >50% with the remaining 27% no longer translated.”

L361: “We now estimate..” do these estimates differ from prior ones across repeat categories, outside satellite/VNTRs?

We compared RepeatMasker output for each T2T assembly presented here to the next best assembly, determined based on the number of contigs and gaps, and estimated N50 in assembly metrics. We find that each T2T genome has a large increase (between 286 and 706 Mbp) in repeat annotations (**Table Repeat.S2**).

The following sentence has been added to the text:

“Compared to previous genome assemblies, repeat content has increased from 286 to 706 Mbp depending on species (**Table Repeat.S2**).”

Table Repeat.S2. RepeatMasker whole-genome summary comparison to previous ape genome assemblies.

		Chimpanzee		Bonobo		Gorilla		Sumatran orangutan		Bornean orangutan		Siamang	
		T2T assembly	previous assembly	T2T assembly	previous assembly	T2T assembly	previous assembly	T2T assembly	previous assembly	T2T assembly	previous assembly**	T2T assembly	previous assembly**
Class	Subclass	Mbp	Mbp	Mbp	Mbp	Mbp	Mbp	Mbp	Mbp	Mbp	Mbp	Mbp	Mbp
DNA	Unclassified	1.18	1.17	1.18	1.23	1.18	1.15	1.18	1.24	1.19	1.16	1.15	1.17
	Crypton	0.05	0.05	0.05	0.05	0.04	0.04	0.05	0.05	0.05	0.05	0.05	0.05
	Crypton-A	0.02	0.02	0.02	0.02	0.02	0.02	0.02	0.02	0.02	0.02	0.02	0.02
	Kolobok	0.07	0.07	0.07	0.07	0.07	0.06	0.07	0.07	0.07	0.06	0.07	0.07

	MULE-MuDR	0.99	0.96	0.99	0.88	1.00	0.93	1.04	0.89	1.01	0.95	0.91	0.95
	Merlin	0.04	0.04	0.04	0.04	0.04	0.04	0.04	0.04	0.04	0.04	0.04	0.04
	PIF-Harbinger	0.07	0.07	0.07	0.07	0.07	0.07	0.07	0.07	0.07	0.07	0.07	0.07
	PiggyBac	0.55	0.54	0.55	0.53	0.55	0.52	0.55	0.54	0.55	0.53	0.51	0.54
	TcMar	0.03	0.03	0.03	0.03	0.03	0.03	0.03	0.03	0.03	0.03	0.03	0.03
	TcMar-Mariner	3.10	2.85	3.08	2.83	4.57	2.8	3.61	2.85	3.59	2.85	2.78	2.85
	TcMar-Pogo	0.00	0.00	0.00	0.00	0.00	0	0.00	0.00	0.00	0.00	0.00	0.00
	TcMar-Tc1	0.14	0.14	0.14	0.14	0.14	0.14	0.14	0.14	0.14	0.14	0.13	0.14
	TcMar-Tc2	1.69	1.64	1.68	1.64	1.67	1.6	1.66	1.64	1.66	1.63	1.69	1.66
	TcMar-Tigger	38.11	37.12	38.15	37.00	38.21	36.41	38.27	37.26	38.31	37.15	37.70	37.82
	hAT	0.80	0.54	0.81	0.53	0.55	0.53	0.54	0.53	0.53	0.53	0.52	0.54
	hAT-Ac	0.64	0.63	0.64	0.63	0.65	0.62	0.64	0.62	0.65	0.62	0.63	0.62
	hAT-Blackjack	2.56	2.47	2.57	2.41	2.51	2.38	2.48	2.44	2.48	2.43	2.41	2.46
	hAT-Charlie	47.68	46.24	47.76	46.04	47.77	45.05	47.94	46.23	47.65	46.02	45.74	46.88
	hAT-Tag1	0.48	0.48	0.48	0.47	0.48	0.46	0.48	0.47	0.48	0.47	0.48	0.49
	hAT-Tip100	12.40	12.05	12.42	12.01	12.28	11.78	12.37	12.06	12.31	11.99	11.83	12.1
	hAT-hAT19	0.00	0.00	0.00	0.00	0.00	0.00	0.00	0.00	0.00	0.00	0.00	0.00
	Total	110.60	107.11	110.75	106.62	111.82	104.63	111.20	107.19	110.83	106.74	106.76	108.51
LINE													
	CR1	10.84	10.75	10.85	10.69	10.88	10.48	10.95	10.71	10.91	10.61	10.50	10.34
	Dong-R4	0.12	0.12	0.12	0.12	0.12	0.12	0.13	0.12	0.13	0.12	0.12	0.12
	I-Jockey	0.02	0.02	0.02	0.02	0.02	0.02	0.02	0.02	0.02	0.02	0.02	0.01
	L1	517.20	500.80	518.83	494.94	528.48	488.31	567.18	525.47	564.30	527.64	514.02	455.39

	L1-Tx1	0.05	0.05	0.05	0.05	0.05	0.05	0.05	0.05	0.05	0.05	0.05	0.05
	L2	104.31	102.67	104.50	102.33	106.09	100.07	104.05	102.50	104.11	101.71	103.03	99.58
	Penelope	0.07	0.07	0.07	0.07	0.07	0.07	0.07	0.07	0.07	0.07	0.07	0.06
	RTE-BovB	0.87	0.86	0.87	0.86	0.88	0.85	0.86	0.86	0.86	0.85	0.83	0.8
	RTE-X	3.20	3.17	3.21	3.15	3.23	3.11	3.18	3.16	3.19	3.14	3.28	3.14
	Total	636.69	618.51	638.51	612.23	649.81	603.08	686.49	642.96	683.63	644.21	631.90	569.49
LTR	Unclassified	3.18	3.15	3.19	3.14	3.23	3.1	3.20	3.15	3.20	3.11	3.27	3.17
	ERV1	86.66	83.33	87.01	80.47	86.78	79.28	88.08	80.81	87.57	80.33	78.19	79.15
	ERVK	9.29	8.53	9.23	7.74	9.05	7.52	10.14	7.91	9.83	7.91	7.97	7.42
	ERVL	59.59	57.69	59.95	57.29	60.94	56.6	60.55	58.30	60.57	57.92	55.58	57.68
	ERVL-MaLR	111.26	108.30	111.40	107.67	112.99	105.71	112.55	108.78	112.26	108.37	106.74	109.09
	Gypsy	4.92	4.80	4.91	4.79	4.87	4.71	4.84	4.79	4.85	4.74	4.74	4.72
	Total	274.90	265.80	275.70	261.10	277.85	256.92	279.36	263.74	278.28	262.38	256.49	261.23
RC													
	Helitron	0.46	0.45	0.46	0.45	0.46	0.45	0.46	0.45	0.46	0.45	0.44	0.46
	Total	0.46	0.45	0.46	0.45	0.46	0.45	0.46	0.45	0.46	0.45	0.44	0.46
Retroposon													
	SVA/LAVA	4.78	4.50	4.90	4.61	5.21	4.69	3.87	3.53	4.39	2.64	6.66	2.09
	Total	4.78	4.50	4.90	4.61	5.21	4.69	3.87	3.53	4.39	2.64	6.66	2.09
SINE													
	5S-Deu-L2	0.22	0.22	0.22	0.21	0.22	0.21	0.22	0.22	0.22	0.22	0.22	0.21
	Alu	308.65	298.34	310.50	295.87	314.39	281.8	314.48	297.69	313.11	303.01	321.85	345.29
	MIR	81.29	79.88	81.51	79.61	81.78	77.67	81.34	79.79	81.34	79.15	79.87	79.11

	tRNA	0.20	0.20	0.20	0.20	0.20	0.2	0.20	0.20	0.20	0.20	0.20	0.19
	tRNA-Deu	0.05	0.05	0.05	0.05	0.05	0.05	0.05	0.05	0.05	0.05	0.05	0.04
	tRNA-RTE	0.60	0.55	0.61	0.55	0.56	0.54	0.56	0.55	0.56	0.54	0.55	0.52
	Total	391.01	379.24	393.08	376.49	397.19	360.47	396.85	378.50	395.47	383.17	402.73	425.36
Other		2.21	2.03	2.61	1.29	2.15	1.1	2.23	1.31	2.35	1.07	1.34	1.12
	Total	2.21	2.03	2.61	1.29	2.15	1.10	2.23	1.31	2.35	1.07	1.34	1.12
Total Interspersed		1,420.65	1,377.59	1,425.99	1,362.79	1,444.50	1,331.34	1,480.45	1,397.69	1,475.41	1,400.64	1,406.33	1,368.25
Low_complexity		6.66	6.13	6.63	5.97	8.78	5.7	6.78	6.27	6.80	6.16	6.30	6.32
Satellite		171.04	6.71	197.06	5.32	338.25	3.87	14.24	3.93	14.53	6.11	4.77	3.11
	Y-chromosome	68.81	9.35	0.00	0.00	0.00	0.00	0.00	0.00	0.00	0.00	0.00	0.00
	acro	7.05	0.61	9.03	0.59	4.00	0.57	2.40	0.74	3.90	0.76	0.88	0.75
	centr	4.70	1.44	64.33	8.99	120.17	7.84	174.74	7.09	140.69	28.42	370.76	8.39
	subtelo	0.08	0.02	0.09	0.03	0.07	0.01	0.07	0.02	0.08	0.01	0.30	0.09
Simple_repeat		62.51	36.07	101.06	36.19	172.58	33.43	127.32	36.98	132.19	46.07	47.99	38.99
	Total	320.85	60.33	378.20	57.09	643.86	51.42	325.54	55.03	298.19	87.53	431.00	57.65
RNA													
	rRNA	0.48	0.21	0.46	0.16	0.31	0.15	0.81	0.15	0.52	0.19	0.32	0.17
	scRNA	0.19	0.18	0.19	0.18	0.19	0.17	0.18	0.17	0.18	0.18	0.19	0.19
	snRNA	0.47	0.44	0.46	0.44	0.48	0.43	0.47	0.45	0.47	0.45	0.47	0.48
	srpRNA	0.30	0.29	0.30	0.29	0.30	0.27	0.30	0.28	0.30	0.29	0.28	0.32
	tRNA	0.12	0.10	0.11	0.10	0.11	0.09	0.12	0.10	0.12	0.10	0.11	0.1

	Total	1.55	1.22	1.53	1.17	1.38	1.11	1.89	1.15	1.59	1.21	1.36	1.26
Total		1,743.04	1,439.15	1,805.73	1,421.05	2,089.74	1,383.88	1,807.88	1,453.90	1,775.19	1,489.38	1,838.69	1,427.17

The previous assemblies for the apes are *Pan paniscus* (GCA_013052645.3; long-read + short-read assembly), *Pan troglodytes* (GCF_002880755.1; long-read + short-read assembly), *Gorilla gorilla* (GCF_008122165.1; long-read + short-read assembly), *Pongo abelii* (GCF_002880775.1; long-read + short-read assembly), *Pongo pygmaeus* (GCA_023767775.1; long-read assembly), *Symphalangus syndactylus* (GCA_028642525.1; short-read assembly). Only fasta sequences associated with a chromosome were considered.

** no chromosome-level assembly was available for these species; all scaffolds/contigs were considered.

L379: Did the authors observe any differences in methylation status of these lineage-specific MEIs compared to similar shared ones, is there any evidence for differences in transcriptional activity?

To address this question, we first identified all heterozygous L1 and *Alu* elements from the generated species-specific CACTUS unaligned callset (all elements were selected in hs1 as this genome only had the one haplotype). The heterozygous elements were chosen as they represent the youngest insertions, and SVA elements were not analyzed as these elements are not shared across all primate species. Transduced sequences were trimmed off of the ends of the L1 insertions. Next, we found L1 and *Alu* elements that were shared across all primate species by extracting full-length LINE1 and *Alu* elements via RepeatMasker (v4.1.5) and BLAT.

We then examined DNA methylation levels of these elements (siamang was not compared as this was used as our outgroup in the species-specific analyses). *Alu* elements are more highly methylated than L1 in all species. We then compared DNA methylation levels of species-specific elements versus those that are shared by all species. To avoid bias due to the small number of CpGs, elements with fewer than four CpGs were excluded from the analysis. The total number of elements compared in each species are shown for species-specific versus shared elements. **Our comparisons demonstrate that in all species, species-specific elements are significantly more methylated than the shared elements (Mann-Whitney tests, P-values shown in the figure below).** This observation mirrors that observed in the comparison of recent versus old gene duplications¹⁶ and LINEs delineated by age in the human genome¹⁷, indicating epigenetic silencing of recent MEIs. With respect to transcriptional activity, TEs are not detected in Iso-Seq nor RNA-seq data; an approach such as PRO-seq, which has not been performed for this study, would be required. We hope to examine their transcriptional activities in future studies.

Figure Met.S3. DNA methylation differences between species-specific elements and elements shared between all species. Results from (top) Alus and (bottom) L1 elements demonstrate that species-specific elements are significantly more highly methylated compared to shared elements (Mann-Whitney tests, P-values show in each panel).

L391: Is there any evidence for competition, i.e. that the difference between L1 and Alu activity is related to each other?

Overall, *Alu* activity is thought to be constrained by the number of intact LINE-1 ORF2p sequences, as this protein is required for its mobility. However, the lack of activity of *Alu* transposons in the orangutan genome does not correlate with SVA activity, which is another SINE that relies on LINE-1 ORF2p. The lack of this correlation between SVA and *Alu* activity in the orangutan lineage potentially indicates either activity or lack of activity of host factors¹⁸ or species-specific LINE-1 elements in this genome having higher *cis*-preference (i.e., LINE-1 mobility over *Alu* transposition). Therefore, variation in LINE-1 elements with intact ORFs, *cis*-preference, and host factors are the most likely explanations for species-specific activity differences, and a less likely explanation would be direct competition between LINE-1 and the SINEs that it mobilizes. Directed research with functional experimentation would be needed to test these hypotheses.

L434: The scales for Ne and time are missing.

We rephrased as follows to include this information:

“Demographic inference. Black and red values refer to speciation times (in million years) and effective population size (Ne, in thousands), respectively.”

L405, L505, supp P60: The authors mapped a diversity panel on NHP to the new assemblies. As most of the improvements within them are due to structurally complex regions such as duplications that are now resolved, it is not evident how they can be reliably genotyped with short reads. Calls that are the result of ambiguous mappings from e.g. duplications have the potential to result in biased estimates of diversity and allele frequencies that the selection scans are based on. The description of the filtering criteria is sparse and vague, e.g. ‘applying standard quality control to SNPs and indels’, ‘bi-allelic sites that fell within high-confidence regions’, but don’t provide definitions of standard quality control or high-confidence regions. Without knowing how the authors ensured correct variant calls and associated summaries in structurally challenging/multicopy genes, and how callable regions were defined in the assemblies, it is not possible to evaluate if the results they present can be considered reliable.

We apologize for not being more explicit. As part of this analysis, we specifically excluded regions where mappings would be ambiguous. To restrict our analysis to the 'accessible' genomic regions, an established protocol was used (<https://www.illumina.com/science/genomics-research/articles/identifying-genomic-regions-with-high-quality-single-nucleotide-.html>) and a short-read accessibility mask for the T2T reference genomes was performed. This mask incorporates three alignment metrics: high-quality bases with normalized coverage within 25% of the median autosomal coverage, positions with a mapping quality score of 50 or higher, and regions where $\geq 90\%$ of base calls have a Q20 score or above. This strategy was used to improve variant calling in the human T2T-chrY study. The sizes and percentages of the accessible regions in the genomes of five great ape species are detailed below.

Species	Size (bp) and percentage (%) of accessible regions
Gorilla	2,562,157,238 (72.3%)
Chimpanzee	2,289,823,041 (72.1%)
Bonobo	2,207,949,129 (68.1%)
S. orangutan	2,079,949,089 (63.8%)
B. orangutan	2,049,638,495 (63.6%)

We added these additional details to the methods of the supplement:

“Variant identification followed the T2T-chrXY ape paper method using GATK v4.4.0.0 HaplotypeCaller³² for initial calling, GenotypeGVCFs for joint genotyping, and applying standard quality control to SNPs and indels. **To further improve variant calling and genotyping accuracy, we restricted our analysis to the ‘accessible’ genomic regions. Following an established protocol (<https://www.illumina.com/science/genomics-research/articles/identifying-genomic-regions-with-high-quality-single-nucleotide-.html>), we performed a short-read accessibility mask for the T2T reference genomes. This mask incorporates three alignment metrics: high-quality bases with normalized coverage within 25% of the median autosomal coverage, positions with a mapping quality score of 50 or higher, and regions where $\geq 90\%$ of base calls have a Q20 score or above.**”

L467: The definition of “ape-specific” to exclude human is awkward. It might be more appropriate to call them ‘genes absent in human’ or something else, as they are neither present across apes, nor does the description suggest that specificity to apes was tested via an outgroup. Though there seems to be a macaque in the lower panels of Fig 3d, a description of its inclusion and treatment and reference seems completely absent.

We rephrased “ape-specific” to “species-specific genes”:

“These large-scale differences frequently correspond to **species-specific genes** (those that comprise phylogenetic clades of genes unique to either bonobo, gorilla or both orangutan species) (**Extended Fig. 2a; Fig. IG.S1c**). We observe the greatest number of **species-specific genes** within IGH (**Fig.IG.S1d**), ...”

Macaque was added in the divergence time estimate analysis in previous Fig. 3d (**Extended figure 3c**) panel to consider the deep coalescent time of the MHC locus. We added the description of its inclusion in the legend as follows:

“The phylogenetic tree includes the macaque genome (MFA)⁹⁴ to better estimate the deep coalescence time observed at this locus.”

L498: How do the coalescent times estimated for these loci compare to those estimated by the trails analysis for the same region?

Although TRAILS can estimate parameters using large genomic regions such as chromosomes, it does not provide exact estimates of coalescent times using posterior decoding. In fact, we can infer in which time interval in AB and ABC species the coalescent occurred. In our analysis we used nAB and nABC to 3. Nevertheless, we can use them together with the tree topology to compare ILS results to coalescent times inferred by IQ-TREE. Our analysis shows a relatively high degree of ILS for the three out of four variable regions, with the exception of region IV. We report the new results in the following paragraph in the Supplementary Note:

“Moreover, we focused on the four HLA variable regions at a base-pair scale considering the tree topology and the first coalescent event (**Fig. ILS.S5**). Three of the variable regions (I, II, and III) show elevated levels of ILS, with the only exception of region IV, confirming their high variability. For region I, 88% of the analyzed sequence was compatible with a topology in which human and gorilla are sister groups, and the first coalescent is shallow, consistent with the reconstructed phylogeny in **Extended Fig. 3c** and **Fig. MHC.S10**, suggesting a possible balancing selection event. In region II, 36% of the sequence is consistent with an HC topology, with the remaining regions equally distributed. In the remaining two regions, we observed 55% (region III) and 71% (region IV) of the region topology consistent with HCGO.”

Figure ILS.S5. Treemaps of the topology inferred for the four HLA variable regions. The topology is indicated in the middle of each rectangle. We are showing four possible topologies: human-chimp (HG recent) with shallow first coalescent, human-chimp with deep first coalescent (HG ancient), chimp-gorilla (CG), and human-gorilla (HG). Labels on the right corner refer to the first coalescent deepness.

L499: How was missing data in the alignment for this analysis dealt with (both row/haplotype-wise and position-wise), given the high degree of polymorphism in this region? Which 3 topologies represent 96% of the region, and how well are they supported?

The regions with sequence present in all ape haplotypes were used, retaining the 76% of the position-wise bins. We added this detail to the supplementary note:

“We extracted homologous sequences in 500 bp increments and optimized local multiple sequence alignments (8,259 bins for 500 bp) using MAFFT and then concatenated to generate 5 kbp regions (953 bins); **the regions with no missing sequences were used retaining 76% of position-wise bins.**”

For demonstrating the support values, we generated a violin plot with average bootstrap values (92.49, 89.08 and 90.29) of the first three topologies, 1, 2 and 3, respectively (**Fig. MHC.rebuttal 1**).

Figure MHC.rebuttal 1. The distribution of average bootstrap values for the first three most prominent topologies in the MHC region. The black crossbar are the mean values.

L533: What does first-generation mean here?

We rephrased the term “first-generation” to “first” to avoid confusion.

L543: There does seem to be no mention of 3D chromatin organization in Supplementary Notes XII-XIII, or anywhere else (other than phasing the haplotypes).

We revised the sentence as follows:

“We overlay these data with a multiscale epigenomic map of the apes, including DNA methylation and replication timing (**Supplementary Note XVI-XVII**).”

L537 & Sup. P96: The main text states that the authors distinguish between hyper- and hypomethylated promoters, but the methods only mention a distinction between consistently and variably methylated promoters, regardless of whether methylation levels are high or low.

For the analyses, the authors establish a threshold of 15% divergence in methylation status to report that ~17% of promoters are variably methylated across species. Neither the rationale for the threshold nor the resulting number of differentially methylated promoters they expect to observe by chance are evident. The absence of a statistical framework or its description does not enable the reader to contextualize the results. The same applies to the 25% threshold established to define lineage-specific methylation. The referenced BioProject PRJNA101639 appears to be associated with a study on gene expression in honeybee antennae. How was callability across complex regions dealt with for the short-read mappings?

Thank you for highlighting this point. We added a more detailed description of our methods in the revised manuscript. To identify variably methylated promoters, we relied on the Central limit theorem to identify promoters whose methylation change exceeded 2 standard deviations from the mean, therefore identifying statistical outliers. We previously used the 15% cutoff value, which approximated the distribution of all species, but now we developed species-specific thresholds, which are also presented in the supplementary file. Species-specific gain or loss of 5mC is defined as a case where only one species is classified as an outlier. To demonstrate the validity of our metrics, we included a visualization of the actual 5mC levels across the species. We also included additional details on the numbers of hyper and hypomethylated promoters in the new supplementary figures. We apologize that the previously referenced BioProject missed the last digit; we updated it to the correct number PRJNA1016395.

The comment regarding callability for the short-read mappings was discussed in the earlier comment on variant calls for “L405, L505, supp P60.”

L573: While previously undescribed events are reported, the recall rate for previously described inversions across the sources they compare is not. If there are differences, what do they attribute them to?

We find that 50% of previously reported inversions larger than 100 kbp (169/339) are recalled in this study. To further explore this difference, we restricted the analysis to inversions ≥ 1 Mbp in size, resulting in a higher recall rate of 69.3% (79/114). Upon examining the 35 inversions not detected in this study, we identified 24 cases with differing breakpoints. The majority of these (n=18) involved multiple smaller inversions that had previously been called as a single large inversion (**Fig. INV.rebuttal 1**) but were more accurately resolved here. In the remaining cases (n=6), the differences in inversion size and breakpoints likely stem from limitations in the resolution of previous methods used to identify inversions.

We also identified nine cases where the previous inversion calls were not detected in our alignment plot, which may be attributed to individual differences between the samples analyzed or previous errors (example shown in **Fig. INV.rebuttal 2**). The remaining two cases involved nested inversions that were undetected in the current study.

Chromosome	Start	End	Inversion	Study	Note
chr1	214800749	239344531	GGO24	Porubsky2020_GA	Two inversions called as a single

Chromosome	Start	End	Inversion	Study	Note
chr7	68484254	78281438	PPA160	Porubsky2020_GA	different breakpoint

Figure INV.rebuttal 1. Example of multiple smaller inversions called as one larger inversion.

Chromosome	Start	End	Inversion	Study	Note
chr2	100006662	102708270	1_chimpanzee	Kronenberg2018_GA	Individual difference

Figure INV.rebuttal 2. Example inversions undetected in the genome assemblies.

We add the recall rate description of the previously reported inversions in the main text for fair comparisons as follows:

“... Focusing on events larger than 10 kbp, we curate 1,140 interspecific inversions—522 are newly discovered^{7,43-65} (Table INV.S2). **For a more fair comparison with previous inversion calls, we also computed recall rates (<1Mbp) and found that 69.3% (79/114) of the previously reported inversions are confirmed here. Among the 35 cases we missed, 24 correspond to change of inversion breakpoints, which were actually recalled and better resolved in this study, nine represent individual differences or previous error, and the remaining two were where our inversion call missed smaller nested inversions.** Assessing the genotypes of the inversion calls, we find 632 are homozygous (found in both the assembled ape haplotypes) ...”

L593: The authors do not report what the r^2 value corresponds to, but to set an expectation for the number of inversions on the human lineage some kind of phylogenetic comparative method probably needs to be applied, rather than an OLS linear model.

Based on maximum parsimony, each inversion event was assigned to a phylogenetic branch, and the r^2 value was computed by scaling the number of inversions with branch length.

For the number of inversions on the human lineage, we also checked human-specific inversions using Bornean orangutan as the reference, to remove potential reference bias.

For clarity, we edited the main text as follows:

“The number of inversions assigned to each branch, based on maximum parsimony, correlates with the length of the branch ($r^2=0.77$) ...”

L617, Sup P108: The methods description on the AQER analyses is difficult to follow, leaving many questions that result in an incomplete understanding of the analyses. I suspect the diploid 16-way cactus alignment was used here, but this is not evident from the text, which suggests that previously unnamed pairwise human-other species alignments were generated for this analysis. If that is the case, please include a full description. For example, what does the scoring threshold refer to, how was the ancestral sequence reconstructed and how were base probabilities calculated, was the ancestral sequence defined individually for each species as suggested by the statement regarding including the extant species' allele? I'm also curious to know how the diploid haplotypes were dealt with when counting variants, and how gaps/missing data were dealt with, particularly for the larger windows.

Somewhat alarmingly, the methods suggest that a hard threshold of >29 substitutions / 500bp window between the ancestral genome and focal extant species was used to define windows with significantly more divergence than expected, without explaining what this expectation or claims of significance are based on, and that this threshold was used across windows and species. If correct, this raises serious concerns about the validity of the analyses, as the expected number of substitutions will be strongly influenced by the sequence composition of the window, the branch length between the ancestral node and extant species, but likely also large-scale regional differences in substitution rates along chromosomes (see e.g. Mikkelsen et al. 2005, Fig 2), all of which need to be considered. While the chosen threshold might happen to be conservative enough to capture AQERS across the different scenarios, this would probably yield differing detection sensitivities across species rendering any quantitative claims on enrichment resulting from subsequent comparative analyses invalid.

The reviewer makes a number of suggestions on how we can improve this section of the paper. We took this feedback into consideration to improve our Methods section, the way we define nonhuman AQERs, and included additional analyses. The main points the reviewer makes are that: (1) we relied too much on our previous publication of the HAQERs to describe the methods and we need to include more detail in the supplement of this paper, (2) we did not sufficiently explain how we take the different branch lengths into account (e.g., human-chimp ancestor to humans versus human-chimp ancestor to chimps), and (3) we did not consider that changes in branch length along the chromosome may influence our set of AQERs.

We address each of these three points individually below:

1) The expanded Methods section now reads:

“We defined highly divergent regions in four great ape lineages by calculating the number of mutations that likely occurred in 500 bp regions between inferred ancestral nodes and extant genomes. The four lineages we analyzed were the inferred human–chimpanzee ancestor to the human (hs1), chimpanzee (primary haplotype), and bonobo (primary haplotype) genomes, as well as the inferred human–gorilla ancestor to the gorilla (primary haplotype) genome. We term these regions ancestor quickly evolved regions (AQERs), with the individual sets being (human) HAQERs, chimp-AQERs, bonobo-AQERs, and gorilla-AQERs. The identification of AQERs is not limited to conserved regions, but rather screens the entire genome, including elements that descended from previously neutral regions. This distinguishes HAQERs from many other searches for the genetic underpinnings of uniquely human traits, which

focused on human-specific divergence in conserved genomic regions and reflect modifications of existing functional elements. AQERs were identified in three steps. First, we aligned both haplotypes from the nearly complete great ape assemblies (<https://github.com/marbl/Primates>) to the T2T human assembly (hs1) with LASTZ²³. We chained the local alignments together with the utilities used by the UCSC Genome Browser group¹⁵³, but with stricter parameters^{154,19}. We used stricter scoring and filtering parameters because our method is sensitive to false-positive alignments; misalignments can often appear as regions of rapid divergence. We filtered these chained alignments to retain alignment fragments scoring greater than 60,000 (approximately 20 kbp of matches). We then used these single-coverage chains to generate a multi-species genome-wide alignment with MultiZ¹⁵⁵. For each of the four lineages being analyzed for highly divergent regions, we inferred the sequence of the ancestral node in a conservative fashion. Specifically, we used the gene annotations made by NCBI RefSeq for hs1 (<https://hgdownload.soe.ucsc.edu/goldenPath/hs1/bigZips/genes/hs1.ncbiRefSeq.gtf.gz>) to extract fourfold degenerate codon sites and estimate branch lengths for a fixed-topology tree using a Jukes-Cantor model of evolution by maximum likelihood. For each position in the alignment, if a base was determined to be present in the ancestral node according to the tree, i.e., a base is present in at least two species on two independent lineages connected to the ancestral node (lineages to parent node and two child nodes). This is equivalent to treating aligned bases as having a common origin. For bases that are present at a node, we reconstructed the probabilities of A, C, G, and T in the ancestral node using the estimated branch lengths and the value of the base position in extant species¹⁵⁶. To assign a single base to the ancestral node from these four probabilities, positions were assigned the value observed in the extant species unless the probabilities of the other three base values summed to greater than 0.8. This ensures that we only count substitutions where we are confident of the change occurring on the lineage being analyzed. Finally, the number of mutations that separate the inferred ancestral node and the extant genome was counted by sliding windows of 500 bp. Windows with significantly more divergences than expected were identified as HAQERs, chimp-AQERs, bonobo-AQERs, and gorilla-AQERs. To conservatively estimate significance, we use the fastest divergence rate in a 10 Mbp window of the genome as the estimate of the expected divergence rate when defining HAQERs. For the other lineages, we scale this conservative divergence rate relative to the ratio of branch lengths between the human branch, and the branch length of the other lineages from fourfold degenerate sites for chimp-AQERs, bonobo-AQERs, and gorilla-AQERs. The p-value of obtaining at least the number of divergences found in each 500 bp window along the assembly was calculated using a binomial distribution, where the number of trials is 500, and the probability of success is the expected divergence rate. This raw p-value was adjusted using the Benjamini-Hochberg procedure by counting the number of 500 bp windows in the genome and ranking their raw p-values. Windows with adjusted p-values of less than 3e-7 were identified as AQERs, which is equivalent to at least 29 mutations in a 500 bp window for most lineages analyzed (i.e., HAQERs, chimp-AQERs, and bonobo-AQERs), and at least 34 mutations in a 500 bp window for gorilla-AQERs, due to the longer phylogenetic branch from the human-gorilla ancestor to gorilla. Additionally, we ensured that chimp-AQERs, bonobo-AQERs, and gorilla-AQERs represent continuous sections of both the human and nonhuman primate genomes. Mangan, R. J. et al. offers a more detailed description of the methods we used¹⁹. We generated additional HAQER sets at different levels of spatial resolution by changing the window size of our sliding window-based HAQER ascertainment program. We generated additional HAQER sets at window sizes of 25, 50, 100, 200, 300, 400, 500, 750, 1 k, 2 k, 5 k, 10 k, and 20 kbp. Furthermore, we identified a repeat-free subset of 500 bp window size T2T HAQERs, which excludes HAQERs that overlap centromeres/satellites, SDs, simple repeats, or RepeatMasker annotations. Complementary to the repeat-free subset of HAQERs is the repeat subset of HAQERs, which consists of HAQERs that have any overlaps with these repeat annotations.

To analyze the relationship between AQER elements and regulatory elements, we calculated binomial-based overlap enrichments between AQERs and chromatin state annotations derived from the Roadmap Epigenomics Consortium^{157,158}. We calculated similar enrichments against RepeatMasker annotations developed for each assembly. Finally, to analyze the relationship between HAQERs and regions of incomplete lineage sorting (ILS), we calculated binomial-based enrichments between HAQERs and regions of the human genome with a posterior probability of at least 0.6 for each TRAILS hidden state.”

2) The reviewer correctly states that we did not explicitly adjust the threshold for calling AQERs based on the relative branch lengths between the human branch and the branches to other species (i.e., chimpanzee, bonobo, gorilla).

Previously, we implicitly adjusted for the different branch lengths on each lineage by using the fastest rate of evolution in a 10 Mbp window as a conservative estimate of the expected rate of evolution on lineage. To improve our approach based on the reviewer's comment, we now explicitly scale the expected mutation rate on each nonhuman lineage relative to the branch length of that lineage when compared to the human lineage. We continue to use the fastest 10 Mbp as the expected mutation rate on the human lineage. We then use a tree based on fourfold degenerate sites in protein-coding exons to scale this conservative expected rate on the human branch to a conservative rate for the other lineages. This allows us to keep a conservative rate estimate, but also allows us to linearly scale this conservative rate across the branches. Due to the lineage from the human–chimp ancestor to humans, being very similar to the lineages from the same ancestor to chimp and bonobo, 29 remains the threshold number of mutations for a AQERs on those lineages. While the HAQER, chimp-AQERs, and bonobo-AQERs sets stayed the same in our manuscript, scaling by the gorilla branch raised the threshold of the gorilla-AQERs to 34 mutations in 500 bp, which reduced the set from 2710 to 1628 elements. We updated our analyses based on this small set of gorilla-AQERs, which does not have a large impact on the results, such as the tendencies for AQERs to overlap chromHMM states across tissues and cell types, or repeat regions:

3) The reviewer asks us to consider local/dynamic thresholding for HAQER significance across the genome, since the nonuniformity of mutation rates causes the neutral rate of evolution to vary across genomic positions. By extension, highly divergent regions ascertained with a constant threshold may have a contribution from regions with elevated mutation rates. However, we note that there is a non-independence between mutation rate and selection, and that many regions of high mutation rates, which are deprioritized in other divergence-based studies, can be under strong selective pressure²⁰. We define HAQERs as the most divergent regions of the human genome, agnostic to the underlying force, or forces, of divergence. Thus, we consider HAQERs to be a heterogeneous set, where the rapid divergence of some elements may be related to selection, while others are the product of elevated local mutation rates. Importantly, many HAQERs may be the product of a mixture of these mutually confounding forces.

To emphasize the reviewer's point, we analyzed the relationship between HAQERs and genomic regions that are likely to have local increases to the expected number of mutations. The two ways we approached this were to investigate the overlap between HAQERs and repeat regions, such as simple repeats, tandem repeats, and segmental duplications, which can have elevated mutation rates^{21,22}, as well as the overlap between HAQERs and regions of ILS, which can also have local effects on the expected number of mutations.

We identified a repeat-free subset of HAQERs, which excludes HAQERs that overlap centromeres/satellites, segmental duplications, simple repeats, or mobile elements. This subset should have a reduced contribution from the repeat-associated increase in mutation rate. The "repeat-free" subset shows a stronger enrichment for overlapping the functional categories of bivalent enhancers and promoters, and the complementary set shows reduced enrichment for these gene regulatory states. These results are consistent with the repeat-free HAQERs having a lesser contribution from elevated mutation rates, and potentially a greater contribution from selection acting on their functional effects.

Regarding ILS, we used the TRAILS analysis to identify genomic segments associated with the four states: V0, corresponding to the standard great ape topology; V1, which shares a topology with V0 but identifies regions with deep coalescence times; V2, which captures human–gorilla ILS; and V3, which captures chimp–gorilla ILS. A genomic segment is associated with one of the four states if it has a posterior probability of 0.6 or greater for that state. We observed a significant depletion between HAQERs and the V0 state and an enrichment in the V1 state. This suggests that non-uniformity in coalescence times across the genome may be an additional consideration for analyzing human divergence rates. However, we note that the V1 state does not measure imbalances between the branch length separating the human from the human–chimp ancestor and separating the chimp from the human–chimp ancestor, which would reflect human-specific divergence. We note only minor enrichments between HAQERs and regions with non-standard topology. This may be a consequence of our conservative thresholding for ancestral state inference, which ignores position with substantial uncertainty in the human–chimp ancestral state, as would be expected in positions associated with ILS.

We have included these two analyses of HAQERs that overlap repeat regions with likely increased mutation rates, and regions of ILS that may have deep coalescent times to emphasize to readers that there are many forces and scenarios that can contribute to elevated interspecies divergence.

L620: Why was the orang excluded?

To analyze the lineage leading to orangutan, we need to be able to confidently estimate the human–orangutan ancestor, which we could not accomplish to the same degree as we could for the human–chimp or human–gorilla ancestor, for multiple reasons. 1) The lineage leading to orangutan is more than twice as long as the other four lineages examined. Already on the human lineage some regions, such as centromeres, change so rapidly that they cannot be aligned with current methods, and therefore cannot be included in our analyses. This would be exaggerated in orangutan-AQERs, making it difficult to compare across the sets. 2) Only 73% of the bases in the orangutan assembly align to a base in the gibbon assembly, which is needed to confidently infer the base at the human–orangutan ancestral node. This is in comparison to the human lineage where 92% of human bases have a base from an outgroup species present. Overall, we do not think our alignment and ancestral inference methods are currently at the point where we can estimate the human–orangutan ancestor to the same degree of confidence that we can for the human–chimp or human–gorilla ancestral nodes.

L621: Could the authors please provide the full comparison with previously defined HAQERS, e.g. a Venn diagram?

We now include the full Venn diagram as a panel in the supplemental figure. There are a number of differences between the underlying data used to identify the previously defined set of

HAQERs and the set of T2T HAQERs beyond gapped and T2T assemblies. An important difference is that the T2T human assembly represents a different set of human haplotypes, often with different ancestry, from those used in hg38. Therefore, the 661 HAQERs that were not re-identified have some divergent mutations that are not fixed in humans, such that these same regions do not cross the significance threshold for calling a HAQER when using another sampling of human haplotypes. The 2,348 T2T HAQERs that were not previously discovered are a mixture of those that do not replicate broadly across human haplotypes (similar to the 661) and those in regions of the genome that could not previously be analyzed due to assembly gaps in humans and/or other great apes.

L624 / Fig. 2a: The statement does not seem to be supported by Figure 2a, which suggests there is an enrichment across NHP AQERs for bivalent promoters, albeit less than for HAQERs. Could the authors please provide summaries for the enrichments? What is the difference between this figure and Fig. AQER.S1c, which appears to provide different results for the same analyses?

It might be possible to ascertain if differences are due to human-based chromatin states by comparing them with NHP-based states (e.g. Garcia-Perez et al., 2021), should the authors wish to corroborate the claim. It might also be interesting to differentiate between preexisting CREs with a newly acquired function versus wholly new CREs by looking at their sequence constraint across other species.

1) We agree with the reviewer's observation that there is an enrichment for NHP AQERs across bivalent promoters, albeit less than for HAQERs. We updated the text to read:

“HAQERs also exhibit a significant enrichment for bivalent chromatin states (repressing and activating epigenetic marks) across diverse tissues, with the strongest enrichment being for the bivalent promoter state (3-fold enrichment, $p < 1e-30$) (**Fig. 4a; Table AQER.S1**)—a signal present to a lesser degree on the chimpanzee, bonobo, and gorilla lineages (1.3-fold, 1.5-fold, and 1.4-fold; $p < 1e-19$, $p < 1e-30$, $p < 1e-12$, respectively). This lesser enrichment may be due to the chromatin states being called from human cells and tissues (**Supplementary Note XIII**).”

2) The reviewer correctly points out that an older version of this figure panel was mistakenly included in the supplemental figure. We removed this panel from the supplemental figure.

3) Following this reviewer suggestion, we analyzed the relationship between AQER elements and chromatin states derived from functional genomic profiling of human and NHP LCLs (Garcia-Perez et al., 2021). While we observed enrichments between AQER elements and bivalent promoter states on average across Roadmap Epigenomic biosamples, we note that these enrichments were variable across cell types and tissues. In fact, HAQERs did not demonstrate an enrichment for the Bivalent Promoter state for LCLs from Roadmap (Biosample ID E116, $p=0.1798$). Consistent with this finding in the Roadmap data, we did not observe significant enrichments between AQER elements (from humans, chimps, and gorillas) and bivalent/poised enhancers found in LCLs from those same three species. While we thank the reviewer for this good idea, it seems that LCLs will not allow us to weigh in on this topic since they are one of the cell types that does not show this general enrichment for bivalent promoters.

4) We agree with the reviewer that intersecting with conserved regions is a good way to identify HAQERs that were already evolving under constraint at the time of the human–chimpanzee ancestor and are likely to represent preexisting CREs (as opposed to newly acquired CREs). There are 776 HAQERs that contain a conserved element as defined by PhastCons and the 100-way UCSC Genome Browser alignment, although the HAQER is often larger with unconserved flanking regions^{23,24}. These conserved elements include eight HARs, including HAR1^{25,26}. While we agree with the reviewer's high-level idea that a lack of conservation is consistent with wholly new elements, we think functional experiments (on the inferred ancestor) would be needed to solidify a claim of newly forged CREs, which we are conducting for our companion paper.

L645: The authors report 770-1420 protein-coding genes as specific to the NHPs and absent in humans in L342, but only 6 as restricted to humans here. I'm having difficulties reconciling these numbers, at face value it appears there should be a comparable rate of such events in human. In the current presentation, it is not clear if the different methods target conceptually different scenarios, as they appear to aim at the same thing. Is there any reason the authors report lineage-specific events only for NHPs in the annotation section, but human-specific events only for toga?

The 770-1420 are the genes with changed transcript model compared to human, based on the pairwise comparison, we find that 1209 genes were assigned to human relative to at least one ape genes. Whereas TOGA is a separate analysis assessing complete loss of sequence, via projection of gene annotation. The differences in numbers are majorly due to different criteria being applied by the two approaches as well as the level of annotation being compared (up to isoform-level vs. gene). To avoid the confusion, we rephrased the misleading term as follows:

“We identify a fraction (3.3–6.4%) of protein-coding genes present in the NHP T2T genomes that contain new transcript models relative to human annotation. This includes 770–1,482 novel gene models corresponding to 315–528 gene families in the NHPs with ~68.6% corresponding to lineage-specific SDs, all supported by Iso-Seq transcripts (**Table. Gene.S1, S2**).”

L806: 'evolutionary periods of time': I suggest rephrasing this, the current statement is also true for non-congeneric species.

We rephrase the statement as follows:

“Congeneric species of *Pan* and *Pongo* present an opportunity to assess the evolution of centromeric α -satellites over relatively shorter evolutionary time.”

L848: Panels d-f present variations of a similar plot that use the same color scheme, but the value ranges of panels d and f have been stretched by 2 or 3 times for different ranges, which seems visually misleading by exaggerating the differences in the former or compressing them in the latter. It is not clear which scale was used for panel d, nor what the additional effect of the various choices of range compressions are compared to a monotonic scale. A scale like those used in Fig. 8. b-c might be more appropriate.

We apologize for the confusion. All scales in panels d-f should have had the same values and color schemes (ranging from 70%-100% and colored from dark purple to dark red). These are now fixed and updated figure below:

L854: Please specify all tests used.

We add details of the tests used in the legend as follows:

“*, Two-tailed Wilcoxon matched-pairs signed-rank test, $p < 0.05$; n.s., not significant”

L905: This statement is certainly true for a comparison with WSSD, but part of the authors have additionally widely used alignment and assembly-based approaches to yield much higher estimates in prior releases. This seems like a more appropriate baseline, though both comparisons are absent. The methods state that calls with up to 70% of satellite content were kept, could they please explain the rationale behind this choice, as it seems this might not have

been included in prior approaches, together with the impact it has on the numeric comparison across species (including the monkeys) given the substantial increase in resolution and variation reported in Fig. 2c for this sequence class. I do not doubt that the claimed increase is true regardless of the analytical setup, but it seems important to understand if it is in part also driven by choices driving a change in the author's definitions of SDs. The method section on the definition of lineage-specific SDs would benefit from distinguishing between orthologs and paralogs, given the ambiguity of the term homology in this context. What does “SDs with sequence content (coverage >20%) changed” refer to regarding query and target sequence? Is my understanding correct that a 300bp MEI of one species in a 1kb ancestral SD would result in classifying it as lineage-specific. If so, this seems to be quite different from other scenarios such as an expansion. I appreciate that the classification of these events is challenging, it would be helpful if the authors could provide summaries of the different scenarios they describe in Fig. SD.S1 across species and internal nodes.

To be clear, our definitions of SDs have not changed across studies^{22,27,28} (1 kbp of aligned sequence >90% identity requiring seed alignments that are devoid of any common repeats as annotated by RepeatMasker before constructing final alignments). This includes applying the comparable filtering criteria (<85% of common repeats and <75% of tandem repetitive DNA in the final alignment). While this may seem generous, it is necessary because such sequences are frequently enriched in SDs or better said get caught up in the SD because of their association and presumably neutral effect on fitness. Compared to our previous study²⁸, we identified a greater yield of nonredundant SDs in the T2T genomes—typically 50-70 Mbp of DNA. This is because of the overall better representation of SDs in acrocentric and euchromatic duplication blocks where gaps frequently persisted in older primate assemblies.

What has changed, however, is our attempt to further classify SDs between species as expanded, new, or content-changed—and this is indeed tricky. The reviewer's interpretation of “SDs with sequence content changed” is correct. And the coverage threshold refers to the target genome; for each of the target genomes, we align with the rest of the species and identify SDs of the target species that contain target-specific sequences. It should be noted, however, that the vast majority of SD base pairs are part of very large alignments where a much more substantial difference (other than an MEI) would be required to be classified as such. With respect to orthology, new SDs are by definition non-orthologous while expanded and content changed are orthologous up to the changed portion of the alignment. We use the unique sequence flanking the SDs to define syntenic positions.

Following the reviewer's suggestion, we included **Fig. SD.S5** to summarize the SDs of different scenarios described by **Fig. SD.S1**.

Figure SD.S5. Summary of average lineage-specific SDs detected per haplotype. Note the quantification excludes acrocentric SDs. The left panel shows ancestral SDs that are shared among apes, representing the inner ancestral nodes. The center panel summarizes lineage-specific SDs. Species-specific SDs for *Pan* and *Pongo* are shown on the right panel, separately due to relatively short divergence times to latest common ancestors.

Please note excluding the acrocentric SDs, the **Fig. SD.S6** was also updated as follows:

FIGURE REDACTED

Figure SD.S6. Identification of lineage-specific SDs. a) Assignment of SDs (in Mbp) to ancestral and lineage-specific terminal branches based on content, location, and copy number differences (**Fig.SD1**). This excludes acrocentric SDs. Asian (dark yellow) and African (red) apes are compared using macaque (MFA) (lighter yellow) as an outgroup. b) Estimated divergence time (based on SNVs) correlates with SD accumulation ($R^2=0.927$) with notable outliers including siamang, macaque, and ancestral branches (e.g., BC= bonobo/chimpanzee, HBC= human/bonobo/chimpanzee ancestral node, etc.).

The annotations in the supplementary table are incomplete for SD status and lineage status and are missing the genotypes. Does SD diversity scale with SNV diversity, or do they behave differently?

We revised supplementary **Table SD.S3** to include genotype information. We added additional description to avoid confusion.

“*SD status shows whether the SD is new, expanded, or new and expanded compared to remaining species. Lineage indicates the phylogenetic lineage the SD is assigned to based on minimum parsimony; “other” represents the SDs that are unable to be assigned to a lineage. Variability of SDs indicate if the SD is haplotype-specific, variable in length, or highly variable in length. Overlapping novel transcripts

indicate the overlap with **Table Gene.S2**. Lastly, rearrangement breakpoints indicate whether the SD is located within 100 kbp from the inversions.”

Also, investigating the correlation between lineage-specific SDs versus k-mer branch length (estimated from mashtree), we find a positive correlation ($R^2=0.864$) among apes (**Fig. SD.rebuttal 1**), with siamang representing an outlier. We also find that the degree of heterozygosity is greater in Sumatran orangutans, which have the highest proportion of SDs that are flagged as variable by **Fig. SD.S7**. Although weaker, there is also a positive correlation between the number of heterozygous SNPs per Mbp and the proportion of variable SDs (R^2 of 0.397; **Fig. SD.rebuttal 2**) among the seven apes.

Rebuttal figure 1. Correlation between mashtree-branch length vs. number of nonredundant segmental duplications assigned to each branch.

Rebuttal figure 2. Correlation between heterozygous SNPs (excluding repeats) vs. proportion of variable SDs (hap-specific, variable length, highly variable in **Fig. SD.S7**).

L908: The definition is redundant, the Siamang is not a great ape.

We deleted the redundant information.

L911, Supp. P137: The builds used for monkeys are not referenced, the referenced table SD2 does not exist. It seems the SD estimates will be largely conflated with the potentially much lower assembly quality for non-ape genomes. Unless the authors reference unpublished T2T builds, the correct conclusion is that the assemblies of great apes have the highest SD content.

Thank you for pointing out the confusion with referenced data. The sentence was originally referencing **Fig. SD.S3 (previous Fig. SD.S2)**. We also added new **Table SD.S1** for the builds used for monkeys and the underlying data of **Fig. SD.S3**.

Table SD.S1. Segmental duplication content in current and previous assemblies.

Name	Haplotype	Assembly ID	Species	SD bases (Mbp)	Proportion of genome
HSA_T2T	h1	T2T-CHM13v2.0 (PRJNA559484)	Human	193.9	0.062
HG002	h1	HG002 (PRJNA794175)	Human	192.7	0.062
HG002	h2	HG002 (PRJNA794172)	Human	188.9	0.061
PTR_p	h1	Clint_PTR (PRJNA941355)	Chimpanzee	184.8	0.059

PTR_p	h2	Clint_PTR (PRJNA941354)	Chimpanzee	189	0.062
PTR_n	h1	mPanTro3	Chimpanzee	197.4	0.062
PTR_n	h2	mPanTro3	Chimpanzee	193.3	0.061
PPA_p	h1	Mhudiblu_PPA (PRJNA941363)	Bonobo	177.9	0.056
PPA_p	h2	Mhudiblu_PPA (PRJNA941362)	Bonobo	171.6	0.054
PPA_n	h1	mPanPan1	Bonobo	205.4	0.063
PPA_n	h2	mPanPan1	Bonobo	205.4	0.063
GGO_n	h1	mGorGor1	Gorilla	215.3	0.059
GGO_n	h2	mGorGor1	Gorilla	207.2	0.059
PPY	h1	mPonPyg2	Bornean orangutan	222.8	0.069
PPY	h2	mPonPyg2	Bornean orangutan	227.2	0.071
PAB_p	h1	Susie_PAB (PRJNA941365)	Sumatran orangutan	201.9	0.064
PAB_p	h2	Susie_PAB (PRJNA941364)	Sumatran orangutan	178.2	0.057
PAB_n	h1	mPonAbe1	Sumatran orangutan	219.5	0.068
PAB_n	h2	mPonAbe1	Sumatran orangutan	230.8	0.071
SSY	h1	mSymSyn1	Siamang	152	0.047
SSY	h2	mSymSyn1	Siamang	152.6	0.046
ANA_p	h1	86718_ANA (PRJNA941351)	Owl monkey	127.7	0.041
ANA_p	h2	86718_ANA (PRJNA941350)	Owl monkey	126.2	0.041
MMU_p	h1	AG07107_MMU (PRJNA941359)	Macaque	109	0.035
MMU_p	h2	AG07107_MMU (PRJNA941358)	Macaque	108.2	0.035
MFA_n	h1	T2T-MFA8v1.0 (PRJNA1037719)	Macaque	112.3	0.035
CJA_p	h1	CJ1700_CJA (PRJNA941357)	Marmoset	161	0.055
CJA_p	h2	CJ1700_CJA (PRJNA941356)	Marmoset	157.7	0.054
TGE_p	h1	ASM4086918v1 (PRJNA1081469)	Gelada	152.9	0.049
TGE_p	h2	ASM4086916v1 (PRJNA1081468)	Gelada	139.5	0.045
MMUR_p	h1	Inina_MMUR (PRJNA1082316)	Mouse lemur	76.6	0.033
MMUR_p	h2	Inina_MMUR (PRJNA1082315)	Mouse lemur	68.8	0.029

We agree with the reviewer’s point on assembly quality bias in this comparison. To clarify the argument, we revised the sentences as follows:

“We also find that the assemblies of great apes, on average, have the highest SD content (208 Mbp per haplotype) when compared to non-ape lineages: assemblies of mouse lemur, gelada, marmoset, owl monkey, and macaque (68.8–161 Mbp) (**Fig. 8a & Fig. SD.S3; Table SD.S1**).”

L923, Fig. SD.S4: Assuming the r^2 value refers to an OLS linear model, it is not clear what criteria were used to conclude that the siamang or the unnamed ancestral nodes do not follow the correlation. Given the phylogenetic non-independence, the authors need to apply a phylogenetic comparative method to measure any deviation from the expected number of SDs in a lineage.

We find that the siamang and macaque show significantly lower numbers of SDs in total as compared by the two-tailed Wilcoxon test ($p=0.011$) compared to other great apes. We also examined Cook's distance from the linear model SDs bases (Mbp) vs. branch length (mya) to check deviation from the regression model and found, $D = 0.484, 0.608$ and 0.516 for SSY, CBHGOS, and MFA, respectively, exceeding the conventional cutoff of $4/n$ (0.333).

To illustrate this, we added descriptions to methods as follows:

“The correlation of SDs (Mbp) versus branch length (mya) was computed using linear regression, excluding siamang and macaque, which contain significantly lower number of SDs compared to other species (two-tailed Wilcoxon ranked sum test) and BCHGOS ancestral node affected by siamang. Examining Cook's distance also identifies significant deviation ($D > 4/n$) in the nodes.”

Referee #3 (Remarks to the Author):

The authors have generated T2T haplotype resolved assemblies for six ape species: chimpanzee, bonobo, gorilla, Bornean orangutan, Sumatran orangutan, and siamang. They resolve challenging regions, such as the major histocompatibility complex and immunoglobulin loci. Comparative analyses, including human enables them to investigate the evolution and diversity of regions previously uncharacterized or incompletely studied. These studies alleviate the previous reference bias from studies based on the human referenced genome. This includes lineage-specific segmental duplications, centromeric DNA, acrocentric chromosomes, and subterminal heterochromatin as well as regions under selection in the different species. These genomes should serve as a resource for studies of the human and ape relatives for all future studies.

The paper is sorted into three sections:

1. The genome assemblies
2. Highlights of analysis based on these assemblies
3. What is in positions in the genome previously not studied (ie assembled).

All sections are summarized in the main paper but have related supp info sections giving more information for each analysis. For the most part there is a good balance between the information given in the main text and the supp info. Both texts are overall well written.

For some of the analysis of segmental duplications, gene family expansion and regions under selection a bit more info could be given in the main text. If there are some examples of these events that can be validated experimentally that might strengthen the paper.

As suggested by the referee, we performed some additional selection analyses and investigated a few of the gene families in more detail. First, we extended our selective sweep analyses to genes that are nested inside an SD, using 100 kbp unique flanking sequences of each SD as a point of reference. Overall, we find that 36 candidate genes under selection are new gene models; seven of these are duplicated genes (*ZN682*, *HYES*, *IPSP*, *TRAC*, *MSTO1*, *CL18A*, *NPB13*) with selective sweep signatures occurring in both up and downstream sequences flanking the SDs. We have included these SD gene models as part of a new supplementary table (**Table Selection.S3**) and now mention it in the main text.

“Among the ape species, we identify 143 and 86 candidate regions for hard and partial selective sweeps (**Table Selection.S3**), respectively, along with seven duplication-overlapping genes showing signatures of selection on both unique flanking sequences of the duplications.”

We also examined the *LRPAP1* gene family expanded in gorilla. We experimentally validated the copy number expansion of *LRPAP1* in seven additional gorilla samples using fastCN (**Fig. Gene rebuttal 1**), which estimates the copy number based simply on read-depth coverage statistics. While variable among gorillas, all show higher copy number (9-16) consistent with the genome assembly when compared to other apes.

Figure. Gene rebuttal 1. FastCN estimated copy number of *LRPAP1* in short-read WGS data⁸ validates a gorilla lineage expansion.

Although we were not able to identify any signatures of selection, we did confirm *LRPAP1* gene expression for at least six of the copies. We added further description in the supplementary notes as follows:

“*LRPAP1* encodes for a protein that interacts with the LDL-receptor protein and is present in Chromosome 4 in humans. *LRPAP1* was also previously known to be associated with dementia and Alzheimer’s disease in humans^{71,72}. The ancestral copy of this gene is present in Chromosome 3 (homologous to HSA4) in gorillas. This gene family has expanded to 10 copies (1 ancestral and 9 copies), in gorilla across multiple chromosomes, all mapping to lineage-specific SD regions (**Table Gene.S5**). We also find that *LRPAP1* gene expansion carried two flanking genes, *DOK7* and *HGFAC*, although expression was only observed for *LRPAP1*. Apart from the ancestral copy in Chromosome 3, four additional copies in Chromosome 12 (copy number 1-4), two in Chromosome 14 (copy 5-6), and one

each in Chromosomes 16, 22, and Y were identified (copy 7-9; **Table Gene.S5**). It is also to be noted that two copies, one in Chromosome 12 (copy 2) and one in Chromosome 14 (copy 6), exist as solitary *LRPAP1*, without *DOK7* and *HGFAC*.

Examining the protein product sequence of *LRPAP1*, we observed high identity of 98–99.1% of peptide sequence identity for the copy numbers 1, 4, 7, intermediate identity of 83.7–91.3% for copy numbers 5, 8, 9, and relatively low identity of 65.2–79.2% for copy numbers 2, 3, 6, compared to the ancestral copy located in Chromosome 3 (**Table Gene.S5**). Investigating the gene expression status of *LRPAP1*, we observed expression of the *LRPAP1* copy in Chr12 (copy 1, 2, 4), Chr14 (copy 5, 6) and Chr22 (copy 8; **Table Gene.S5**). The latter is expressed only in the testes, suggesting tissue-specific differentiation. In terms of methylation, we observed hypomethylation of promoter regions without much difference among the different copies.”

We also add additional sentences to the main as follows:

““This includes 770–1,482 novel gene models corresponding to 315–528 gene families in the NHPs with ~68.6% corresponding to lineage-specific SDs, all supported by Iso-Seq transcripts (**Table. Gene.S1, S2**). In this list, we also find biomedical genes such as *LRPAP1* whose tissue-specific expression is suggested in **Supplementary Note VIII**, as well as human-specific genes including *NOTCH2NL*²⁹, *SRGAP2C*^{30,31}, and *ARHGAP11*³² (**Table. Gene.S7**)”

The results presented in the manuscript is novel and the six genomes will be resources valuable for a very long time.

In many places in the manuscript and supp info numbers are given with 4 significant digits. That is too much, please use two or three significant digits both in text and display items (for example extended data table 2). This should be fixed throughout the main and supp info text as well as in tables and figures.

Throughout the manuscript we reduced the number of significant digits to two or three including tables and figures.

Main text comments:

L 219 collapse -> collapsed sequence

We edited the term following the reviewer’s suggestion:

“In addition to gaps, we searched specifically for collapsed and misassembled sequences using dedicated methods (**Table Assembly.S3, Methods**).”

L224). Overall, we estimate 99.2–99.9% of each genome is completely and accurately assembled, including heterochromatin. Define how this was calculated.

The numbers are obtained by subtracting the putative erroneous bases from the total bases. The erroneous bases were obtained by summing the regions flagged by NucFreq and Flagger (union); they consist of regions aligned with higher or lower than expected read depths. In addition to this, we further investigated the error-free regions separately accounting for hard-masked rDNA arrays and updated the **Table Assembly.S3**.

Table Assembly.S3. Summary read-depth QC of genome assemblies.

Assembly	Num. of rDNA gaps	rDNA gaps	Collapsed rDNA	Non-rDNA gaps	Unlocalized contigs	Other assembly issues	Non-rDNA Issues
mGorGor1_v2.0	4	4,000,000	543,917	15,000	40,200	1,152,228	1,207,428
mPanPan1_v2.0	9	9,000,000	1,455,488	80,000	4,400	280,789	365,189
mPanTro3_v2.0	10	10,000,000	1,849,308	0	0	209,930	209,930
mPonAbe1_v2.0	15	15,000,000	2,594,102	576,256	5,800	660,093	1,242,149
mPonPyg2_v2.0	17	17,000,000	1,572,743	5,000	2,000	259,514	266,514
mSymSyn1_v2.0 / mSymSyn1_v2.1	2	2,000,000	668,087	1,150,000	3,200	430,084	1,583,284

*Issues in base pairs. rDNA related issues are highlighted in gray. rDNA gaps were intentionally placed as 1 Mbp gaps to distinguish from other assembly gaps. Collapsed rDNA are the flanking rDNA sequences around the rDNA gap, which will be flagged as ‘collapsed assembly issues’ in assembly evaluation methods. To distinguish intended rDNA-related regions from assembly issues, the “Non-rDNA assembly issues” have been calculated by excluding rDNA gaps and flanking rDNA regions. All units are in bp, except for the Num. of rDNA gaps.

Accordingly, the estimates of completely assembled regions were updated in the main as follows:

“Overall, we estimate 99.2–99.9% of each genome (**99.98-99.99% corresponding to non-rDNA bases**), is completely and accurately assembled, including heterochromatin. This is consistent with the T2T-CHM13v1.1 assembly, for which potential issues remained for 0.3% of the genome¹⁴”

L239 Table 1. Define the numbers within parenthesis in columns six and seven

The numbers in the parentheses represent the respective statistics in haplotype 2 or paternal haplotype for trio-phased genomes, bonobo and gorilla. The following information was added to the legend to clarify this information:

“The parentheses denote the values for the haplotype 2 or the paternal haplotype.”

L255-257 “Schematic of the T2T siamang genome highlighting segmental duplications (Intra SDs; blue), inverted duplications (InvDup; green), centromeric, subterminal and interstitial α -satellites (red), and other satellites (pink).” Is the siamang genome compared to human here or to all of the other apes?

The image depicted here is a stand-alone summary of the siamang without comparison to other apes. Given the extensive chromosomal rearrangement that occurred during gibbon evolution, it would be challenging to incorporate comparative chromosomal views. Wherever possible, however, we performed comparative analyses between siamang and apes for select loci (i.e., MHC - extended Fig. 3, ILS - Fig. 2, chromosomal rearrangement - Fig.3, acrocentrics - Fig. 5, subterminal heterochromatins - Fig.7, and lineage-specific segmental duplications - Fig. 8).

L269-270 We annotated over 18million base pairs for chromosome Y. What does “annotate” mean here?

Original: “We annotated over 18 million base pairs for chromosome Y, which is 4.67 times the annotated base pairs in the Ensembl annotation.”

“Annotate” refers to the reconstructed ancestral sequence. We removed this text for clarity.

L462. Including ORF genes means what?

The term, “ORF genes” was originally referring to “in-frame genes without stop codons.” The statement containing the terminology has been removed based on suggestions from other reviewers.

L503 “The remaining 4% are discordant topologies that cluster within 200–500 kbp regions.” Are these in chunks?

Thank you for pointing out this confusing statement. The 4% discordant topologies in fact cluster within 395-1,275 kbp regions. We revised this sentence to accurate sizes. Also, to clarify on these regions, we confirm that these are the >100 kbp regions that are enriched by 5 kbp bins with discordant tree topologies (4-19) that are less than 500 kbp from each other. We added Table MHC.S3 presenting the precise coordinates.

Table MHC.S3. Coordinates of variable regions with discordant topologies.

Regions	Chromosome (T2T-CHM13)	Start	End	Size	Region name
Variable region	chr6	28669878	29944878	1275000	VR1

	chr6	30924878	31319878	395000	VR2
	chr6	32274878	32934878	660000	VR3
Narrower regions (coalescent tree)	chr6	29295878	29302378	6500	I
	chr6	30925378	31026878	101500	II
	chr6	32310378	32325378	15000	III
	chr6	32916878	32937878	21000	IV

Also, we add additional description in the Methods to describe the variable regions:

“We identified three regions that are enriched by 5 kbp bins with discordant tree topologies (4-19) by merging discordant bins that are within 500 kbp distance from each other (**Table MHC.S3**). For the divergence time estimates, we narrow down to >50 kbp regions after merging discordant 500 bp bins within 10 kbp distance to identify four regions. From these four regions, we estimated the coalescence time. To estimate within species variation, we repeated the analysis after adding more genomes from Mao et al., 2024¹¹⁵, and also repeating the procedure of defining four subregions by merging them to the nearest 10 kbp of discordant bins. The divergence trees were calibrated by *Macaque - Human* split in all the four trees and *Siamang - Human* split in the second tree (II) (**Figure MHC.S9**). We only used *Siamang - Human* split once in a single tree, as it was the only topology where *Siamang* (Jambi_SSY) was the outgroup to African great apes and orangutans.”

L 620 We identified 14,210 AQER sites (Table AQER) across four primate lineages, including 3,268 on the human branch. So these are divergent sites, but how do you know if they are functional or just random changes happening on the different lineages? How sensitive to faulty alignments is this analysis? Having experimental support would be great but maybe not trivial.

We agree that this is the big question related to the AQERs. These highly divergent regions are likely to be a mixture of the two things the reviewer described (i.e., functionally important changes and random neutral changes in regions with higher mutation rates). Supporting many of the HAQERs being functional are the overlaps and enrichments for epigenetic marks associated with gene regulatory activity (e.g., bivalent promoters and enhancers) across a number of human cell lines and tissues (main text Fig. 4). In our initial publication about HAQERs, we developed a multiplex *in vivo* single-cell enhancer assay and demonstrated that 7 of the 13 HAQERs we tested showed a functional change in their ability to act as neurodevelopmental enhancers when we compared the inferred human–chimp ancestral sequence¹⁹. Additionally, we tested several more HAQERs in a new multiplex *in vivo* single-cell enhancer assay in developing brain tissue (see figure below). Of the eight HAQERs we recently tested, seven showed both overall and cell-type-specific enhancer activity. We plan to publish these results in an upcoming companion paper.

We anticipate that the analysis will be sensitive to faulty alignments of the human genome, while other species are properly aligned. This will result in an apparent divergence of humans due to a faulty alignment. To reduce the chance of this occurring, we use stringent alignment and filtering parameters, along with requiring all alignments to be at least 20 kbp in length¹⁹. To make this clear, we include the following sentence in the supplemental methods:

“We used stricter scoring and filtering parameters because our method is sensitive to false-positive alignments; misalignments can often appear as regions of rapid divergence.”

L672 New genomic regions – maybe say “newly identified”, “newly examined” or “newly assayed” regions

We rephrased the title following the reviewer’s suggestion:

“Section III. Newly characterized regions”

L815 “over <2mya” is confusing. Please explain/rephrase

We rephrased the wording following the reviewer’s suggestion:

“Thus, in the short evolutionary time since bonobo and chimpanzee divergence (1-2mya), a new α -satellite HOR has arisen and expanded to become the predominant HOR distinguishing these two closely related species (Fig. 6e).”

L841panel a – should it be marked that the figure is based on HSA1, HSA2 etc (not just labelled chrom 1)

This is a great suggestion. We added “HSA” beside the chromosome name in Fig. 6a to indicate that we are using human chromosome nomenclature. We also made the same modification to the supplemental figure for consistency and have provided both below for convenience.

Figure 7a:

Figure CEN.S1:

L859 In total, these account for 1.05 Gbp of subterminal satellite sequences (642 Mbp or 18.2% of the siamang genome). How to these numbers match with the total genome size?

The breakdown of 1.05 Gbp per species = 270.0 Mbp (PTR; out of diploid genome of 6.166 Gbp) + 261.6 Mbp (PPA; out of 6.278 Gbp) + 522.9 Mbp (GGO; out of 6.903 Gbp). Therefore, subterminal satellites constitute 4.38, 4.17 and 7.58% of PTR, PPA and GGO genomes, respectively. For the siamang, 642 Mbp is 10.1% of its diploid genome size, 6.357 Gbp (previous percentage was showing for haploid genome size).

We added this detail as follows:

“In total, the subterminal satellites account for 1.05 Gbp (270.0, 261.6, and 522.9 of chimpanzee, bonobo, and gorilla, respectively, or 4-7%) of the genome. In the case of the siamang, 642 Mbp (10.1%) of the genome is made up of subterminal satellites (**Extended Fig. 1**).”

Comments on supplemental notes:

Page 9 Read mapping “The X was included to the paternal and Y to the maternal haploid genome.” What does this mean? Why?

We added more detail to explain the motivation to include both sex chromosomes in the all-to-hap alignments:

“The X was included to the paternal (or hap2) and Y to the maternal (or hap1) haploid genome in the all-to-hap alignment to prevent over-polishing of the highly diverged sex chromosomes.”

Page 23: “Autosome SNV divergence between human and NHPs was lowest for human-chimpanzee and human-bonobo (0.15-0.16%), then human-gorilla (0.19-0.20%) and lastly human-orangutan (0.36%)”. Which orangutan?

This is compared to Sumatran orangutan. We edited the text to clarify the genomes used:

“human-Sumatran orangutan”

Page 39: “We used the same mutation rate of TRAILS analysis and the following generation times:” Where/how did you get generation times?

Generation times were extracted from Table S1 in Rivas-González et al. 2023, and these, in turn, were extracted “from IUCN red list reports when available (<https://www.iucnredlist.org/species>)”. We added this information to Supplementary Note VI.

Page 46: “The effective population time through time is shown in Fig. ILS.S5” Do you mean population size?

Thank you for pointing this out. We corrected this.

Page 49: “This gene undergoes copious expansion across chromosomes, all of these falling in lineage-specific SD regions” Any biological interpretation if this does something or is just a random event? If it does something maybe add to main text.

We have no functional insights on this gene family but did not make it the focus of experimental validation in additional gorilla genomes and investigated its expression as discussed above. To reiterate, *LRPAP1* is an important gene previously known to be associated with dementia and Alzheimer's disease in humans^{29,30}. We first validated the copy number of this gene across multiple ape Illumina WGS data from Prado-Martinez et al., 2013⁸, as shown above (**Fig. Gene rebuttal 1**). Although we could not find overlap with selective sweep signatures around copies of *LRPAP1* genes, investigating the product protein sequence, we observe high identity of 98-99.1% of peptide sequence identity for the copy number 1, 4, 7, intermediate identity of 83.7-91.3% for copy 5, 8, 9, and relatively low identity of 65.2-79.2% for copy 2, 3, 6, relative to the original copy in chr3.

We also investigated expression and methylation of *LRPAP1* genes, which are expanded across different chromosomes. We observed expression in the original copy of *LRPAP1* located in chromosome 3. In the expanded copies, we observed expression in chromosome 12 (copy 1, 2, 4), chromosome 14 (copy 5, 6) and chromosome 22 (only testes; **Table Gene.S5**), which could suggest the possibility of functional effect due to differential expression; however, we also believe the function of this gene needs to be more carefully studied. In terms of methylation, we observed hypomethylation of promoter regions without much difference among the different copies.

We added the further description to the supplementary notes as follows:

“*LRPAP1* encodes for a protein that interacts with the LDL-receptor protein and is present in Chromosome 4 in humans. *LRPAP1* was also previously known to be associated with dementia and Alzheimer's disease in humans^{71,72}. The ancestral copy of this gene is present in Chromosome 3 (homologous to HSA4) in gorillas. This gene family has expanded to 10 copies (1 ancestral and 9 copies), in gorilla across multiple chromosomes, all mapping to lineage-specific SD regions (**Table Gene.S5**). We also find that *LRPAP1* gene expansion carried two flanking genes, *DOK7* and *HGFAC*, although expression was only observed for *LRPAP1*. Apart from the ancestral copy in Chromosome 3, four additional copies in Chromosome 12 (copy number 1-4), two in Chromosome 14 (copy 5-6), and one each in Chromosomes 16, 22, and Y were identified (copy 7-9; **Table Gene.S5**). It is also to be noted that two copies, one in Chromosome 12 (copy 2) and one in Chromosome 14 (copy 6), exist as solitary *LRPAP1*, without *DOK7* and *HGFAC*.”

Examining the protein product sequence of *LRPAP1*, we observed high identity of 98–99.1% of peptide sequence identity for the copy numbers 1, 4, 7, intermediate identity of 83.7–91.3% for copy numbers 5,

8, 9, and relatively low identity of 65.2–79.2% for copy numbers 2, 3, 6, compared to the ancestral copy located in Chromosome 3 (**Table GeneS5**). Investigating the gene expression status of *LRPAP1*, we observed expression of the *LRPAP1* copy in Chr12 (copy 1, 2, 4), Chr14 (copy 5, 6) and Chr22 (copy 8; **Table GeneS5**). The latter is expressed only in the testes, suggesting tissue-specific differentiation. In terms of methylation, we observed hypomethylation of promoter regions without much difference among the different copies.”

In addition, we mention the relevant genes in main as follows:

“This includes 770–1,482 novel gene models corresponding to 315–528 gene families in the NHPs with ~68.6% corresponding to lineage-specific SDs, all supported by Iso-Seq transcripts (**Table. Gene.S1, S2**). In this list, we also find biomedical genes such as *LRPAP1* whose tissue-specific expression is suggested in **Supplementary Note VIII**, as well as human-specific genes including *NOTCH2NL*²⁹, *SRGAP2C*^{30,31}, and *ARHGAP11*³² (**Table. Gene.S7**).”

Page 51: “Then for a given gene in T2T, either protein-coding (Fig. Gene.S3f) or member of a multicopy gene family (Fig. Gene.S3g), we find...” Unclear what the difference between protein-coding and gene family is.

We rephrased the statement to refer to two different types of analyses, one focusing on the similarity of individual protein-coding genes and one on copy number counts:

“when comparing protein-coding genes (**Fig. Gene.S3f**) or copy number counts (**Fig. Gene.S3g**) to previous assemblies, we find the read bundle spanning a given gene in T2T ape assemblies by querying an interval tree (<https://github.com/chaimleib/intervaltree>) that stores all T2T bundles, with the start and end coordinates of the gene”

Page 59: Figure Species-Specific MEI Classifications. Color should be red not magenta in label

The text has been corrected following the reviewer’s comment.

Page 115: “For primate lineages (i.e., PTR, PAB, GGO, PAB, PPY, SSY), sequences that were not aligned or aligned of identity.” PAB listed twice, incorrect?

The typo has been corrected “PAB” → “PPA”

Page 135: “In addition to the subterminal satellites, spacer SDs interrupting the satellites were investigated were investigated. “ repeated were investigated

Thank you for carefully proofing the document. We fixed the typo.

Page 136: Add more info to figure legends for both figures

The figure legend has been edited following the reviewer's comment:

“Figure Subterminal.S1. Sequence organization of the subterminal spacer SDs and their ortholog copies. The top shows the stacked plot of gorilla spacer SDs, followed by the ancestral orthologous/paralogous copies of other species below. Below is the subterminal *Pan* lineage spacer SDs followed by the orthologous/paralogous copies of other species. The color scheme represents different classes of repeats; red: LTR, blue: DNA, green: simple repeat, purple: SINE, orange: LINE, yellow: single recognition particle RNA, brown: retrotransposon, gray: low complexity. The bottom right shows the distribution of the spacer SD sizes identified across the diploid genomes.

Figure Subterminal.S2. Epigenetic property of spacers, forming hypomethylated pockets, similar to African great ape spacers. The example Chr1 p and q arm spacer SDs are shown as the genome browser format. For each view, top shows the CpG methylation track indicating % of CpGs in the locus methylated. Below that shows α -satellite arrays are indicated by the red track followed by interchromosomal SDs.”

Response References

- 1 Mao, Y. *et al.* Structurally divergent and recurrently mutated regions of primate genomes. *Cell* **187**, 1547-1562. e1513 (2024).
- 2 Sollis, E. *et al.* The NHGRI-EBI GWAS Catalog: knowledgebase and deposition resource. *Nucleic acids research* **51**, D977-D985 (2023).
- 3 Potapova, T. A. *et al.* Epigenetic control and inheritance of rDNA arrays. *bioRxiv*, 2024.2009.2013.612795 (2024).
- 4 Logsdon, G. A. *et al.* Complex genetic variation in nearly complete human genomes. *bioRxiv*, 2024.2009.2024.614721 (2024).
- 5 Hartley, G. A. *et al.* Centromeric transposable elements and epigenetic status drive karyotypic variation in the eastern hoolock gibbon. *bioRxiv*, 2024.2008.2029.610280 (2024).
- 6 Novo, C. *et al.* The heterochromatic chromosome caps in great apes impact telomere metabolism. *Nucleic acids research* **41**, 4792-4801 (2013).
- 7 Hirai, H. *et al.* Structural variations of subterminal satellite blocks and their source mechanisms as inferred from the meiotic configurations of chimpanzee chromosome termini. *Chromosome Research* **27**, 321-332 (2019).
- 8 Prado-Martinez, J. *et al.* Great ape genetic diversity and population history. *Nature* **499**, 471-475 (2013).
- 9 Pendleton, A. L. *et al.* Comparison of village dog and wolf genomes highlights the role of the neural crest in dog domestication. *BMC biology* **16**, 1-21 (2018).
- 10 Hedrick, P. W. Adaptive introgression in animals: examples and comparison to new mutation and standing variation as sources of adaptive variation. *Molecular ecology* **22**, 4606-4618 (2013).
- 11 Ségurel, L. & Quintana-Murci, L. Preserving immune diversity through ancient inheritance and admixture. *Current Opinion in Immunology* **30**, 79-84 (2014).
- 12 Marchal, C. *et al.* Genome-wide analysis of replication timing by next-generation sequencing with E/L Repli-seq. *Nature protocols* **13**, 819-839 (2018).
- 13 Koren, S. *et al.* Gapless assembly of complete human and plant chromosomes using only nanopore sequencing. *bioRxiv* (2024).
- 14 Rivas-González, I. *et al.* Pervasive incomplete lineage sorting illuminates speciation and selection in primates. *Science* **380**, eabn4409 (2023).
- 15 Carbone, L. *et al.* Gibbon genome and the fast karyotype evolution of small apes. *Nature* **513**, 195-201 (2014).
- 16 Keller, T. E. & Yi, S. V. DNA methylation and evolution of duplicate genes. *Proceedings of the National Academy of Sciences* **111**, 5932-5937 (2014).
- 17 Hoyt, S. J. *et al.* From telomere to telomere: The transcriptional and epigenetic state of human repeat elements. *Science* **376**, eabk3112 (2022).
- 18 Moldovan, J. B. *et al.* Variable patterns of retrotransposition in different HeLa strains provide mechanistic insights into SINE RNA mobilization processes. *Nucleic Acids Research*, gkae448 (2024).
- 19 Mangan, R. J. *et al.* Adaptive sequence divergence forged new neurodevelopmental enhancers in humans. *Cell* **185**, 4587-4603. e4523 (2022).
- 20 Xie, K. T. *et al.* DNA fragility in the parallel evolution of pelvic reduction in stickleback fish. *Science* **363**, 81-84 (2019).
- 21 Nesta, A. V., Tafur, D. & Beck, C. R. Hotspots of human mutation. *Trends in Genetics* **37**, 717-729 (2021).
- 22 Vollger, M. R. *et al.* Increased mutation and gene conversion within human segmental duplications. *Nature* **617**, 325-334 (2023).

- 23 Siepel, A. *et al.* Evolutionarily conserved elements in vertebrate, insect, worm, and yeast genomes. *Genome research* **15**, 1034-1050 (2005).
- 24 Tyner, C. *et al.* The UCSC genome browser database: 2017 update. *Nucleic acids research* **45**, D626-D634 (2017).
- 25 Pollard, K. S. *et al.* Forces shaping the fastest evolving regions in the human genome. *PLoS genetics* **2**, e168 (2006).
- 26 Girskis, K. M. *et al.* Rewiring of human neurodevelopmental gene regulatory programs by human accelerated regions. *Neuron* **109**, 3239-3251. e3237 (2021).
- 27 Vollger, M. R. *et al.* Segmental duplications and their variation in a complete human genome. *Science* **376**, eabj6965 (2022).
- 28 Marques-Bonet, T. *et al.* A burst of segmental duplications in the genome of the African great ape ancestor. *Nature* **457**, 877-881 (2009).
- 29 Pandey, P., Pradhan, S. & Mittal, B. LRP-associated protein gene (LRPAP1) and susceptibility to degenerative dementia. *Genes, Brain and Behavior* **7**, 943-950 (2008).
- 30 Singh, N. K. *et al.* APOE and LRPAP1 gene polymorphism and risk of Parkinson's disease. *Neurological Sciences* **35**, 1075-1081 (2014).

Responses to Referee Comments:

Referee #1 (Remarks to the Author):

I find that the manuscript has improved immensely by careful rewriting during the revision. It now reads very well, and it is very clear that it contains a large amount of novel findings. My comments and suggestions have all been addressed satisfactorily, with several additional analyses. I appreciate the carefully crafted supplement and the detailed way the genomic resources have already been made available. I see no reason for not immediately adopting these new genomes as the reference genomes.

The paper represents a monumental effort. I believe it will have an immediate impact on the field and a future impact once we understand the many new insights provided into acrocentric chromosomes, subtelomeric repeats, centromere evolution, and duplications better.

Thus, I have no further comments.

We appreciate the referee's additional review and time.

Referee #2 (Remarks to the Author):

I thank the authors for the effort they took to answer my questions concerns. The restructured manuscript they provide in this revision has greatly improved in clarity and enables the reader to much better appreciate the importance of the presented work via the stronger focus on analytical aspects in which the T2T assemblies are able to truly shine. While this has unfortunately required to move a large portion of the analyses to the supplement, I appreciate that this has not resulted in a lack of detail in the authors responses to comments on those sections.

While most of my concerns were addressed, I have remaining questions on the AQER, together with some minor in addition comments.

The authors have greatly expanded the details of their method to defined AQERS, which are now clearly explained. They also provided additional analyses to address some of my previous concerns, such as accounting for differences in branch-lengths, most of which are sensible.

We thank the reviewer for pointing out that the initial version of our methods section did not have sufficient detail, and we are glad that the reviewer now finds it to be clearly explained. We are also grateful that our initial responses helped to clarify some of the previous comments and concerns.

However, while the authors aimed to provide clarity on the impact of local variation in mutation rates on the expected number of changes by using repeats and regions with ILS as proxies, this does not address concerns raised by the lack of consideration of a windows' sequence composition when establishing the expected number of substitutions. It seems the counts required to establish an AQER are agnostic to the underlying base-composition, which is surprising given the substantial variation in frequencies at which different mutation types occur, resulting in different expectations for e.g. GC-rich vs GC-poor windows. A single threshold that averages these differences is thus likely inadequate to set an expectation of what should be consider "quickly evolving" and will potentially lead to GC-content driving the resulting AQERs, while also undercalling regions at the other end of the composition spectrum despite exhibiting an excess of substitutions. This would be consistent with e.g. their observed enrichment in (generally CpG-rich) promotor regions.

The reviewer is correct. Base composition is just one of many factors that may be contributing to mutation rate differences. To address this with respect to GC-composition, we compared the HAQER GC-composition with respect to the human genome average (**Fig. AQER.S1h**) and **include this as a supplemental figure**. Overall, the genome-wide and HAQER distributions are similar, but there are also many differences. The HAQER distribution shows greater variance, with both very AT- and GC-rich regions being more common, and also has an overall bias towards regions of greater GC-content. This confirms the reviewer's hypothesis that bivalent promoter would be expected to be enriched in HAQERs—thus, elevated divergence here may have nothing to do with selection and just a property of the region having an increased mutation rate.

Figure AQER.S1h. HAQER GC-composition. The plot compares the GC-content in 500 bp windows across the human genome (randomly sampling 1%) compared to sites defined as HAQERs. HAQERs have anomalously high GC-content.

To improve the manuscript and make the readers aware of this insight, we updated the supplemental figure to include panel h (as shown above), where we now show the GC-content of HAQERs compared to the genome as a whole. We also updated panel d to show the functional enrichments for GC-poor and GC-rich subsets of HAQERs (see below).

Figure AQR.S1d. Enrichment for bivalent chromatin states. The HAQER sets based on both gapped and T2T assemblies show enrichments for bivalent gene regulatory elements across 127 cell types and tissues. The set of HAQERs shared between the gapped and T2T sets shows an even stronger enrichment for this functional state. The repeat-free subset of T2T HAQERs, which excludes HAQERs that overlap centromeres/satellites, SDs, simple repeats, or MEIs, shows a stronger bivalent enrichment, and the complementary set shows reduced bivalent enrichments. **GC-rich HAQERs show stronger enrichments than GC-poor HAQERs. As the threshold is increased from 29 to 31, 38 or 39 mutations in a 500 bp window, the set of HAQERs shows a slightly greater enrichment for the bivalent state. These thresholds correspond to using the fastest 10 Mbp, 5 Mbp, 1 Mbp, or 500 kb as the expected rate of divergence in a 500 bp window.**

We also revised the text of the supplement so that the paragraph emphasizing the overlap with locally elevated mutation rates now includes the following (new edits highlighted in bold):

"Thus, we consider HAQERs to be a heterogeneous set, where the rapid divergence of some elements may be related to selection, while others are the product of elevated local mutation rates. To emphasize this point, we analyzed the relationship between HAQERs and genomic regions that are likely to have local increases to the expected number of mutations. **The three ways we approached this was to investigate HAQER overlap among repeats, GC-content, and incomplete lineage sorting. Repeat regions, such as simple repeats, tandem repeats, and SDs, can have elevated mutation rates^{163,164}. GC-rich HAQERs may have locally elevated mutation rates due to error-prone polymerases, polymerase slippage, or spontaneous deamination of methylated CpG dinucleotides¹⁶⁵. ILS can also have local effects on the expected number of mutations.**

We identified a repeat-free subset of HAQERs, which excludes HAQERs that overlap centromeres/satellites, SDs, simple repeats, or mobile elements. This subset should have a reduced contribution from the repeat-associated increase in mutation rate. The repeat-free subset shows a stronger enrichment for overlapping the functional categories of bivalent enhancers and promoters, and the complementary set shows reduced enrichment for these gene regulatory states (Fig. AQER.S1d). These results are consistent with the repeat-free HAQERs having a lesser contribution from elevated mutation rates, and potentially a greater contribution from selection acting on their functional effects.

Compared to the genome-wide distribution of GC-content, HAQERs show an overall bias towards regions of greater GC-content (Wilcoxon rank sum test, $p < 1e-16$; Fig. AQER.S1h). HAQERs also show more variation in their GC-content by being fourfold enriched for occurring in GC-poor (<25%) regions ($p < 1e-16$), and twofold enriched for occurring in GC-rich (>75%) regions ($p < 1e-16$). Many of these GC-poor and GC-rich HAQERs overlap AT-rich or GC-rich VNTRs (64% of the GC-poor HAQERs and 29% of GC-rich HAQERs). For this reason, the GC-poor regions may still have elevated mutation rates from the VNTRs, even though their GC-content is low. The GC-rich subset of HAQERs shows a stronger enrichment for overlapping the functional categories of bivalent enhancers and promoters, and GC-poor subset shows reduced enrichment for these gene regulatory states (Fig. AQER.S1d). The overall bias for HAQERs to occur in regions of greater GC-content may contribute to them having an overall greater rate of mutation and also be associated with their enrichment for occurring in promoter regions."

The primary issue may be that we were not clear, and in fact gave no definition, for what we mean by "quickly evolving." We now realize that this phrase may have communicated something different to the reviewer (and readers) than we had intended. In a later comment the reviewer mentioned that they felt phrasing in the supplement was an accurate description of the AQERs, so we added that phrasing to the main text as an additional sentence that provides a definition of the AQERs. The start of the section of the main now reads:

"Regions of rapid divergence. Because rapidly evolving regions can help pinpoint genes under positive selection⁴¹ or cis-regulatory elements (CREs) undergoing functional changes⁶⁶, we applied a mutation-counting approach to identify ancestor quickly evolved regions (AQERs) that diverged during ape evolution⁶⁶ (Methods; Supplementary Note XIII). **These AQERs are the most divergent regions of the genome and do not consider mutation rate differences (e.g., tandem repeats, GC-bias); AQERs are essentially agnostic to the underlying force, or forces, of divergence.**"

The authors reference additional confounding factors, such as the non-independence between mutation rates and selection, or selective pressures in regions with high mutation rates that have been deprioritized in other studies. While accounting for all confounders is probably not feasible anyway, it is not evident why this should justify not accounting for any confounder. Doing could similarly deprioritize certain regions, but still make the calls subject to the pitfalls the authors bring up,

We are interested in studying regions that have both elevated mutation rates and are also functional. For this reason, we are hesitant to downweight regions with an elevated mutation rate, given that recent case studies have highlighted the contribution of such elevated mutation rates to evolution. For example, there is evidence that CpG mutations may be preferentially used for high-altitude adaptation in birds^{1,2}. Similarly, there have been examples in humans of hypermutable VNTRs functioning as enhancers of potential adaptive significance³. Such regions have sometimes been overlooked in the past because of their hypermutability. One of the advantages of the T2T genomes is that we can systematically assess these for the first time. We realize, however, that readers may want to assess, remove, or downweight such regions. In response to the referee's comment, instead of excluding AQERS, we added additional annotation to the four supplemental tables to now include GC and repeat content (Supplementary tables AQERs.S1 & S5-7). This will allow readers to customize the set of highly divergent regions to address their own biological questions or interests.

1 Galen, S. C. et al. Contribution of a mutational hot spot to hemoglobin adaptation in high-altitude Andean house wrens. *Proceedings of the National Academy of Sciences* 112, 13958-13963 (2015).

2 Storz, J. F. et al. The role of mutation bias in adaptive molecular evolution: insights from convergent changes in protein function. *Philosophical Transactions of the Royal Society B* 374, 20180238 (2019).

3 Song, J. H., Lowe, C. B. & Kingsley, D. M. Characterization of a human-specific tandem repeat associated with bipolar disorder and schizophrenia. *The American Journal of Human Genetics* 103, 421-430 (2018).

and. Additionally, it is neither evident why the authors chose to base the thresholds on substitution rates estimated from 10mb windows, a size at which the dispersion of expected number of changes and GC-content is likely to substantially differ from those at the 500bp mainly used to define AQERs, nor what influence this cutoff has on the calls.

To describe this choice and explain how other window size would have modified the set, we added the following text to the supplement.

"We used the maximum rate of evolution (mutations/bp) in a 10 Mbp window as the expected rate of evolution. If this window size were to decrease, to say 1 Mbp, the maximum rate (mutations/bp) would increase since the fastest 1 Mbp window must be at least as fast as the fastest 10 Mbp window. The goal is to identify forces shaping the divergence of short genomic regions (e.g., 500 bp) that is unique compared to the forces shaping large genomic regions (i.e., 10 Mbp). Decreasing from a 10 Mbp window size to 5 Mbp, 1 Mbp, or 500 kb would have increased the threshold for calling HAQERs from 29 mutations to 31, 38, or 39."

We also updated panel d of the AQER.S1 figure to show how changing this threshold would have influenced functional enrichments (see below).

Figure AQER.S1d. Enrichment for bivalent chromatin states. The HAQER sets based on both gapped and T2T assemblies show enrichments for bivalent gene regulatory elements across 127 cell types and tissues. The set of HAQERs shared between the gapped and T2T sets shows an even stronger enrichment for this functional state. The repeat-free subset of T2T HAQERs, which excludes HAQERs that overlap centromeres/satellites, SDs, simple repeats, or MEIs, shows a stronger bivalent enrichment, and the complementary set shows reduced bivalent enrichments. **GC-rich HAQERs show stronger enrichments than GC-poor HAQERs.** As the threshold is increased from 29 to 31, 38 or 39 mutations in a 500 bp window, the set of HAQERs shows a slightly greater enrichment for the bivalent state. These thresholds correspond to using the fastest 10 Mbp, 5 Mbp, 1 Mbp, or 500 kb as the expected rate of divergence in a 500 bp window.

While the approach described in the methods is ultimately consistent with the authors' subsequent definition of AQERs as the "most divergent regions of the human genome, agnostic to the underlying force, or forces, of divergence", their current naming suggests they evolve more quickly than expected, which I believe is currently incompletely accounted for. Should the authors therefore choose to not consider nucleotide composition, the potentially resulting biases need to be explicitly communicated somewhere.

This is a fair point and, as discussed above, we opted to more explicitly state these biases. In particular, the term "quickly evolving" needed a better explanation so we revised the main text to read:

"Regions of rapid divergence. Because rapidly evolving regions can help pinpoint genes under positive selection⁴¹ or cis-regulatory elements (CREs) undergoing functional changes⁶⁶, we applied a mutation-counting approach to identify ancestor quickly evolved regions (AQERs) that diverged during ape evolution⁶⁶ (Methods; Supplementary Note XIII). **These AQERs are the most divergent regions of the genome and do not consider mutation rate differences (e.g., tandem repeats, GC-bias); AQERs are essentially agnostic to the underlying force, or forces, of divergence.**"

As discussed above, we also updated the supplemental text and figure to include more detail about the diverse and elevated GC-content of HAQERs. We also improved four supplemental tables to include the GC-content of each AQER so that readers are aware of this statistic and able to filter or otherwise modify the set based on this variable.

Additional questions on AQERs:

- The calls for larger window sizes use a multiplier of the same 10Mb substitution rate cutoffs, which may result in differences in sensitivity across window-sizes. Given the dependency of the observed enrichments on AQER window sizes, this might warrant consideration.

We agree with the reviewer that our methods are geared more to keep specificity the same and that sensitivity could change across the windows sizes of divergence. We include the following text in the supplement when we discuss varying window sizes of divergence:

"We generated additional HAQER sets at window sizes of 25, 50, 100, 200, 300, 400, 500, 750, 1k, 2k, 5k, 10k, and 20k bp. **There are varying levels of statistical power across these sets to detect a deviation from a set rate of divergence. The larger windows have more statistical power and greater sensitivity to detect a deviation from the set rate of divergence.**"

- I appreciate the comparison to hg38 AQERs, but am surprised by the low intersection of calls between assemblies. These differences attributed to different haplotypes in either assembly and newly resolved sequence, but it is unclear how this conclusion was reached, particularly for the T2T calls. Are the distributions of mutation counts comparable for the different sets?

Simply mapping these locations to the complete T2T genome shows that many (98.5% of new T2T HAQERs) cluster within newly resolved sequence, or hypervariable, or repetitive regions of the genome (Fig. AQER.S4). 66.3% of total 2,348 new HAQERs mapped to acrocentric, pericentric, subterminal regions and segmental duplications, while 32.2% mapped to repeats.

There are differences in the mutation count distributions between the set defined with GRCh38 and other incomplete reference genomes, and the current set of HAQERs using T2T genomes. There are many more windows of increased divergence, which we now show as figure panel AQER.S1i (and include below). A fraction of these T2T-only HAQERs are the result of polymorphism especially within subtelomeric structural variant VNTRs. We look forward to having a larger set of T2T genomes with diverse ancestries for each great ape species. That would ultimately let us distinguish intraspecies divergence from interspecies divergence.

Figure AQER.S1i. T2T primate genomes have increased our ability to identify highly divergent regions of the human genome. Here we show the distribution of divergence for windows with 24 or more mutations on the human lineage (29 is the HAQER threshold) that could be identified with incomplete genomes (Gapped) or T2T genomes (T2T). For these high levels of divergence visualized in the plot, there are consistently more windows detected with T2T genomes.

Figure AQER.S4. Genome-wide distribution of HAQERs (the hg38 shared ones in green and T2T-only ones in red). SDs (intrachromosomal and interchromosomal in blue and black, respectively) and the satellite-rich sequences (pinks and purples) are indicated to show regional bias of the newly discovered HAQERs.

Additional questions:

L134: Please drop the “all” or tone down the sentence in some other way.

We rephrased the statement as follows:

“This resource should serve as a baseline **for future evolutionary studies** of humans and our closest living ape relatives.”

L231, Figure PanGenome.S1D/S2: I appreciate the authors response to my previous comments by establishing a more appropriate background model. Could they please clarify what might cause the increase counts around scores of ~.3 in S2. The (averaged?) values of different classes are -at least to me- not necessarily expected as claimed. Several putatively functional sequence classes appear to be less constrained than the presumably neutrally evolving background in S2, which I suspect is the result of a lack of power over short branch-lengths of apes.

We find that the inflated counts near 0.3 (0.35) in Fig. PanGenome.S2 are due to alignment blocks in long tandem repeats, and we also found that the largest fraction of it is coming from centromeric alpha satellite arrays. Regarding the reviewer's comment on Fig. PanGenome.S1D, we believe this could be the result of reduced sensitivity given the divergence among the species sampled. We find that 8 out of 9 groups below conservation score of 0.27 were pseudogenes, or immunoglobulin and T-cell receptor genes which we predict to be even more variable than neutral sequences. Among the exceptions, we find the "snRNA" group to be one of the less constrained than expected but this is almost certainly a direct consequence of lack of power over short branch lengths.

Additional notes have been added to the legends to clarify on these points.

Original: "Conservation scores relative for specific gene annotation classes. Transcripts with retained introns, NMD transcripts, and protein-coding sequences are the most conserved, while pseudogenes and IGV genes are the among least conserved classes."

Revised: "Conservation scores relative for specific gene annotation classes. Transcripts with retained introns, NMD (nonsense-mediated decay) transcripts, and protein-coding sequences are the most conserved, while pseudogenes and IGV genes are the among least conserved classes. **The relatively low conservation observed for the snRNA group is expected to be due to lack of power over short branch lengths.**"

L280: Please rephrase "biomedical genes" and provide an example of the biomedical relevance for a general audience. Please clarify what tissue comparisons the claim of tissue-specificity is based on in the supplement.

We revised those sentences to be as specific as possible:

"In this list, we also find *LRPAP1* whose **paralogs show tissue-specific expression across fibroblast and testes** (Supplementary Note VIII), as well as human-specific genes **associated human evolution of the frontal cortex**, including *SRGAP2C*^{29,30} and *ARHGAP11*³¹ (Table Gene.S7) **as well as *NOTCH2NL*^{32,33} (neuronal intranuclear inclusion disease)**. The non-SD overlapping gene copies consist of 73% of transcript models that are changed in their sequence by >50% with the remaining 27% no longer translated."

L460-475/throughout: Please specify the test statistic for all p-values and other coefficients throughout the main text.

We went through the manuscript and specified the test statistics for all p-values and coefficients as follows:

Line 440 - "Of these inversions, 63.5% (724/1,140) have annotated human SDs at one or both ends of the inversion representing a significant 4.1-fold enrichment (**one-sided, simulation empirical $p < 0.001$**). The strongest signal was for inverted SDs mapping to the breakpoints (6.2-fold; **one-sided, simulation empirical $p < 0.001$**). We also observed significant enrichment of novel transcripts (Table Gene.S2) at the breakpoints of the inversions of African great apes (**one-sided, simulation empirical $p < 0.036$**)."

Line 447 - "The number of inversions assigned to each branch, based on maximum parsimony, correlates with the length of the branch (**linear regression, $R^2 = 0.77$**)"

Line 460 - "Such elements are highly enriched in repetitive DNA (e.g., 5-fold for SDs $p < 1e-30$; 2-fold for simple repeats $p < 1e-30$ **one-sided binomial test**). HAQERs also exhibit a significant enrichment for bivalent chromatin states (repressing and activating epigenetic marks) across diverse tissues, with the strongest enrichment being for the bivalent promoter state (3-fold enrichment, $p < 1e-30$ **one-sided binomial test**) (Fig. 4a; Table AQERs.S1)—a signal present to a lesser degree on the chimpanzee, bonobo, and gorilla lineages (1.3-fold, 1.5-fold, and 1.4-fold; $p < 1e-19$, $p < 1e-30$, $p < 1e-12$, **one-sided binomial test**, respectively)"

Line 668 – "we show that CDR length and centromere length correlate (**linear regression, $R^2 = 0.41$**)"

Line 746 – "In general, the number of SDs assigned to different lineages correlates with branch length (**linear regression, $R^2 = 0.927$** ; Fig. SD.S6)"

Supp. P52: Please provide definitions of "standard quality control" and the specific filters applied to SNVs and indels.

We added additional details on quality control as follows:

"Variant identification followed the T2T-chrXY ape paper method using GATK v4.4.0.0 HaplotypeCaller³² for initial calling, GenotypeGVCFs for joint genotyping, **and applying the QC filters to SNPs (" $QD < 2.0 || QUAL < 30.0 || SOR > 3.0 || FS > 60.0 || MQ < 40.0$ ") and indels (" $QD < 2.0 || QUAL < 30.0 || FS > 200.0$ ")."**

Referee #3 (Remarks to the Author):

Overall the authors have done a good job at addressing the reviewers comment. I have only two smaller comments.

We appreciate the referee's detailed comments and additional time.

Line 300 We now estimate that the autosomes of the ape genomes contain 53.21–57.99% detectable repeats,... Still has four significant digits. Please fix throughout paragraph/manuscript

We revised the following sentences to one decimal point:

Line 300 - "We now estimate that the autosomes of the ape genomes **contain 53.2–58.0%** detectable repeats,"

Line 226 – "with the greatest autosomal increase for chromosome 19 (**21.5%**; Fig. Ancestral.S1)."

Line 302 - "Autosomal 303 content is significantly lower than the sex chromosomes (X [**61.8–66.3%**] and Y [**71.1–85.9%**])"

Line 309 - "**4.9%** satellite content in Bornean orangutan (159.2 Mbp total) to **13.0%**"

Line 312 - "159 previously unknown satellite monomers (Tables Repeat.S1-S10), ranging from **0.5 to 7.1 Mbp**"

Line 325 L1s with intact ORFs (more than 500 in both orangutan species compared to 203 in gorilla). Humans and gorillas fall in between this spectrum. "Gorilla mentioned twice is that correct?"

We revised the statement as follows:

"...at least 2.5 times more L1s with intact ORFs (more than 500 in both orangutan species compared to 203 in gorilla). **Humans fall in between this spectrum.**"